# Ångström-resolution fluorescence microscopy

Susanne C. M. Reinhardt[1,2,5], Luciano A. Masullo[1,5], Isabelle Baudrexel[1,3,5], Philipp R. Steen[1,2,5], Rafal Kowalewski[1,2], Alexandra S. Eklund[1,3], Sebastian Strauss[1,2], Eduard M. Unterauer[1,2], Thomas Schlichthaerle[1,2], Maximilian T. Strauss[1,2], Christian Klein[3,4] & Ralf Jungmann[1,2✉]

Fluorescence microscopy, with its molecular specificity, is one of the major characterization methods used in the life sciences to understand complex biological systems. Super-resolution approaches[1–6] can achieve resolution in cells in the range of 15 to 20 nm, but interactions between individual biomolecules occur at length scales below 10 nm and characterization of intramolecular structure requires Ångström resolution. State-of-the-art super-resolution implementations[7–14] have demonstrated spatial resolutions down to 5 nm and localization precisions of 1 nm under certain in vitro conditions. However, such resolutions do not directly translate to experiments in cells, and Ångström resolution has not been demonstrated to date. Here we introduce a DNA-barcoding method, resolution enhancement by sequential imaging (RESI), that improves the resolution of fluorescence microscopy down to the Ångström scale using off-the-shelf fluorescence microscopy hardware and reagents. By sequentially imaging sparse target subsets at moderate spatial resolutions of >15 nm, we demonstrate that single-protein resolution can be achieved for biomolecules in whole intact cells. Furthermore, we experimentally resolve the DNA backbone distance of single bases in DNA origami with Ångström resolution. We use our method in a proof-of-principle demonstration to map the molecular arrangement of the immunotherapy target CD20 in situ in untreated and drug-treated cells, which opens possibilities for assessing the molecular mechanisms of targeted immunotherapy. These observations demonstrate that, by enabling intramolecular imaging under ambient conditions in whole intact cells, RESI closes the gap between super-resolution microscopy and structural biology studies and thus delivers information key to understanding complex biological systems.

The localization precision ($\sigma_{SMLM}$) of a target molecule in widefield single-molecule localization microscopy (SMLM)[15] is ultimately and fundamentally limited by the number of photons ($N$) collected per blinking event: $\sigma_{SMLM} \approx \frac{\sigma_{DIFF}}{\sqrt{N}}$ ($\sigma_{DIFF}$ is the s.d. of the point spread function (PSF) of the optical imaging system[16]; Fig. 1a). Multiple localizations of the same target (Fig. 1b, top) are distributed around the true position due to their finite precision. Two or more points not resolvable by SMLM produce overlapping distributions of localizations, thus precluding unique assignment of localizations to respective targets (Fig. 1b, bottom). However, if each localization could be assigned to a specific target by colour, barcode or any other molecular identity, they could be unambiguously grouped per target[2].

The centre of each group of localizations can be calculated with a precision far better than $\sigma_{SMLM}$. In essence, applying the principle of localization microscopy to distinguishable groups of $K$ super-resolved localizations, precision is increased from the s.d. ($\sigma_{SMLM}$) to the s.e.m. ($\frac{\sigma_{SMLM}}{\sqrt{K}}$). Collecting an arbitrarily large number of localizations yields an arbitrary increase in precision. Notably, this increase in precision occurs regardless of the precision achieved in individual localizations ($\sigma_{SMLM}$).

We introduce a straightforward implementation of this concept using Exchange-PAINT[17], a variant of DNA-PAINT[18], for identical target molecules (Fig. 1c). DNA-PAINT uses the programmable, repetitive but transient binding of dye-labelled 'imager' strands to their complementary 'docking' strands on target molecules of interest[9,18]. The transient nature of the binding leads to an apparent 'blinking' of the target, necessary to perform SMLM. Exchange-PAINT uses orthogonal DNA barcodes combined with imaging and washing cycles to allow for sequential target multiplexing. In our implementation we 'multiplex' a single target species by separating it into multiple, sparser subsets. By imaging them sequentially, sufficiently spaced and isolated groups of localizations are measured. Determining the centre of each group of localizations yields a resolution enhancement (Fig. 1d). We call this implementation resolution enhancement by sequential imaging (RESI), and the resulting localizations RESI localizations.

[1]Max Planck Institute of Biochemistry, Planegg, Germany. [2]Faculty of Physics and Center for NanoScience, Ludwig Maximilian University, Munich, Germany. [3]Department of Chemistry and Biochemistry, Ludwig Maximilian University, Munich, Germany. [4]Roche Innovation Center Zurich, Roche Pharma Research and Early Development, Schlieren, Switzerland. [5]These authors contributed equally: Susanne C. M. Reinhardt, Luciano A. Masullo, Isabelle Baudrexel, Philipp R. Steen. ✉e-mail: jungmann@biochem.mpg.de

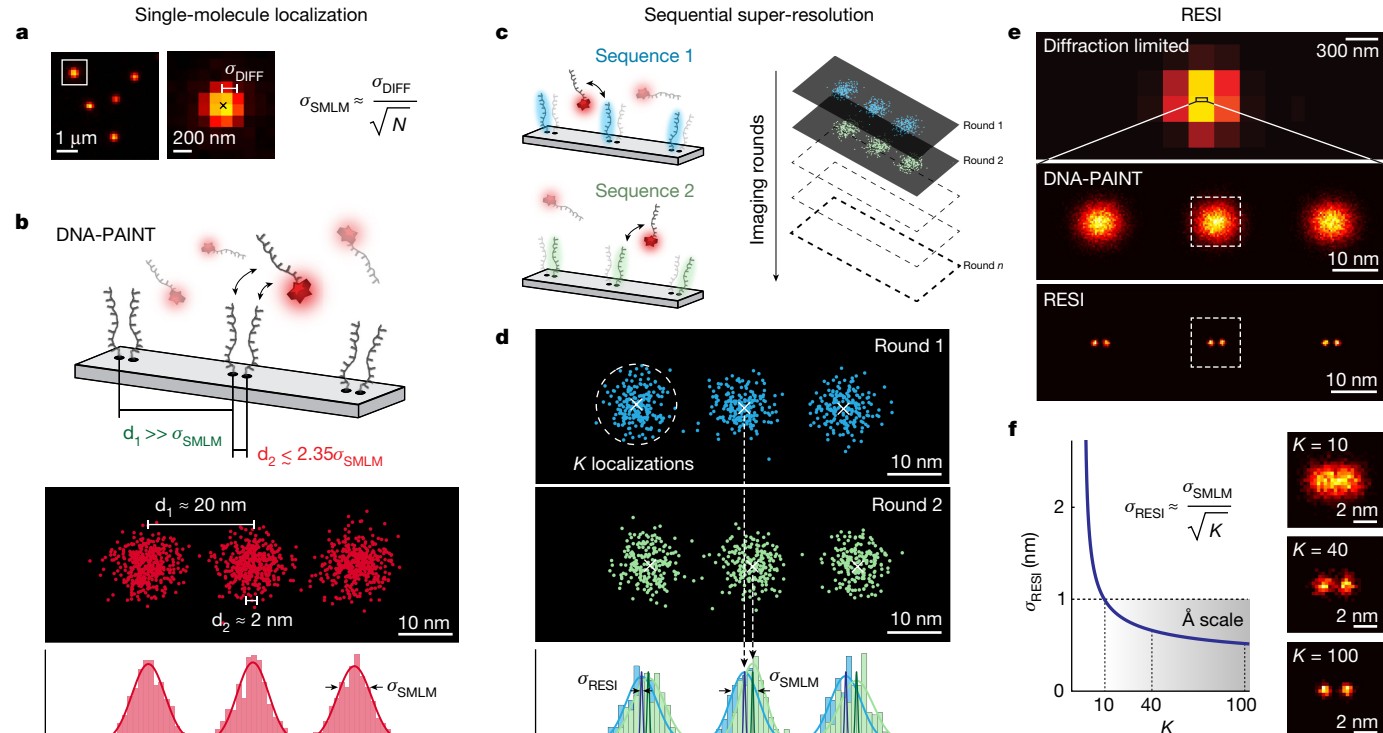

**Fig. 1 | RESI concept. a**, In SMLM, $\sigma_{SMLM}$ of a single dye scales with $\frac{\sigma_{DIFF}}{\sqrt{N}}$, ultimately limiting the achievable spatial resolution. **b**, SMLM approaches such as DNA-PAINT feature approximately 10 nm spatial resolution (resolution approximated as full-width at half-maximum $\approx 2.35\,\sigma_{SMLM}$). Whereas targets separated by 20 nm ($d_1$) can thus be routinely resolved, objects spaced 2 nm apart ($d_2$) are unresolvable because the resulting distributions of localizations overlap. **c**, Using orthogonal DNA sequences (blue and green) and sequential acquisition as in Exchange-PAINT, localizations from targets spaced more closely than the SMLM resolution limit can be unambiguously assigned for each target. **d**, Combining all localizations per target ($K$) for each imaging round improves localization precision from s.d. ($\sigma_{SMLM}$) to s.e.m. ($\sigma_{RESI}$). **e**, As super-resolution revolutionized fluorescence microscopy, RESI results in another paradigm shift by reapplying the concept of localization to super-resolution data. **f**, Localization precision in RESI scales with $\frac{1}{\sqrt{K}}$, and thus resolution improvement in RESI is independent of $\sigma_{SMLM}$, reaching localization precision on the Ångström scale.

By application of RESI in silico (Methods), we demonstrated a resolution improvement (Extended Data Fig. 1) over super-resolution akin to the improvement of super-resolved over diffraction-limited measurements (Fig. 1e). For routinely obtainable DNA-PAINT localization precision (approximately 3 nm) and number of localizations per target (in the order of hundreds), RESI could achieve precision well below one nanometre, thus entering the Ångström scale (Fig. 1f) according to $\sigma_{RESI} = \frac{\sigma_{SMLM}}{\sqrt{K}}$.

For an experimental proof of principle of RESI we used self-assembled DNA origami structures to precisely position orthogonal DNA strands[9,19]. We first designed DNA origami featuring two docking strands spaced 5 nm apart, a distance previously resolved with DNA-PAINT[7,9], to verify the accuracy and precision of RESI. Using two sequential imaging rounds and an alignment procedure (Methods) to conduct RESI, we were able to accurately recapitulate the 5 nm point-to-point distance with precision improved by a factor of $\sqrt{K_{average}} = \sqrt{381} \approx 20$ (Extended Data Figs. 2 and 3).

We next performed RESI in three dimensions (3D) using recently developed 3D DNA origami disk structures[20] and measured distances between docking strands of $2.5 \pm 0.4$ nm in $xy$ and $11.3 \pm 0.8$ nm in $z$. This demonstrates that RESI resolution enhancement applies in all three dimensions, surpassing current state-of-the-art 3D super-resolution capabilities (Extended Data Figs. 4 and 5; for imaging parameters see Extended Data Table 1).

## RESI resolves single nuclear pore complex proteins

To demonstrate the applicability of RESI in a cellular context, we next imaged structural proteins of the nuclear pore complex (NPC). As the major gatekeeper of nucleocytoplasmic transport, the NPC is a key target for structural biology research[21]. We furthermore chose the NPC as a model system because it has been well studied using a variety of imaging approaches, including cryo-electron microscopy (cryo-EM)[22], fluorescence microscopy and super-resolution techniques[23,24]. Figure 2a presents a typical diffraction-limited and DNA-PAINT image of Nup96 molecules (tagged with monomeric enhanced green fluorescent protein (mEGFP)) labelled with DNA-conjugated anti-GFP nanobodies. Nup96 is a structural NPC protein (part of the so-called Y-complex) present in eight pairs exhibiting an eight-fold symmetry on both cytoplasmic and nuclear rings, totalling 32 copies per NPC (Fig. 2b). Individual pairs of Nup96 proteins, spaced approximately 10 nm laterally and 3 nm axially, cannot be routinely resolved with current state-of-the-art super-resolution implementations[25–28]. To enable RESI, neighbouring copies of Nup96 proteins must be labelled with orthogonal DNA sequences. To this end, we opted for a stochastic labelling approach by incubating the sample with anti-GFP nanobodies, each conjugated with one out of four orthogonal sequences (Fig. 2c). We note that, with an increasing number of expected targets below the classical DNA-PAINT resolution limit, a larger number of orthogonal labelling sequences[29]—and thus imaging rounds—is necessary to guarantee sufficiently spaced groups of localizations (for details on this requirement see Methods). Sequential 3D image acquisition in four rounds led to sufficiently spaced localization groups representing single Nup96 target molecules (Fig. 2d). Subsequent RESI super-localization of these groups allowed us to routinely visualize individual copies of Nup96 proteins (Fig. 2e). We note that this was achieved across the whole field of view (roughly $67 \times 67\ \mu m^2$) totalling over 1,000 NPCs during a total image acquisition time of

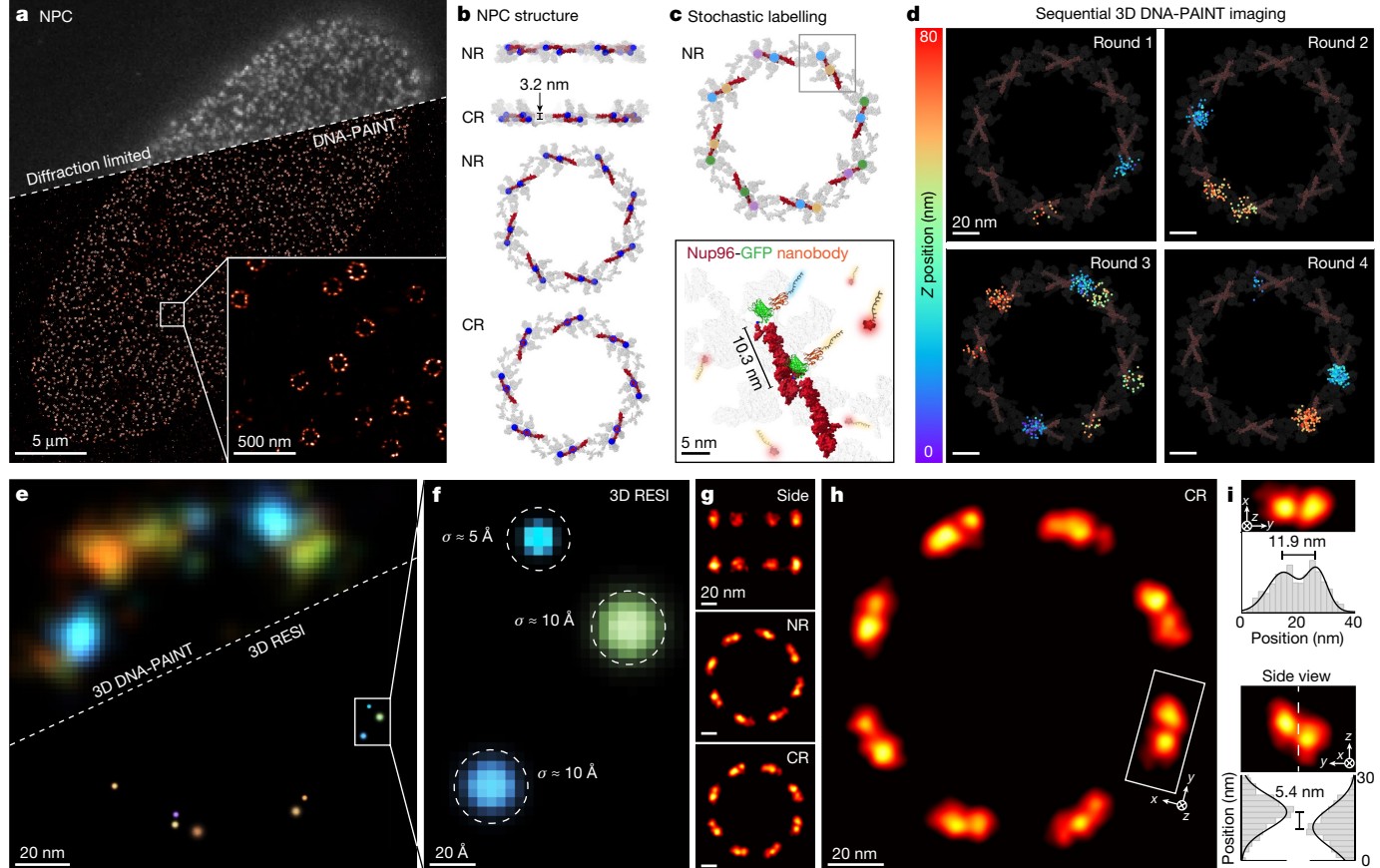

**Fig. 2 | NPC proteins in whole cells resolved with Ångström precision by RESI. a**, Diffraction-limited and DNA-PAINT overview image of Nup96-mEGFP labelled with DNA-conjugated anti-GFP nanobodies. Zoomed-in view (bottom right) shows high labelling efficiency and image quality for standard DNA-PAINT conditions, recapitulating the eight-fold symmetry of the NPC. **b**, Cryo-EM structure representation of the location of Nup96 proteins (red; C-terminal mEGFP position marked in blue) as part of the Y-complex in nuclear and cytoplasmic rings (NR and CR, respectively). Adapted from PDB 7PEQ. Nup96 is present in 32 copies per NPC. **c**, To enable RESI, Nup96-mEGFP proteins were stochastically labelled with orthogonal DNA sequences by incubation of the sample with anti-GFP nanobodies, each conjugated with one of four orthogonal sequences (represented by blue, yellow, magenta and green dots). **d**, Sequential 3D imaging (colour represents $z$ position) of the four labels yielded sufficiently

spaced localization distributions. The average number of localizations per target is $K_{average} = 38$ (background represents cryo-EM structure from **b** for context). **e**, Comparison of 3D DNA-PAINT (top left) and 3D RESI (bottom right) for the same NPC illustrating improvement in spatial resolution by RESI. Localizations are rendered as gaussians with $\sigma_{DNA-PAINT}$ and $\sigma_{RESI}$, respectively. **f**, Localization precision ($\sigma_{RESI}$) as good as 5 Å was achieved by combining $K$ localizations for each target, unambiguously resolving single Nup96 proteins. **g**, The 3D NPC cryo-EM structure was recapitulated using optical microscopy by applying a model-free average[30] of 1,217 NPCs from a single nucleus. **h**, RESI resolved adjacent Nup96 in a structural average by optical microscopy. **i**, Consistent with the cryo-EM structure (taking into account linkage error arising from label size), adjacent Nup96 proteins were spaced 11.9 ± 1.2 nm apart laterally (top) and 5.4 ± 0.4 nm axially (bottom).

100 min (see Extended Data Fig. 6 for representative data). The reconstructed RESI image features an average lateral localization precision of approximately 1 nm, representing a sixfold improvement over the individual DNA-PAINT acquisition rounds. We therefore achieved label-size-limited resolution, allowing us to resolve individual Nup96 molecules (Fig. 2f). Generally, label size not only limits spatial resolution but furthermore could lead to inaccuracies such as biased observed distances due to linkage errors.

We then performed unbiased 3D averaging of 1,217 NPCs using a recently developed model-free approach for SMLM data[30]. The resulting 3D average (Fig. 2g) not only allows recapitulation of the eight-fold symmetry of Nup96 in both cytoplasmic and nuclear rings (which has previously been achieved with super-resolution[23–28]), but enables resolution of individual Nup96 proteins in a structural average (Fig. 2h). Enabled by RESI's unprecedented spatial resolution, we were able to recapitulate distances of Nup96 proteins of 11.9 ± 1.2 nm laterally and 5.4 ± 0.4 nm axially from the structural average image (Fig. 2i). Both lateral and axial orientation, as well as tilt, of Nup96 pairs are consistent with cryo-EM data[22]. We resolved this spatial arrangement for most

Nup96 protein pairs (Extended Data Fig. 7), which was previously out of reach for optical microscopy.

## Imaging DNA bases at Ångström resolution

To assay the ultimately achievable spatial resolution by RESI, we designed a flat, rectangular DNA origami structure featuring six pairs (spaced 20 nm apart) of directly adjacent orthogonal docking strands at a distance of only one DNA base pair (red and blue strands in Fig. 3a). This yielded a designed in-plane distance of around 7 Å along the backbone of one strand of the DNA double helix[31]. The structures also contain DNA-PAINT docking strands for precise alignment between sequential imaging rounds (green strands in Fig. 3a). State-of-the-art DNA-PAINT image acquisition[32] at approximately 5 nm spatial resolution yielded six localization clouds in a 20 nm grid arrangement but failed to resolve the individual docking strands at subnanometre single-base-pair distances (Fig. 3b).

Remarkably, RESI resolves the individual docking strand positions (Fig. 3c) in all DNA origami structures. We note that RESI achieved this

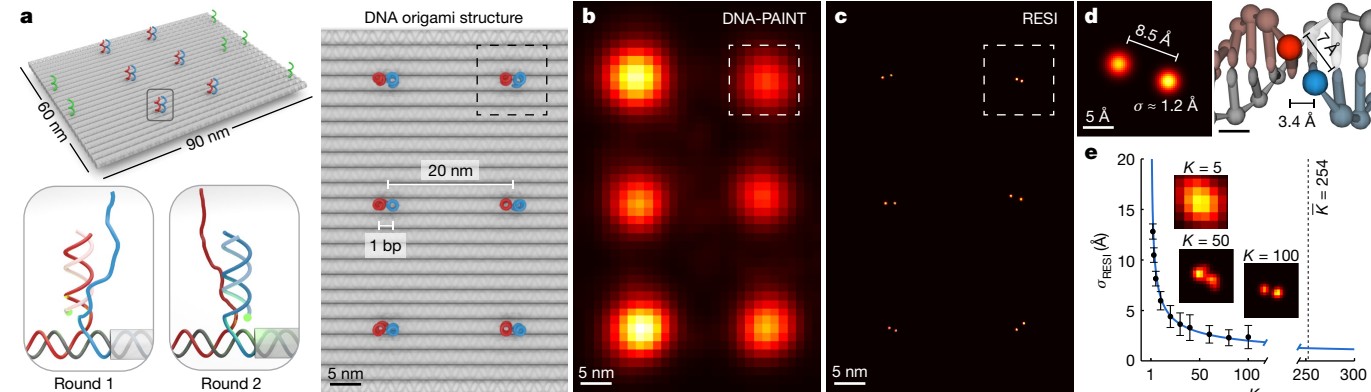

**Fig. 3 | RESI resolves the distance of single DNA base pairs at Ångström resolution. a**, DNA origami with docking strands spaced by a single base pair (bp; red and blue strands, with alignment markers in green) provided a platform to demonstrate the highest resolution achievable by RESI. **b**, DNA-PAINT resolved 20 nm spacing but the resolution was insufficient to distinguish individual docking sites, spaced one base apart. **c**, RESI resolves the adjacent docking strands. **d**, A Euclidean distance of 8.5 ± 1.7 Å was calculated from

individual localizations with an average precision of 1.2 Å (left) for the single-base-pair backbone distance, which is within 1 s.d. of the expected value of roughly 7 Å (right). **e**, Experimental localization precision in RESI is in good agreement with $\frac{\sigma_{SMLM}}{\sqrt{K}}$ (blue line, $K$), yielding an average localization precision of 1.3 Å for the experimental data from all $n$ = 42 DNA origami (insets correspond to exemplary point pair in **d**). Error bars represent mean ± 1 s.d.

in an image acquisition time of 100 min featuring an approximately $67 \times 67 \, \mu m^2$ field of view containing more than 2,000 DNA origami structures (see Extended Data Fig. 8 for representative DNA origami structures). RESI allows us to routinely resolve strands spaced apart by only one DNA base pair. Strikingly, we measured a distance of 8.5 ± 1.7 Å between two single docking strands in an individual DNA origami structure (Fig. 3d). This demonstrates an unprecedented resolution in optical microscopy by distinguishing structures closer than one nanometre. We note that our resolution claim is based on the most fundamental and strict definition: the ability to spatially distinguish point objects. We measured a distance of 9.5 ± 2.6 Å between adjacent docking strands in an average of 42 DNA origami (Extended Data Fig. 9), which is within 1 s.d. of the expected backbone distance[31] of around 7 Å.

To quantify resolution gain, we calculated RESI localizations for different values of $K$ underlying DNA-PAINT localizations (Methods). We demonstrate that the effective localization precision scales as $\sigma_{RESI} = \frac{\sigma_{SMLM}}{\sqrt{K}}$, yielding an average localization precision of 1.3 Å for an average $K$ = 254 (Fig. 3e), experimentally confirming the in silico results (Fig. 1f). RESI not only yields virtually unlimited numbers of localizations per target, but also avoids detrimental photophysical effects caused by spatial proximity of fixed-dye labels because, in DNA-PAINT imaging, two adjacent dyes are never present simultaneously. It has recently been reported[33] that, at sub-10-nm distances, photophysical near-field interactions play a major role in modulation of photoswitching kinetics, thus effectively preventing fixed-dye SMLM techniques from accessing this resolution scale. This ultimately limits the achievable resolution of even the most photon-efficient techniques available for single-molecule localization, such as MINFLUX or MINSTED, despite their subnanometre precision, unless combined with DNA-PAINT. The experimentally demonstrated subnanometre resolution illustrates the capacity of RESI to enable structural biology studies using DNA-based imaging at hitherto elusive scales.

## CD20 receptor organization

Finally, we applied RESI to address and resolve a cell-biological question currently under debate that has so far been beyond reach for both cryo-EM in a native cellular context and incumbent super-resolution techniques. Specifically we studied the organization of CD20 membrane receptors, which are prime targets for therapeutic antibody treatment of B cell-derived blood cancers and autoimmune diseases[34].

In the case of the most frequently used therapeutic anti-CD20 antibody, rituximab (RTX), the spatial rearrangement of CD20 in the cell membrane is thought to play an important role in its efficacy[35,36]. Recent cryo-EM studies detected CD20 as a dimer in complex with two individual RTX-fragment antigen-binding regions[37,38], suggesting a linear chain-like assembly of CD20 in the presence of the full antibody[38]. On the other hand, when incubated with the full RTX antibody, a trimeric ring of alternating RTX molecules and CD20 dimers was detected in EM images[37]. The fact that cryo-EM experiments are performed in detergent solution raises the question about which molecular arrangements are actually present in the cell. Currently CD20 organization when bound to full RTX antibodies in intact cells cannot be assessed, thus precluding the investigation of whether CD20 clusters are of linear or circular nature. Moreover, even though in vitro studies showed that CD20 dimers can form without antibody binding, the quantitative assessment of CD20 dimerization in untreated cells is currently limited.

Here we applied RESI to study the molecular arrangement of CD20 in Chinese hamster ovary (CHO) cells transiently transfected with mEGFP-CD20, using four rounds of probe exchange in a total imaging time of 4.4 h. In the diffraction-limited overview and DNA-PAINT super-resolution image of untreated cells, CD20 appeared homogeneously distributed (Fig. 4a,b (top)) and Extended Data Fig. 10a) whereas RTX-treated cells exhibited apparent CD20 clusters (Fig. 4b (bottom) and Extended Data Figs. 11a and 12a).

Comparison of DNA-PAINT and RESI for both untreated and RTX-treated cells shows sub-10-nm-spaced CD20 pairs in the RESI images (Fig. 4c, right) that were unresolvable with DNA-PAINT (Fig. 4c, left). RESI images suggest that CD20 is present in dimers and chain-like, higher-order structures in untreated and RTX-treated cells, respectively (Fig. 4d).

To quantitatively assess the existence of dimers in untreated cells, we performed first nearest-neighbour distance (NND) analysis for both DNA-PAINT and RESI data, demonstrating nonrandom distributions in both cases (Fig. 4e). RESI at 1 nm localization precision shows a substantial fraction of sub-10-nm distances in the NND histogram, which enables quantitative assessment of the degree of CD20 dimerization. We performed numerical simulations and a least-squares fit (Methods) that yielded a composition of 53 ± 1% monomers and 47 ± 1% dimers with average intradimer distance of 13.5 ± 0.3 nm (Fig. 4f, solid line). For comparison, we simulated NND distributions corresponding to a population of 100% monomers (Fig. 4f, dotted line),

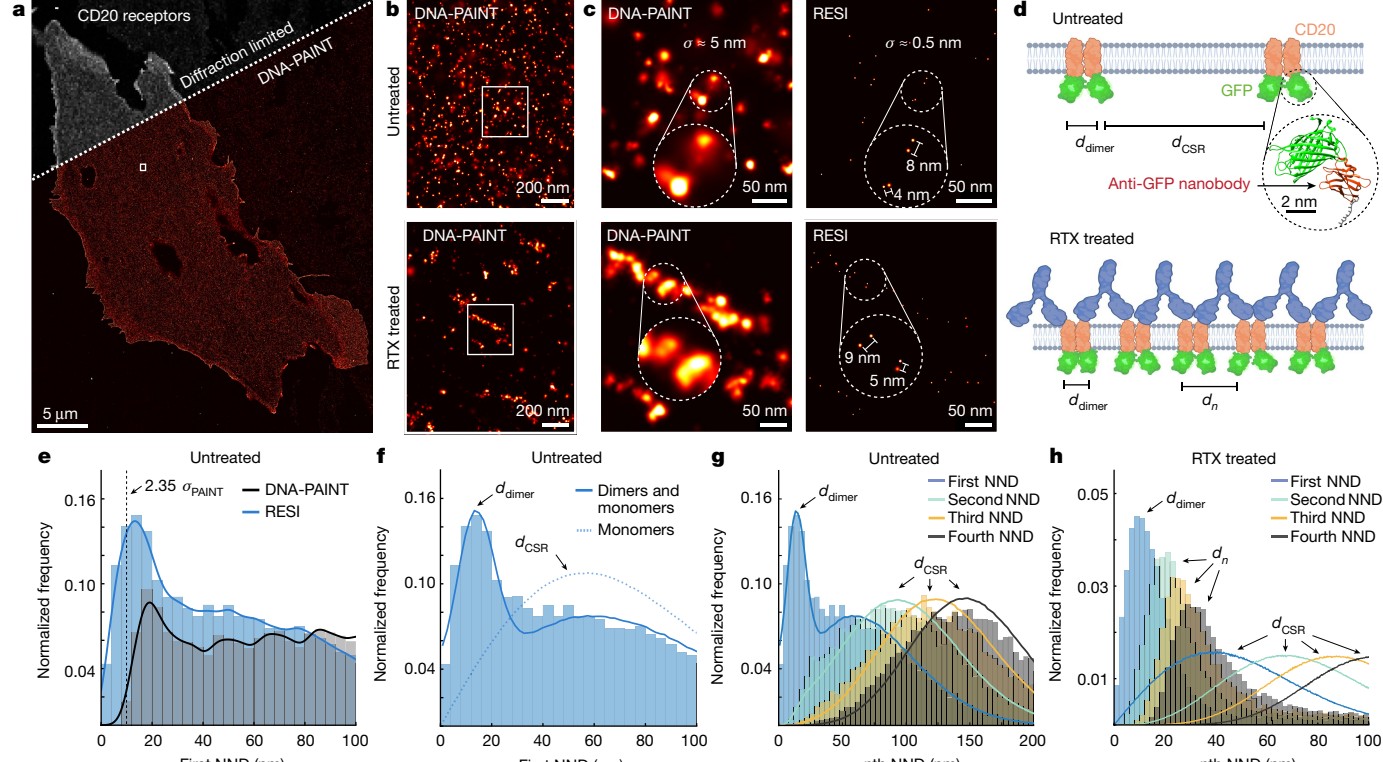

**Fig. 4 | RESI shows CD20 receptor (re)organization at subnanometre precision following drug treatment. a**, Diffraction-limited and DNA-PAINT overview image of CHO cells expressing mEGFP–CD20 labelled with anti-GFP nanobodies. **b**, Zoomed-in DNA-PAINT image showing apparently randomly distributed CD20 receptors for untreated cells (top) and clustered receptor arrangement for RTX-treated cells (bottom). **c**, Comparison of DNA-PAINT and RESI for both untreated and RTX-treated cells showing sub-10-nm-spaced receptor pairs in the RESI images, which are unresolvable with DNA-PAINT. **d**, RESI data suggest that CD20 proteins occur in dimers (spaced at $d_{dimer}$), which are in turn distributed according to complete spatial randomness (CSR; distances between dimers, $d_{CSR}$) in untreated cells. Chains of dimers were

observed following administration of RTX. **e**, Whole-cell analysis of first NNDs of CD20 receptors (histograms of distances and kernel density estimation are shown). Only RESI, but not DNA-PAINT, allows the routine detection of sub-10-nm distances between proteins. Whereas DNA-PAINT overestimates dimer distance, RESI shows a label-limited distance of 13.5 nm (see main text for discussion). **f**, Fitting RESI NND data from **e** to a numerical model reveals CD20 dimers and monomers. **g**, CD20 receptors in untreated cells showed second to fourth NNDs consistent with CSR, thus excluding the presence of higher-order protein complexes. **h**, CD20 receptors in RTX-treated cells, however, showed first to fourth NNDs, inconsistent with complete spatial randomness.

further demonstrating that CD20 molecules are not present solely as monomers. Because all NND distributions except for the first order are consistent with a complete spatial random (CSR) distribution at the experimentally measured density, we exclude the presence of higher-order assemblies for untreated CD20 (Fig. 4g). Our findings present quantitative experimental evidence that CD20 exists as dimers in an intact cell membrane.

By contrast, RESI analysis of CD20 in RTX-treated cells yielded first to fourth NND distributions inconsistent with a CSR model (Fig. 4h and Extended Data Fig. 12d,e). This suggests a higher-order arrangement of CD20 molecules after RTX treatment and confirms recent cryo-EM-derived models[37,38].

Finally we probed the existence of hexameric, ring-like arrangements by comparison with numerical simulations (Extended Data Fig. 13). The characteristics of the experimentally detected CD20 clusters suggest the absence of isolated hexamers and support the hypothesis of predominantly linear, chain-like structures (Extended Data Fig. 13h).

## Discussion

We introduce RESI, a conceptually new approach in SMLM to improve the spatial resolution of optical microscopy to the Ångström scale. RESI achieves this by combining multiple localizations from single targets to obtain a 'super-super-resolution' image after separating their localizations by sequential imaging (for example, using DNA-barcoded probes).

In this way RESI precision—and thus resolution—scales not only with the number of photons ($N$) detected per localization but also with the number of localizations ($K$) acquired per target. RESI thus provides a new precision scaling law: $\sigma_{RESI} = \frac{\sigma_{SMLM}}{\sqrt{K}} \approx \frac{\sigma_{DIFF}}{\sqrt{K}\cdot\sqrt{N}}$. This applies if a sufficiently large number of orthogonal labelling sequences and thus imaging rounds guarantee adequately spaced groups of localizations (Extended Data Fig. 14). Importantly, resolution enhancement is isotropic in three dimensions. For our current experimental implementation, RESI approaches structural biology resolution with an all-optical approach in intact cells using off-the-shelf labelling reagents and a simple inverted fluorescence microscope operated under ambient conditions. We were able to experimentally demonstrate Ångström spatial resolution below the physical size of a dye. This was achieved due to three specific advantages of DNA-PAINT leading to unbiased target sampling: (1) the rotational flexibility of the target-bound docking strand (even in the case of longer repetitive-sequence motifs[32]); (2) the freely rotating dipole of the dye attached to the imager sequence; and (3) the fact that two adjacent imagers are never present simultaneously.

Furthermore, because RESI images are not obtained from single localizations but from groups of localizations per target, the method presents a uniquely robust feature compared with other SMLM techniques: it shifts the focus from enhancement of only optical precision ($\sigma_{OPT}$) to improvement in overall precision ($\sigma_{SMLM} \approx \sqrt{\sigma_{OPT}^2 + \sigma_{MEC}^2}$) by averaging out the uncertainty effects of mechanical instability ($\sigma_{MEC}$), provided the latter is normally distributed.

With RESI we measured areas of $67 \times 67 \mu m^2$ in 100 min, making it applicable as a sufficiently high-throughput tool for cell biology. Resolving receptor patterns at single-protein resolution could enable 'spatial diagnostics' as a prescreening method for personalized treatments, and serve as a tool for biomedical discovery of patterned therapeutics—for example, by guiding drug design principles.

RESI performance and accuracy could be further improved by advances in intramolecular labelling approaches such as orthogonal unnatural amino acids[39]. RESI is thus poised to close the gap between 3D fluorescence super-resolution microscopy in whole intact cells and cryo-EM structural studies of individual supramolecular complexes, introducing a paradigm shift in fluorescence imaging by pushing optical microscopy to Ångström resolutions.

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

# Methods

## Materials

Unmodified DNA oligonucleotides, as well as DNA oligonucleotides modified with C3-azide and Cy3B, were purchased from MWG Eurofins and Metabion. The M13mp18 and p7560 scaffold was obtained from Tilibit. Magnesium chloride (1 M, no. AM9530G), sodium chloride (5 M, no. AM9759), ultrapure water (no. 10977-035), Tris (1 M, pH 8.0, no. AM9855G), EDTA (0.5 M, pH 8.0, no. AM9260G) and 10× PBS (no. 70011051) were purchased from Thermo Fisher Scientific. BSA (no. A4503-10G) was ordered from Sigma-Aldrich. Triton X-100 (no. 6683.1) was purchased from Carl Roth. Sodium hydroxide (no. 31627.290) was purchased from VWR. Paraformaldehyde (no. 15710) and glutaraldehyde (no. 16220) were obtained from Electron Microscopy Sciences. Tween-20 (no. P9416-50ML), glycerol (no. 65516-500 ml), methanol (no. 32213-2.5L), protocatechuate 3,4-dioxygenase pseudomonas (PCD, no. P8279), 3,4-dihydroxybenzoic acid (PCA, no. 37580-25G-F) and (±)-6-hydroxy-2,5,7,8-tetra-methylchromane-2-carboxylic acid (trolox, no. 238813-5G) were ordered from Sigma-Aldrich. Neutravidin (no. 31000) was purchased from Thermo Fisher Scientific. Biotin-labelled BSA (no. A8549) and sodium azide (no.769320) were obtained from Sigma-Aldrich. Coverslips (no. 0107032) and glass slides (no. 10756991) were purchased from Marienfeld and Thermo Fisher Scientific, respectively. Fetal bovine serum (FBS, no. 10500-064), 1× PBS (pH 7.2, no. 20012-019), 0.05% trypsin-EDTA (no. 25300-054), salmon sperm DNA (no. 15632011), OptiMEM (no. 31985062) and Lipofectamine LTX (no. A12621) were purchased from Thermo Fisher Scientific. Gold nanoparticles (90 nm, no. G-90-100) were ordered from Cytodiagnostics. Nanobodies against GFP (clone 1H1) with a single ectopic cysteine at the C terminus for site-specific conjugation were purchased from Nanotag Biotechnologies. DBCO-PEG4-Maleimide (no. CLK-A108P) was purchased from Jena Bioscience.

## Buffers

The following buffers were used for sample preparation and imaging.
- Buffer A: 10 mM Tris pH 8.0, 100 mM NaCl and 0.05% Tween-20
- Buffer B: 10 mM MgCl$_2$, 5 mM Tris-HCl pH 8.0, 1 mM EDTA and 0.05% Tween-20 pH 8.0
- Buffer C: 1× PBS, 1 mM EDTA, 500 mM NaCl pH 7.4, 0.02% Tween, optionally supplemented with 1× trolox, 1× PCA and 1× PCD
- Blocking buffer: 1× PBS, 1 mM EDTA, 0.02% Tween-20, 0.05% NaN$_3$, 2% BSA, 0.05 mg ml$^{-1}$ sheared salmon sperm DNA
- Two-dimensional (2D) DNA origami folding buffer: 10 mM Tris, 1 mM EDTA, 12.5 mM MgCl$_2$ pH 8.0
- 3D DNA origami folding buffer: 5 mM Tris, 1 mM EDTA, 5 mM NaCl, 20 mM MgCl2 pH 8.0
- 1× TA buffer: 40 mM Tris pH 8.0, 20 mM acetic acid

## PCA, PCD and trolox

Trolox (100×) was made by the addition of 100 mg of trolox to 430 µl of 100% methanol and 345 µl of 1 M NaOH in 3.2 ml of water. PCA (40×) was made by mixing 154 mg of PCA in 10 ml of water and NaOH and adjustment of pH to 9.0. PCD (100×) was made by the addition of 9.3 mg of PCD to 13.3 ml of buffer (100 mM Tris-HCl pH 8.0, 50 mM KCl, 1 mM EDTA, 50% glycerol).

## DNA-PAINT docking and imager sequences

Four orthogonal DNA sequence motifs were used to label targets in four RESI rounds. The docking strands were 5xR1 (TCCTCCT CCTCCTCCTCCT), 5xR2 (ACCACCACCACCACCACCA), 7xR3 (CTCTCTCTCTCTCTCTCTC) and 7xR4 (ACACACACACACACACACA). The respective imagers were R1 (AGGAGGA-Cy3B), R2 (TGGTGGT-Cy3B), R3 (GAGAGAG-Cy3B) and R4 (TGTGTGT-Cy3B). The design of 2D RESI origami required extension of the R1 site at the 5′ end such that the adjacent R1 and R3 docking strands could be spaced apart by a single base pair. Thus, the docking strand 5′ 5xR1 (TCCTCCTCCTCCTCCTCCT)

and the 5′ R1 imager (Cy3B-AGGAGGA) were used rather than the 3′ versions for both 2D DNA origamis.

## DNA origami self-assembly (2D)

All 2D DNA origami structures were designed in caDNAno[40]. Self-assembly of DNA origami was accomplished in a one-pot reaction mix with a total volume of 40 µl, consisting of 10 nM scaffold strand (for sequence, see Supplementary Data 2), 100 nM folding staples (Supplementary Data 1), 500 nM biotinylated staples (Supplementary Data 1) and 1 µM staple strands with docking site extensions (Supplementary Data 1) in 2D DNA origami folding buffer. The reaction mix was then subjected to a thermal annealing ramp using a thermocycler. First, it was incubated at 80 °C for 5 min, cooled using a temperature gradient from 60 to 4 °C in steps of 1 °C per 3.21 min and finally held at 4 °C.

## DNA origami self-assembly (3D)

The 3D DNA origami disk structure was designed in caDNAno[40]. Self-assembly of the DNA origami disk was accomplished in a one-pot reaction mix of 50 µl total volume, consisting of 20 nM scaffold strand p7560 (for sequence, see Supplementary Data 3), 200 nM core folding staples (Supplementary Data 1), 200 nM staple sequences without handle extension (Supplementary Data 1), 500 nM biotinylated staples (Supplementary Data 1), 2 µM staple strands with R4 docking site extensions and 4 µM staple strands with R1 or R3 docking site extensions (Supplementary Data 1) in 3D DNA origami folding buffer. The reaction mix was then subjected to a thermal annealing ramp using a thermocycler. It was first incubated at 80 °C for 5 min then cooled using a temperature gradient from 60 °C to 20 °C in steps of 1 °C h$^{-1}$ and finally held at 20 °C.

## DNA origami purification

After self-assembly, structures were purified by agarose gel electrophoresis (1.5% agarose, 1× TA, 10 mM MgCl$_2$, 0.5× SybrSafe) at 4.5 V cm$^{-1}$ for 1.5 h. Gel bands were cut, crushed and the origami stored in low-binding Eppendorf tubes at −20 °C.

## DNA origami sample preparation and imaging

For sample preparation, a bottomless six-channel slide (ibidi, no. 80608) was attached to a coverslip. First, 80 µl of biotin-labelled BSA (1 mg ml$^{-1}$, dissolved in buffer A) was flushed into the chamber and incubated for 5 min. The chamber was then washed with 360 µl of buffer A. A volume of 100 µl of neutravidin (0.1 mg ml$^{-1}$, dissolved in buffer A) was then flushed into the chamber and allowed to bind for 5 min. After washing with 180 µl of buffer A and subsequently with 360 µl of buffer B, 80 µl of biotin-labelled DNA structures (approximately 200 pM) in buffer B was flushed into the chamber and incubated for 5 min. For measurement of the DNA origami disk, additional 2D DNA origami structures with 12 target sites[9] spaced 20 nm apart were incubated together, with the 3D disk origami serving as fiducials for drift correction. After DNA origami incubation the chamber was washed with 540 µl of buffer B. For DNA origami disk structures, 150 µl of gold nanoparticles (diluted 1:10 in buffer B) was flushed through and incubated for 5 min before washing with 540 µl of buffer B. Finally, 180 µl of the imager solution in buffer B was flushed into the chamber. The chamber remained filled with imager solution and imaging was then performed. Between imaging rounds, the sample was washed three times with 1 ml of buffer B until no residual signal from the previous imager solution was detected. Then, the next imager solution was introduced. For RESI, two imaging rounds were performed with imagers R1 and R4 present in round 1 and the imagers R3 and R4 in round 2 (R1 and R3 probe the sites of interest for RESI and R4 serves alignment purposes).

## Nanobody–DNA conjugation

Nanobodies were conjugated as described previously[32]. Unconjugated nanobodies were thawed on ice, then 20-fold molar excess of

bifunctional DBCO-PEG4-Maleimide linker was added and reacted for 2 h on ice. Unreacted linker was removed by buffer exchange to PBS using Amicon centrifugal filters (10,000 MWCO). The DBCO-modified nanobodies were reacted with 5× molar excess of azide-functionalized DNA (R1, R2, R3 and R4) overnight at 4 °C. Unconjugated protein and free DNA were removed by anion exchange chromatography using an ÄKTA pure system equipped with a Resource Q 1 ml column.

## Cell culture

CHO cells (CCL-61, ATCC) were cultured in Gibco Ham's F-12K (Kaighn's) medium supplemented with 10% FBS (no. 11573397, Gibco). U2OS-CRISPR-Nup96-mEGFP cells (a gift from the Ries and Ellenberg laboratories) were cultured in McCoy's 5A medium (Thermo Fisher Scientific, no. 16600082) supplemented with 10% FBS. Cells were passaged every 2–3 days using trypsin-EDTA.

## Nup96 EGFP imaging

U2OS-CRISPR-Nup96-mEGFP cells were seeded on ibidi eight-well high glass-bottom chambers (no. 80807) at a density of 30,000 cm$^{-2}$. Cells were fixed with 2.4% paraformaldehyde in PBS for 30 min at room temperature. After fixation, cells were washed three times with PBS. Gold nanoparticles (200 μl) were incubated for 5 min and washed three times with PBS. Blocking and permeabilization were performed with 0.25% Triton X-100 in blocking buffer for 90 min. After washing with PBS, cells were incubated with 100 nM anti-GFP nanobodies in blocking buffer for 60 min at room temperature. To enable RESI, the nanobody solution consisted of 25 nM R1, R2, R3 and R4 docking-strand-coupled anti-GFP nanobodies with a total nanobody concentration of 100 nM. Unbound nanobodies were removed by washing three times with PBS, followed by washing once with buffer C for 10 min. Postfixation was performed with 2.4% paraformaldehyde in PBS for 15 min. After washing 3× with PBS, the imager solution in buffer C was flushed into the chamber. Between imaging rounds the sample was washed with 1–2 ml of PBS until no residual signal from the previous imager solution was detected. Then, the next imager solution was introduced. First, imagers R1, R2, R3 and R4 were added simultaneously to the sample to perform a standard DNA-PAINT measurement; then, RESI imaging was conducted via four subsequent imaging rounds with only one of the imagers.

## Cloning

mEGFP-Alfa-CD20 was cloned by insertion of Alfa-CD20 into the mEGFP-C1 plasmid (no. 54759, Addgene). An Alfa-CD20 gblock (obtained from Integrated DNA Technologies) was amplified with primers cggcatggacgagct and gtacaagtccgga and, after cutting with restriction enzymes BsrGI and BamHI, Gibson assembly was performed (2× mix, NEB).

## mEGFP-CD20 imaging

CHO cells were seeded on ibidi eight-well high glass-bottom chambers (no. 80807) the day before transfection at a density of 15,000 cm$^{-2}$. Transfection with mEGFP-CD20 was carried out with Lipofectamine LTX as specified by the manufacturer. CHO cells were allowed to express mEGFP-CD20 for 16–24 h. Then, the medium was replaced with fresh F-12K medium + 10% FBS (in the untreated case) or with F-12K medium + 10% FBS + 10 ug ml$^{-1}$ RTX-Alexa 647 (a gift from Roche Glycart) (in the RTX-treated case), followed by incubation for 30 min. After washing two times with fresh medium for 5 min, cells were fixed with 250 μl of prewarmed 4% PFA + 0.1% glutaraldehyde in PBS for 15 min. CHO cells were washed three times with PBS and quenched with 100 mM Tris pH 8.0 for 5 min. Permeabilization was carried out for 5 min with 0.2% Triton X-100 in PBS, followed by three washes with PBS. Cells were blocked in blocking buffer for 1 h at room temperature (RT). Anti-GFP nanobodies were incubated at a total concentration of 25 nM overnight at 4 °C; for RESI with four rounds this yielded 6.25 nM

each of GFP-Nb-R1/2/3/4. After washing three times with PBS at RT for 15 min, cells were postfixed with 4% PFA at RT for 10 min followed by washing and postfixation as described above. Gold nanoparticles (90 nm) were diluted 1:3 in PBS and incubated for 10 min at RT and the sample was washed two times with PBS to remove unbound gold. The imager solution in buffer C for the first round was incubated for 5 min and then replaced with fresh imager, after which the first acquisition round was started. Between imaging rounds the sample was washed with at least 2 ml of PBS until no residual signal from the previous imager solution was detected. Then, the next imager solution was introduced. RESI imaging was conducted via four subsequent imaging rounds with only one of the imagers. In the final imaging round, imagers R1, R2, R3 and R4 were added simultaneously to the sample to perform a standard DNA-PAINT measurement.

## Microscopy setup

Fluorescence imaging was carried out using an inverted microscope (Nikon Instruments, Eclipse Ti2) with the Perfect Focus System, applying an objective-type TIRF configuration equipped with an oil-immersion objective (Nikon Instruments, Apo SR TIRF ×100/numerical aperture 1.49, oil). A 560 nm laser (MPB Communications, 1 W) was used for excitation. The laser beam was passed through a cleanup filter (Chroma Technology, no. ZET561/10) and coupled to the microscope objective using a beam splitter (Chroma Technology, no. ZT561rdc). Fluorescence was spectrally filtered with an emission filter (Chroma Technology, nos. ET600/50m and ET575lp) and imaged on an sCMOS camera (Andor, Zyla 4.2 Plus) without further magnification, resulting in an effective pixel size of 130 nm (after 2 × 2 binning). The readout rate was set to 200 MHz. Images were acquired by choosing a region of interest of size 512 × 512 pixels. 3D imaging was performed using a cylindrical lens (Nikon Instruments, N-STORM) in the detection path. Raw microscopy data were acquired using μManager[41] (v.2.0.1). Total internal reflection illumination was used for 2D and 3D DNA origami data, as well as for CD20 acquisition. Highly inclined and laminated optical sheet (HILO) illumination was employed for the acquisition of NPC data. Detailed imaging conditions for the respective experiments are shown in Extended Data Table 1.

## Imaging parameters and duration

Due to target and sample heterogeneity the optimal imager concentration, $c$, used to achieve sparse blinking varies. Here we used concentrations from 100 pM (Nup96) to 800 pM (DNA origami). Optimal imager concentrations were determined visually for each sample. Concentrations were altered until blinking was frequent but sufficiently sparse to achieve good DNA-PAINT resolution.

The average number of expected binding events per binding site during a DNA-PAINT measurement is given by the duration of the measurement $t_{measurement}$ and the mean dark time $\tau_{dark}$ (defined as $\tau_{dark} = \frac{1}{k_{on} \times c}$, with $k_{on}$ being the on-rate of a given imager strand) as:

$$n_{binding\,events} = \left(\frac{t_{measurement}}{\tau_{dark}}\right) = t_{measurement} \times k_{on} \times c.$$

The average number of localizations per binding event is given by the mean bright time $\tau_{bright}$ and camera exposure time $t_{exposure}$ as

$$n_{locs\,per\,binding\,event} = \left(\frac{\tau_{bright}}{t_{exposure}}\right).$$

Therefore, the average number of localizations expected per binding site over the course of the measurement is

$$n_{loc} = \left(\frac{t_{measurement}}{\tau_{dark}}\right) \times \left(\frac{\tau_{bright}}{t_{exposure}}\right) = t_{measurement} \times k_{on} \times c \times \left(\frac{\tau_{bright}}{t_{exposure}}\right).$$

It follows that the total acquisition time necessary to collect $n_{loc}$ localizations is, on average,

$$t_{measurement} = \frac{t_{exposure} \times n_{loc}}{\tau_{bright} \times k_{on} \times c}.$$

The necessary number of localizations, $n_{loc}$, is calculated using $\sigma_{RESI} = \frac{\sigma_{DNA\text{-}PAINT}}{\sqrt{n_{loc}}}$, and thus $n_{loc} = \left(\frac{\sigma_{DNA\text{-}PAINT}}{\sigma_{RESI}}\right)^2$ with the DNA-PAINT localization precision $\sigma_{DNA\text{-}PAINT}$.

For expected imager concentrations between 50 and 800 pM, exposure times between 100 and 200 ms and kinetics reported previously[32], the times required to collect 16 localizations (1 nm RESI precision given $\sigma_{DNA\text{-}PAINT} = 4$ nm) vary between 42 s (R2, 800 pM, 100 ms exposure time) and 314 min (R5, 50 pM, 200 ms exposure time).

## DNA-PAINT analysis

Raw fluorescence data were subjected to super-resolution reconstruction using the Picasso software package[9] (latest version available at https://github.com/jungmannlab/picasso). Drift correction was performed with a redundant cross-correlation and gold particles as fiducials for cellular experiments, or with single DNA-PAINT docking sites as fiducials for origami experiments.

## Channel alignment

Alignment of subsequent imaging rounds was performed iteratively in Picasso[9], starting with a redundant cross-correlation and followed by gold fiducial alignment for cellular experiments. Every DNA origami was equipped with additional DNA-PAINT docking sites that were imaged simultaneously with the sites of interest in all imaging rounds, thus enabling their use as fiducials. First, redundant cross-correlation (2D and 3D origami measurements) and gold alignment (3D measurements) were performed in Picasso Render. To correct for nanoscopic movement of individual DNA origami during buffer exchange, channel alignment was not only performed on the full field of view but, additionally, small regions of interest containing only one DNA origami were selected. Within each region of interest, alignment was then conducted via the fiducial docking sites of the DNA origami. This was performed outside of Picasso in a custom Python script, not only to find the optimal translation between channels but also to correct for possible rotations of the DNA origami.

## Clustering and RESI

**Clustering of DNA-PAINT localizations.** After channel alignment, DNA-PAINT data were analysed using a custom clustering algorithm for each imaging round. This algorithm is based on the fact that, in DNA-PAINT, localizations are independent measurements of the position of a target molecule and are observed to be Gaussian distributed. To assign localizations to a specific target molecule, we first used a gradient ascent method to find the centre of a localization cloud for each target. We then assigned all localizations circularly distributed around the centre point to the same target molecule. This is a valid approximation because, due to the reduction of effective target density by RESI's sequential imaging approach, the majority of localization clouds from single targets are spaced sufficiently apart.

The clustering algorithm uses two input parameters: radius $r$, which sets the final size of the clusters and defines a circular environment around each localization, and the minimal number of localizations, $n_{min}$, representing a lower threshold for the number of DNA-PAINT localizations in any cluster.

First, the number of neighbouring localizations within distance $r$ from each localization is calculated. If a given localization has more neighbours within its $r$ radius than all neighbouring localizations, it is considered a local maximum. If there are more than $n_{min}$ localizations within a circle of radius $r$ around such a local maximum, these localizations are assigned to the same cluster; the remainder are not considered to be part of a cluster and are omitted from further analysis.

Further filtering of clusters is performed to exclude clusters that originate from unspecific sticking of imagers to the sample. Firstly, the mean frame (mean value of the frame numbers in which localizations occurred) of all localizations assigned to the same cluster is calculated. In the case of repetitive blinking the mean frame is expected to be around half the total number of frames[42]. The algorithm therefore excludes all clusters with a mean frame in the first or last 20% of frames. Secondly, sticking events in the middle of the acquisition time can be identified by dividing the acquisition time into 20 time windows each containing 5% of frames. If any of these time windows contains more than 80% of localizations in the cluster, it is excluded as a sticking event.

The choice of the clustering radius $r$ and the threshold $n_{min}$ depend on the respective experimental conditions. A suitable value for $n_{min}$ can be estimated by picking localization clouds originating from single target molecules (that is, well separated) in Picasso Render, exporting pick properties and plotting a histogram of the number of localizations in each pick. $n_{min}$ is chosen to differentiate between populations corresponding to single targets and to background localizations.

The radius $r$ scales with the size of the localization clouds and thus the localization precision. If too large a value is chosen, adjacent clusters might not be separated; if $r$ is too small, 'subclustering' within one localization can occur. The latter also translates to a peak in NND at twice the clustering radius. A good a priori starting value for $r$ is represented by approximately twofold the localization precision of the underlying DNA-PAINT measurement. Picasso Render offers a tool (Test Clusterer) in which the effect of different clustering parameters can be tested for a small region of interest.

For 3D clustering, an additional radius for the $z$ direction is introduced because the spread of localizations in $z$ is approximately twofold greater compared with $x$ and $y$.

**Calculation and rendering of RESI localization.** Following cluster analysis, the centres of the DNA-PAINT localization groups were calculated as weighted (wtd) means by employing the squared inverse localization precisions $\left(\frac{1}{lp^2}\right)$ as weights. For $x$ and y coordinates:

$$\bar{x}_{wtd} = \frac{\sum_{i=1}^{N} w_i x_i}{\sum_{i=1}^{N} w_i}, \; w_i = \frac{1}{lp^2}.$$

For $z$ coordinates a standard mean without weights is used to calculate $z$ positions. The precision of the resulting RESI localization is the weighted s.e.m. of the underlying grouped localizations:

$$(s_{\bar{x}})_{wtd} = \frac{(s_x)_{wtd}}{\sqrt{N}}$$

$$= \sqrt{\frac{Var(x)_{wtd}}{N}}, \text{ where } Var(x)_{wtd}$$

$$= \frac{N}{N-1} \frac{\sum_{i=1}^{N} w_i(x_i - \bar{x}_{wtd})^2}{\sum_{i=1}^{N} w_i}.$$

The choice for $1/lp^2$ as weights is based on the following argument: under the hypothesis that localizations are independent and normally distributed with the same mean, the weighted mean based on inverse variances as weights is the maximum likelihood estimator of the mean of the whole set of localizations. Therefore, the variance of the weighted mean is minimal (the estimator is optimal) when the inverse variances of individual measurements $1/lp^2$ are chosen as weights.

Finally, we take the average of the resulting $x$ and $y$ s.e.m. as the final precision of each RESI localization. For $z$ coordinates the precision is estimated to be two times $xy$ precision. Saving RESI localizations in a Picasso hdf5 file allowed us to render them as Gaussians with s.d. corresponding to their respective precision.

## RESI resolution estimation

**Evaluation of in silico RESI precision with numerical simulations.** To evaluate the performance of RESI, in silico numerical simulations were performed. The algorithm consists of the following steps.

(1) A grid of defined positions of the binding sites (ground truth) is generated. Typically, a grid of positions was generated (Extended Data Fig. 1a, top left).

(2) SMLM (DNA-PAINT) localizations are simulated as samples from a 2D Gaussian distribution with $\sigma = \sigma_{SMLM}$. A large number ($M$) of localizations is generated per binding site (Extended Data Fig. 1a, top right).

(3) For each binding site, subsets of $K$ localizations are randomly selected ($K << M$). This results in $n = \frac{M}{K}$ subsets of SMLM localizations (Extended Data Fig. 1a, bottom left) that are then averaged to generate $n$ RESI localizations (Extended Data Fig. 1a, bottom right).

(4) The resulting RESI localizations are then shown in a histogram (Extended Data Fig. 1b) and the trace ($tr$) of the covariance matrix is calculated. RESI precision is estimated as $\sigma_{RESI} = \sqrt{\frac{1}{2} tr(\mathrm{cov}(x, y))}$ (Extended Data Fig. 1c). This definition has been used before in the field as a scalar metric for 2D variance[8].

(5) Steps 3 and 4 are repeated for different values of $K$ to numerically study $\sigma = \sigma_{RESI}(K)$.

**Evaluation of experimental RESI precision by resampling of localizations.** To evaluate the precision of RESI in experimental data, an analogous method was used. Briefly, the $M$ total of DNA-PAINT localizations of each group corresponding to a single binding site was randomly resampled into subsets of $K$ localizations, then steps 4 and 5 above were performed to evaluate $\sigma_{RESI}$. The plotted $\sigma_{RESI}$ in Fig. 3d is the average value of all single binding sites in the dataset. Error bars represent the s.d. of the different $\sigma_{RESI}$ values calculated for different binding sites.

Note that this analysis can be performed only for $K << M$ to have sufficient $n = \frac{M}{K}$ RESI localizations for a statistically significant estimation. Because final RESI localization takes into account all $M$ DNA-PAINT localizations, final precision is extrapolated as $\sigma_{RESI} = \frac{\sigma_{SMLM}}{\sqrt{M}}$.

## Stochastic labelling: simulations and user guidelines

In RESI, the sparsity of binding sites in the sample is achieved by labelling a single species of biomolecules with different orthogonal DNA sequences. The labelling process is performed in a stochastic manner: $n$ different labels (for example, DNA-conjugated nanobodies) targeting the same protein species are simultaneously incubated in the sample and thus the probability of each single protein being labelled with a certain sequence $i$ ($i = 1, ..., n$) is $p_i = \frac{1}{n}$, given that the same concentration of each label is used. Subsequently, $n$ imaging rounds are performed to record all groups of localizations required to obtain the final RESI image.

The minimum number of labels ($n$) and rounds necessary to achieve sufficient sparsity of binding sites in each imaging round will depend mainly on three factors: SMLM localization precision and density and the molecular arrangement of the protein of interest. Here we describe how these parameters affect the final RESI results using a few practical examples.

**Case 1: protein structure with oligomers not resolvable with DNA-PAINT.** A typical study case is that of single proteins arranged in dimers, which in turn present another specific spatial organization in space. This is the case, for example, of the Nup96 in the NPC. In this case stochastic labelling has to be such that the probability of labelling two proteins forming a dimer with different sequences is sufficiently high. For $n$ rounds of labelling/imaging, the probability is

$$P(\mathrm{diff.\ seq.}) = 1 - p_i = 1 - \frac{1}{n}$$

for $n = 4$ labelling/imaging rounds $P(\mathrm{diff.\ seq.}) \approx 75\%$. We chose $n = 4$ to demonstrate that it provides a relatively high $P(\mathrm{diff.\ seq.})$ with only a few imaging rounds. We note, however, that $n > 4$ could be used to increase $P(\mathrm{diff.\ seq.})$ and hence to maximize the sparsity of labelled binding sites in each round.

To resolve a set of an arbitrary number of molecules, $m$, spaced more closely than the resolution of DNA-PAINT, they must be labelled with $n$ orthogonal sequences. In general, the proportion of $m$ molecules labelled with $n$ orthogonal sequences, and thus the proportion of resolvable sets of molecules, follows the equation

$$P(m, n) = \frac{n!}{(n - m)!\, n^m}.$$

**Case 2: proteins distributed similarly to CSR at a certain density.** This is a common case—for example, for membrane receptors. If proteins are distributed in a CSR fashion (Extended Data Fig. 14a) at a given density, DNA-PAINT can already resolve single proteins that are sufficiently spaced from their NNs. We will consider that proteins at a distance $d = 4 \times \sigma_{DNA\text{-}PAINT}$ are reliably resolved (note that this criterion is significantly stricter than $2.35 \times \sigma_{DNA\text{-}PAINT}$). Then, for a given density, the NND histogram can be computed and the fraction of distances below $d$ calculated (Extended Data Fig. 14b). This represents the fraction of single proteins, $F$, that will not be resolved by DNA-PAINT. Here we plot $F$ as a function of both density and resolution (Extended Data Fig. 14c). Such a map already provides a tool to understand the level of SMLM resolution needed to resolve single proteins at a given density.

RESI can be interpreted here as a way to reduce the effective density by splitting targets into different stochastically labelled subsets. Hence, the effective density of each round will be reduced according to the formula $\rho = \frac{\text{density}}{n}$. Extended Data Fig. 14d shows one-dimensional cuts of the 2D map to provide guidelines to choosing the number of orthogonal sequences (and hence imaging rounds) needed to be able to perform RESI efficiently. For example, for an initial resolution of 20 nm ($\sigma = 5$ nm), which is typical for DNA-PAINT in a cellular context, and a density of density $= 200 \frac{\text{molecules}}{\mu m^2}$ (relatively high), $n = 4$ different sequences are sufficient to provide $P(\mathrm{diff.\ seq.}) \approx 90\%$ for proteins below $d$ (Extended Data Fig. 14d). These proteins will then be resolvable by RESI.

## Model-free averaging

Model-free averaging of Nup96 data was performed for both DNA-PAINT and RESI measurement of the same nucleus, as described by Wu et al.[30]. The respective Picasso hdf5 files were segmented in SMAP[43] and saved in a file format compatible for averaging by employing plugins *segmentNPC*, *NPCsegmentCleanup* and *sitenumbers2loc*. Model-free averaging was then performed on the resulting _sml.mat files with default parameters by running the particleFusion.m script in Matlab (available with the SMAP source code). The averages shown correspond to the result of the final iteration, in which each point is rendered with a Gaussian of $\sigma = 2$ nm in $x$, $y$ and $z$.

## Numerical simulations for CD20 distribution

To interpret the results of the NND data in untreated cells, numerical simulations were performed. Briefly, two populations, one of CD20 monomers and one of dimers with a CSR distribution, were simulated and then their NNDs calculated. The algorithm can be summarized as follows:

(1) Choice of parameters. Density of monomers: number of monomers per unit area; density of dimers: number of dimers per unit area; dimer distance: expected distance between the two molecules including the labelling construct; uncertainty: variability in the position of each molecule due to labelling and localization errors; labelling efficiency: fraction of ground-truth molecules that will actually be

labelled and measured. The observed density, which has to match the experimental parameter, then becomes observed density = (density of monomers + density of dimers) × labelling efficiency. For quantification of the labelling efficiency of the DNA-conjugated GFP nanobody we used a transiently transfected CHO cell line expressing a GFP- and Alfa-tag at the C terminus of a monomeric membrane protein (for example, CD86). We then labelled GFP- and Alfa-tag using their cognate nanobodies conjugated to two orthogonal docking sequences and performed two rounds of Exchange-PAINT. We then obtained the best-fitting parameters for a sample comprising pairs of GFP/Alfa-tag, and isolated Alfa-tags, similarly to how CD20 dimer/monomer analysis is performed. The ratio of these two populations is then used as an estimation of labelling efficiency. Full details of the quantification approach will be available in a manuscript currently in preparation.

(2) Simulation of monomers: a set of spatial coordinates with CSR distribution and given density are drawn; simulation of dimers: a set of spatial coordinates with CSR distribution are drawn, representing the centre of each dimer. For each dimer centre, two positions are generated with a random orientation and a distance with expected value dimer distance. The position of each pair of molecules is drawn, taking into account the uncertainty parameter (drawn from a Gaussian distribution).

(3) A random subset of 'detectable' molecules is taken from the ground-truth set (fraction = labelling efficiency) to simulate the labelling process.

(4) NNDs are calculated on the subset of detectable molecules.

The parameters density of monomers = 212 $\mu m^{-2}$, density of dimers = 0 $\mu m^{-2}$, uncertainty = 5 nm and labelling efficiency = 50% were used to compare data for RTX-treated cells with a CSR distribution of monomers.

For the untreated case, the best-fit parameters were obtained through an iterative, nonlinear, least-squares algorithm. The experimentally observed density (50 molecules $\mu m^{-2}$) is used for the simulation.

### Description of the iterative nonlinear, least-squares algorithm
For every set of parameters a simulation is performed, NNDs are histogrammed and the sum of the squared differences between the simulation and experimental histogram are computed. A fit consists of finding the parameters that minimize the sum of the squared differences.

### Parameters
- D, average dimer distance (nm)
- σ_label, variability introduced by the labelling (nm)
- frac_of_dimers, fraction of dimers (%)

Note: frac_of_monomers = 100 − frac_of_dimers

### Estimation of parameters.
(1) Coarse-fit over a large range of parameters to determine the range of the best-fit parameters. Range D = 1–20 nm, σ_label = 1–20 nm, frac_of_dimers = 0–100%.
(2) Fine-fit over a reduced parameter space around the best-fit results in the previous step.

The parameters D_opt, σ_label_opt and frac_of_dimers_opt that best match the proposed model and the data are now found. In this case it resulted in D_opt = 13.5 nm, σ_label_opt = 5.5 nm, frac_of_dimers_opt = 47% (Fig. 4e,f).

### Estimation of parameter uncertainty.
(1) *M* is created (in this case, *M* = 100), simulated (using datasets D_opt, σ_label_opt, frac_of_dimers_opt) with the same number of molecules as the experimental data (around 21,000).

(2) *M* datasets are fine-fitted and the best-fit parameters D_opt, σ_label_opt and frac_of_dimers_opt are obtained. Three sets are obtained: D_opt, σ_label_opt and frac_of_dimers_opt.

(3) The distributions of D_opt, σ_label_opt and frac_of_dimers_opt are studied. Standard deviation can be used as an estimation of the parameter uncertainties obtained in b.

The uncertainties of the parameters D_opt, σ_label_opt and frac_of_dimers_opt are now obtained.

## Data availability
Localization data from this study are available at Zenodo (https://doi.org/10.5281/zenodo.7795826). Raw microscopy data obtained during this study are available from the corresponding author on reasonable request.

## Code availability
RESI can be performed using Picasso v.0.6.0, available at https://github.com/jungmannlab/picasso with documentation provided at https://picassosr.readthedocs.io/en/latest/render.html. The custom-written scripts used in this study are available at https://github.com/jungmannlab/resi.

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

**Acknowledgements** We thank Y.-L. Wu and J. Ries for valuable assistance with model-free averaging. We thank J. Schmied and F. Schueder for helpful discussions. We thank M. K. Steen-Mueller and I. Glueck for proofreading the manuscript. This research was funded in part by the European Research Council through an ERC Consolidator Grant (ReceptorPAINT, grant agreement no. 101003275), the German Research Foundation through SFB1032 (project A11, no. 201269156), the Danish National Research Foundation (Centre for Cellular Signal Patterns, DNRF135), the Human Frontier Science Program through a Young Investigator Grant (no. HFSP RGY0065/2018), the Volkswagen Foundation through the initiative 'Life?'—A Fresh Scientific Approach to the Basic Principles of Life' (grant no. 98198), the Max Planck Foundation and the Max Planck Society. S.S. and T.S. acknowledge support by the QBM graduate school. S.C.M.R., I.B., P.R.S., A.S.E., E.M.U. and M.T.S. acknowledge support by the IMPRS-LS graduate school. I.B. acknowledges funding support by Roche. L.A.M. acknowledges a postdoctoral fellowship from the European Union's Horizon 20212022 research and innovation programme under Marie Skłodowska-Curie grant agreement no. 101065980.

**Author contributions** S.C.M.R. designed and conducted 2D and 3D DNA origami as well as Nup96 experiments, developed the analysis software and analysed DNA origami and Nup96 data. L.A.M. designed and conducted computer simulations, contributed to analysis software and analysed DNA origami, Nup96 and CD20 data. I.B. designed and conducted CD20 experiments and analysed CD20 data. P.R.S. designed and conducted 2D DNA origami experiments, contributed to analysis software and analysed DNA origami and Nup96 data. R.K. and T.S. contributed to analysis software. S.S. developed labelling probes. A.S.E., S.S., E.M.U. and M.T.S. performed preliminary RESI experiments. C.K. contributed to the design of studies targeting CD20 and their interpretation. S.C.M.R., L.A.M., I.B., P.R.S. and R.J. interpreted data and wrote the manuscript. R.J. conceived the concept, designed experiments and supervised the study. S.C.M.R., L.A.M., I.B. and P.R.S. contributed equally. All authors reviewed and approved the final manuscript.

**Funding** Open access funding provided by the Max Planck Society.

**Competing interests** C.K. declares employment, patents (unrelated to this work) and stock ownership with Roche.

**Additional information**
**Correspondence and requests for materials** should be addressed to Ralf Jungmann.

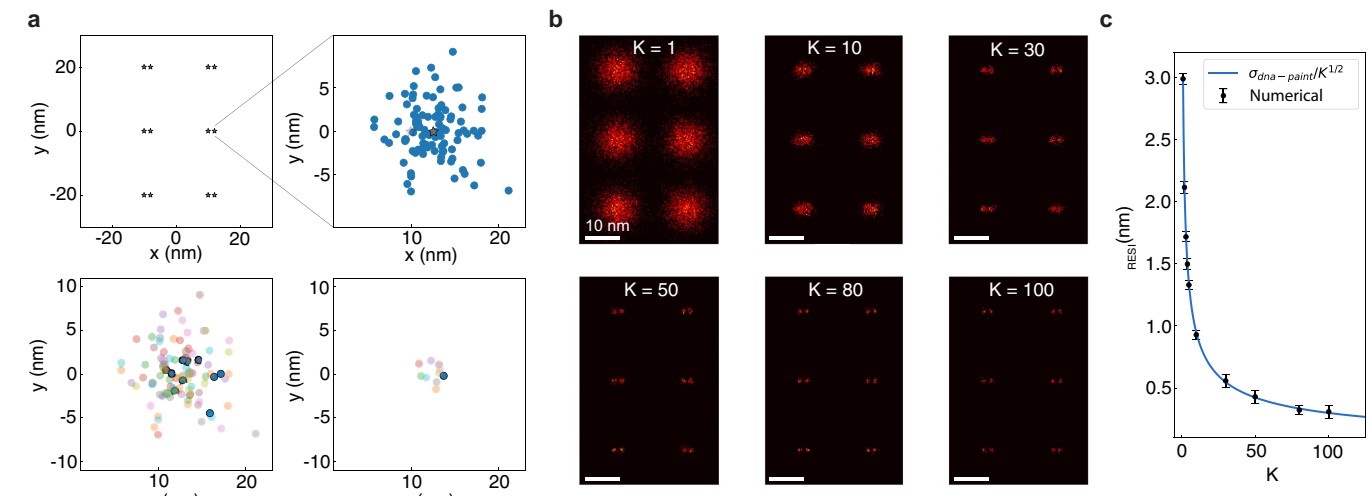

**Extended Data Fig. 1 | RESI resolution estimation. a**, A grid of defined positions of binding sites is generated (top left), SMLM (DNA-PAINT) localizations are simulated as samples from a gaussian distribution (top right). Localizations for only one binding site were plotted for clarity. For each binding site, subsets of K localizations are randomly selected (bottom left) and averaged (bottom right). One exemplary subset and its average is highlighted. **b**, Resulting RESI-localizations are histogrammed to produce images at different resolutions (K values). **c**, RESI-localization precision $\sigma_{RESI}$ vs K. Analytical dependence on $\sqrt{K}$ (blue line) and numerical results (black dots). A total of 1200 SMLM localizations per site are simulated. Error bars represent mean ± 1 s.d.

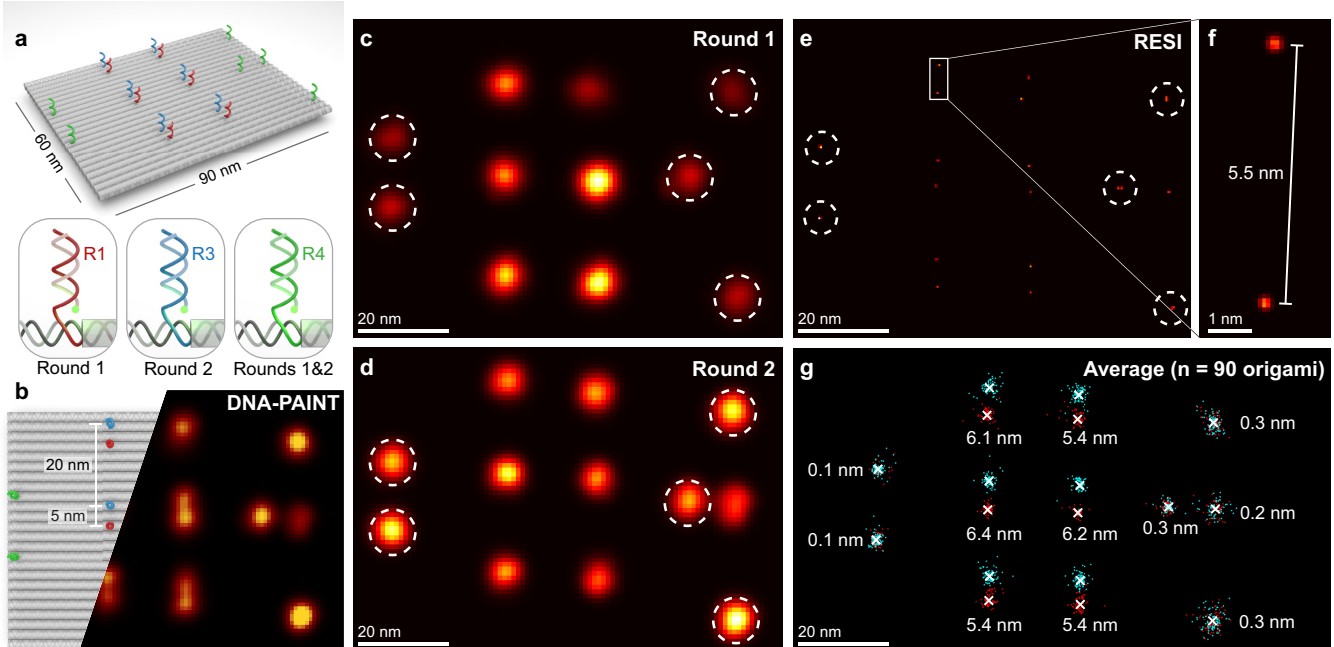

**Extended Data Fig. 2 | RESI in 2D DNA origami. a**, DNA origami design featuring six 5 nm-spaced orthogonal docking strand pairs (red R1, blue R3) and six alignment docking strands (green R4). See Methods for sequence details. **b**, DNA-PAINT acquisition parameters were tuned such that 5 nm were not consistently resolvable. **c**, First imaging round conducted with R1 (target) and R4 imagers (alignment, sites circled). **d**, Second imaging round conducted with R3 (target) and alignment imagers (R4, sites circled). The alignment sites were used for translational and rotational alignment between rounds. **e**, RESI resolves the 5 nm distances. **f**, The distance and orientation between R1 and R3 docking strands are consistent with the design. **g**, An average of 90 DNA origami structures reveals consistent results and excellent alignment performance. The numbers indicate the distance between rounds.

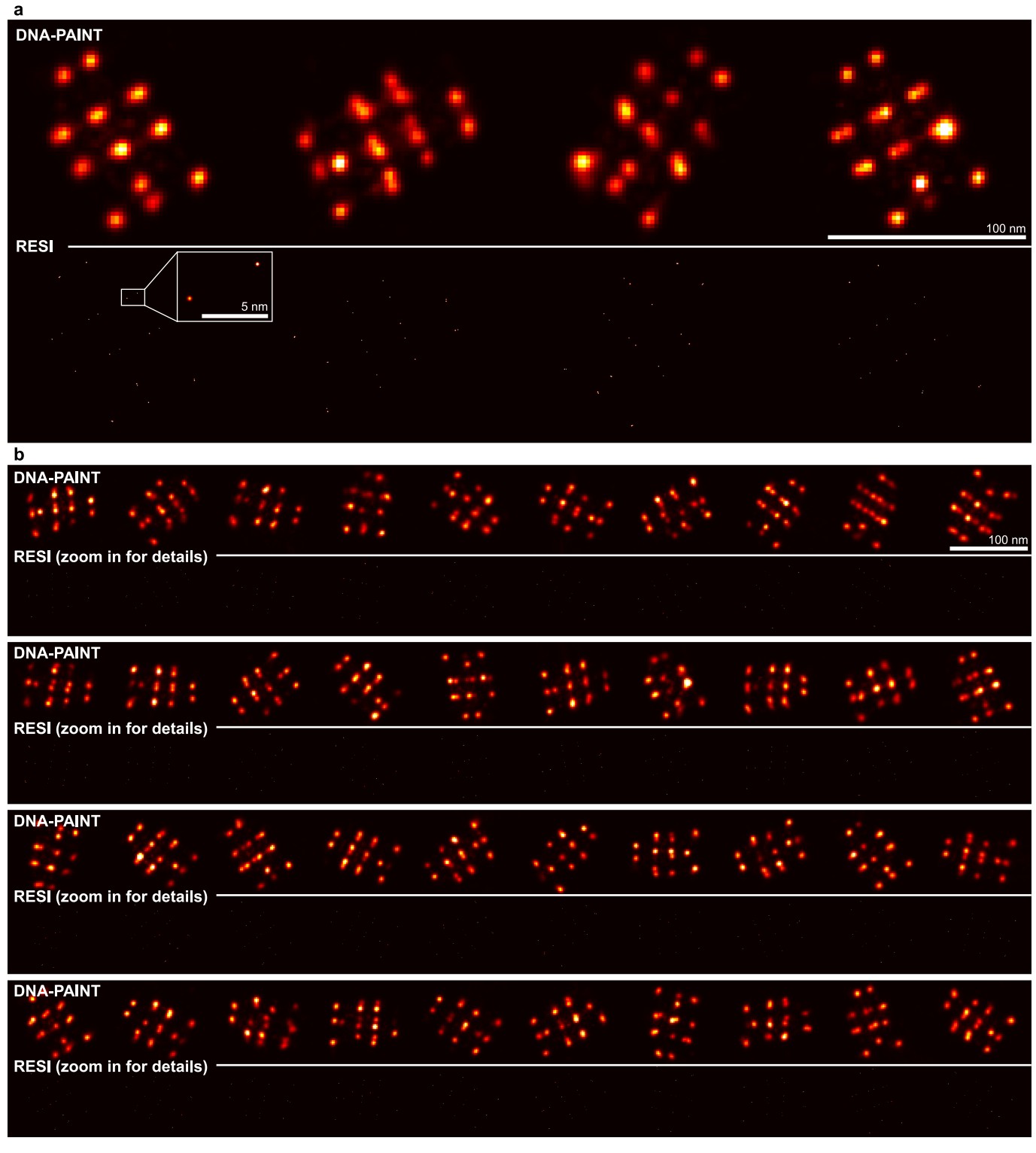

**Extended Data Fig. 3 | 2D DNA origami.** Representative DNA origami from across the field of view of the measurement. **a**, Four DNA origami, shown at DNA-PAINT resolution (upper row) and RESI resolution (lower row). The insert depicts a pair of docking strands spaced at approx. 5 nm. **b**, 40 additional DNA origami, shown at DNA-PAINT resolution (upper rows) and RESI resolution (lower rows).

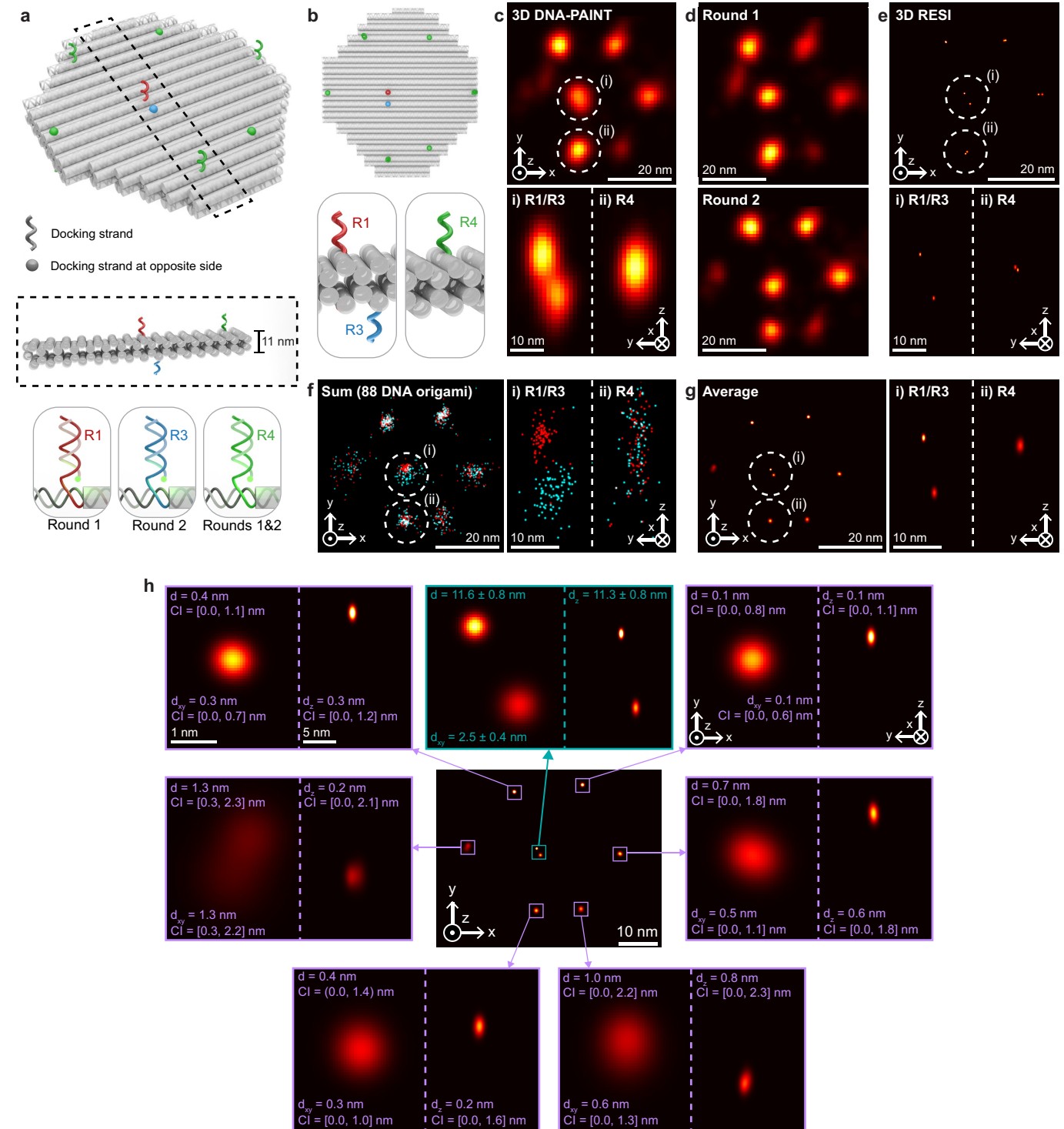

**Extended Data Fig. 4 | RESI in 3D DNA origami. a**, DNA origami design featuring one pair of orthogonal docking strands (red R1, blue R3) as well as six alignment docking strands (green R4). Docking strands extend from both the top and bottom surface of the DNA origami (insert). **b**, The design ensures that all but the R1/R3 docking strand pair are spaced sufficiently to be resolved by DNA-PAINT. **c**, 3D DNA-PAINT imaging resolves R4 alignment sites, barely resolves R1/R3 axially and does not resolve R1/R3 laterally. **d**, Sequential 3D DNA-PAINT imaging with R4 sites used for alignment. **e**, RESI resolves R1/R3

both axially and laterally. **f**, An overlay of 88 DNA origami reveals overall good alignment despite structural heterogeneity. **g**, Average of 88 DNA origamis. **h**, The particle average recovers the structure with an alignment uncertainty of 0.7 nm CI = [0, 1.6] nm, showing a distance between the average R1/R3 positions of 11.6 ± 0.8 nm (xy-distance: 2.5 ± 0.4 nm, z-distance: 11.3 ± 0.8 nm), matching the designed distances[20]. Same scale applies to all magnification panels. CI describes 68% confidence interval.

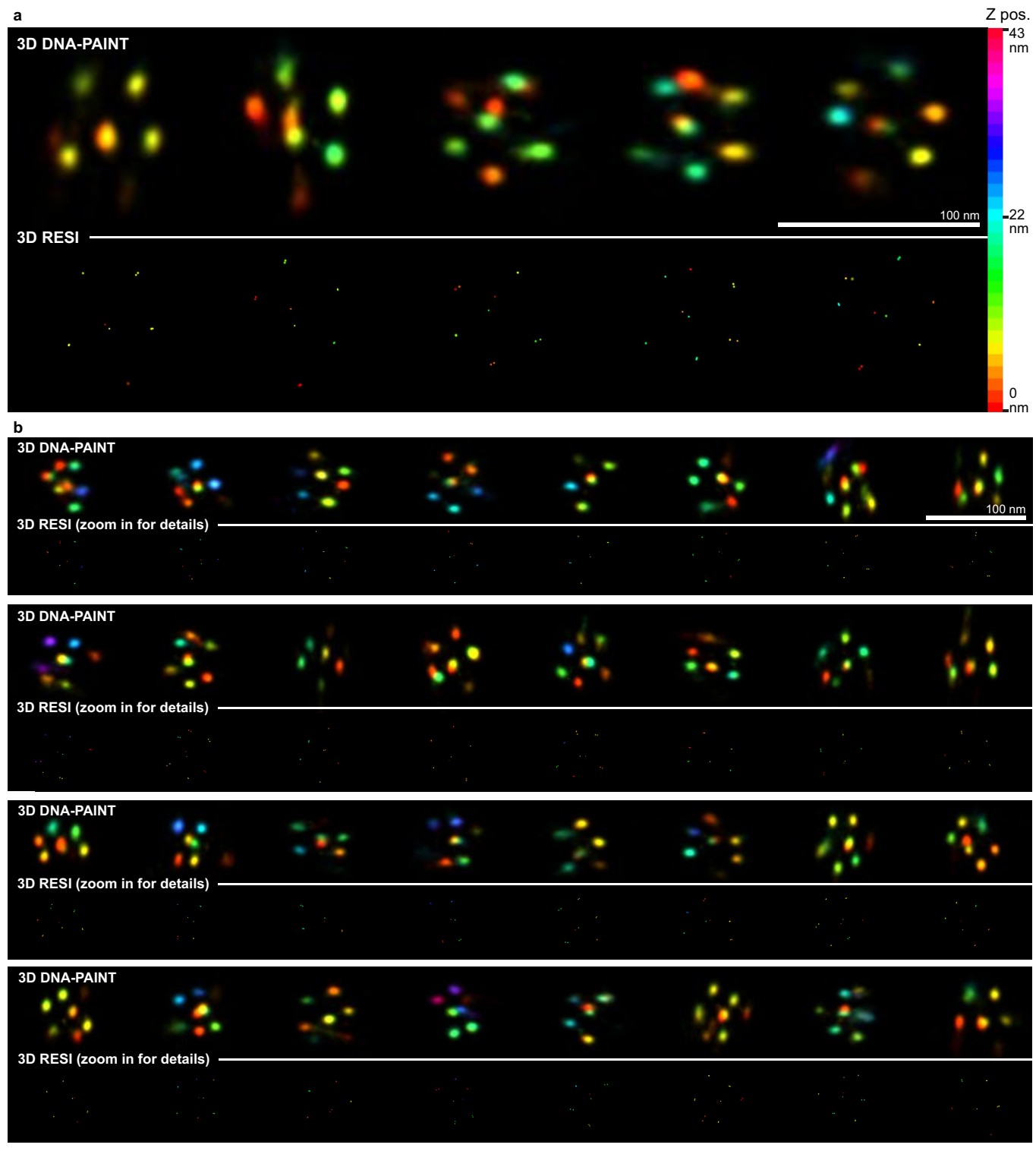

**Extended Data Fig. 5 | 3D DNA origami.** Representative 3D DNA origami from across the field of view of the measurement. **a**, Five DNA origami, shown at DNA-PAINT resolution (upper row) and RESI resolution (lower row). The color scale to the right represents the z position of localizations. The measured z coordinates for each DNA origami have been shifted by a constant such that the lowest localization for a given structure is defined to be at z = 0. This ensures full use of the color range. **b**, 32 additional DNA origami, shown at DNA-PAINT resolution (upper rows) and RESI resolution (lower rows). The z positions are colored according to the color scale in panel **a**.

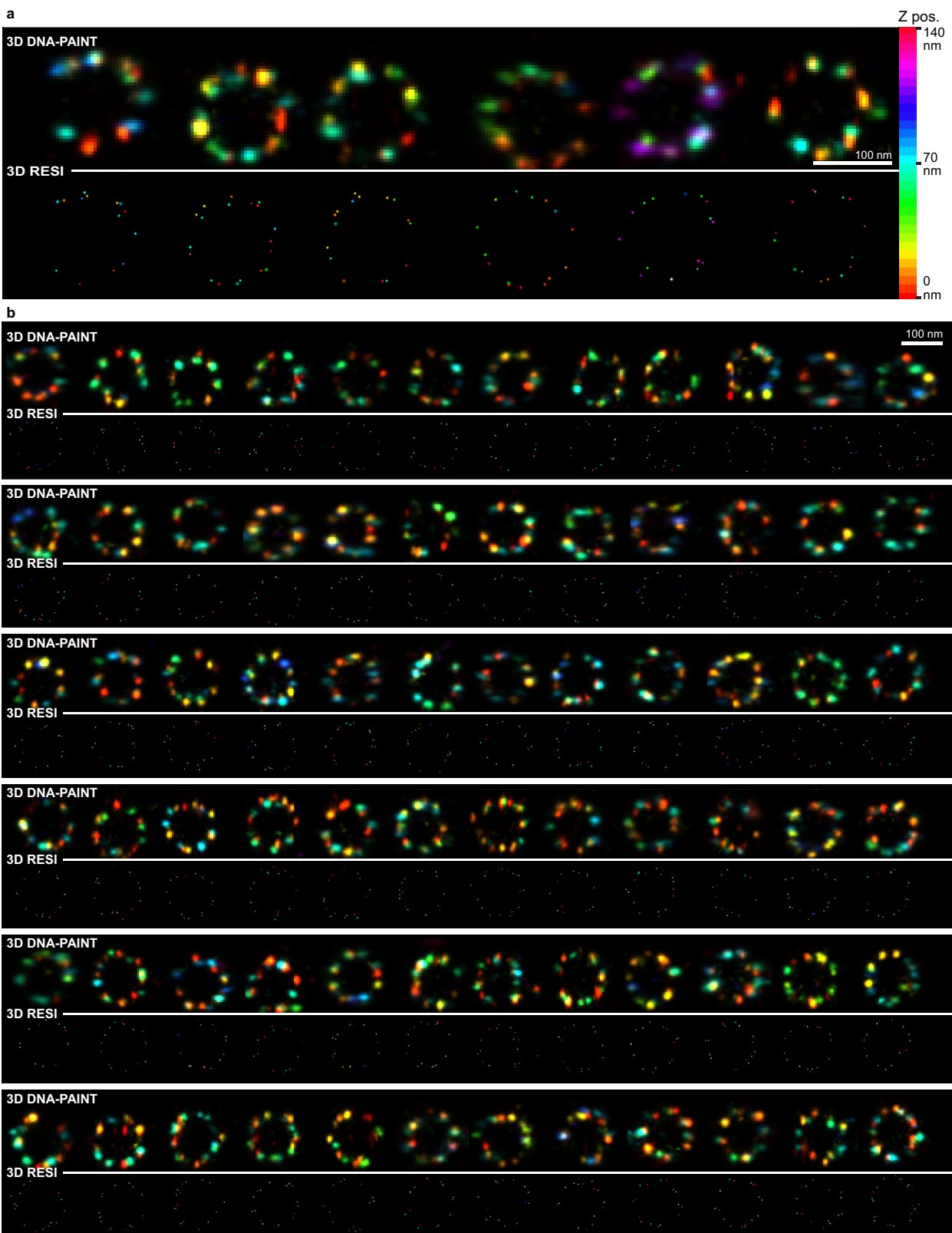

**Extended Data Fig. 6 | U2OS Nup96-mEGFP.** Representative NPCs from across the field of view of the measurement. **a**, Six NPCs, measured using DNA-PAINT (upper row) and RESI (lower row). The color scale to the right represents the z position of localizations. The measured z coordinates for each NPC have been shifted by a constant such that the lowest localization for a given structure is defined to be at z = 0. This ensures full use of the color range. **b**, 72 additional NPCs, measured using DNA-PAINT (upper rows) and RESI (lower rows). The z positions are colored according to the color scale in panel **a**.

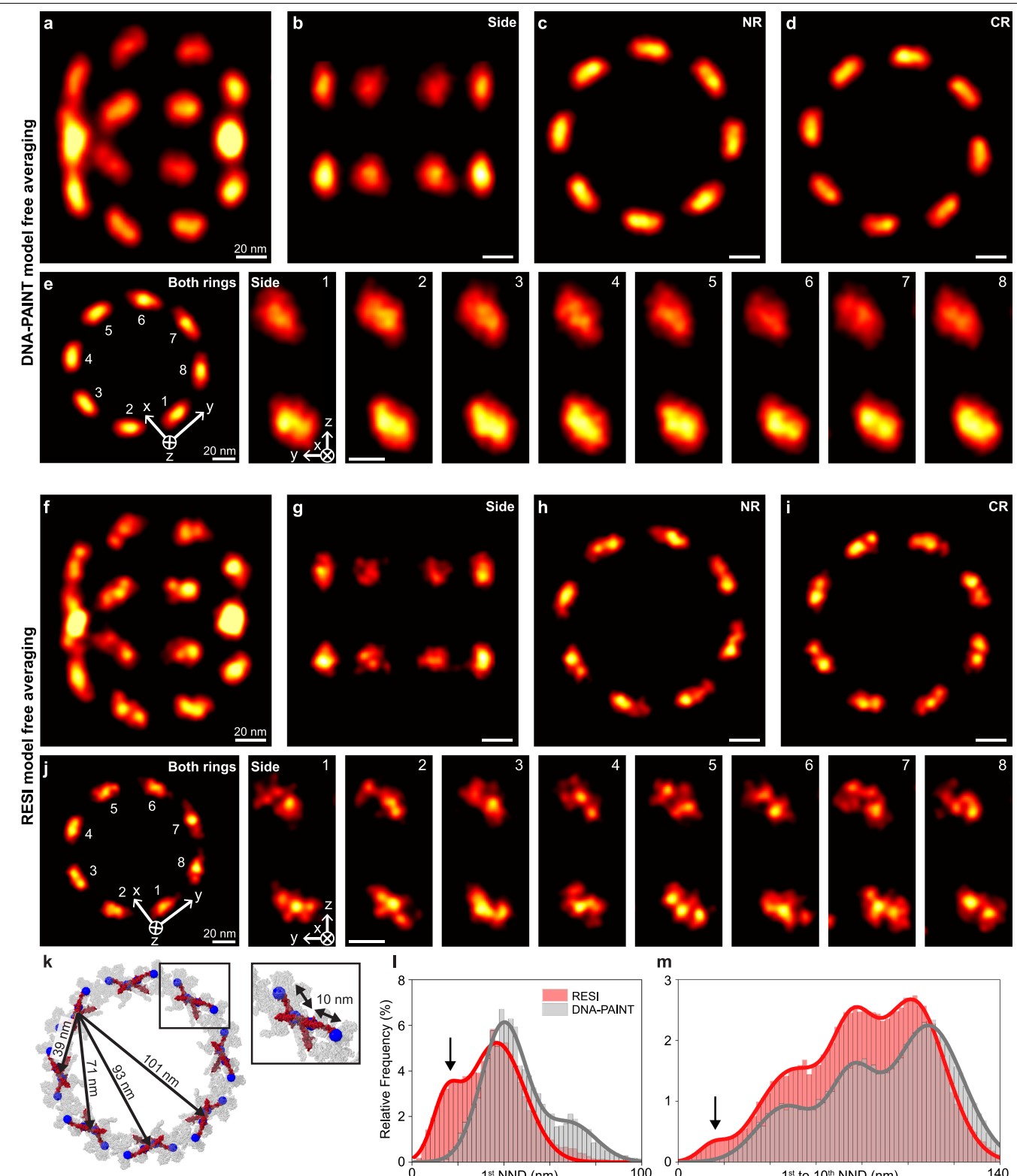

**Extended Data Fig. 7 | Averaging of Nup96 proteins. a**, Model-free averaging for DNA-PAINT measurements of Nup96 (N = 1045 NPCs). An angled isometric view is shown. **b**–**d**, DNA-PAINT resolves nucleoplasmic and cytoplasmic rings and recapitulates their eight-fold symmetry, but fails to resolve individual Nup96 proteins. **e**, Side views of all Nup96 pairs in both rings reveal the angled orientation but do not resolve individual Nup96 proteins. **f**, Model-free averaging for RESI measurements of Nup96 (N = 1190 NPCs). **g**–**i**, RESI recapitulates nucleoplasmic and cytoplasmic rings as well as their eight-fold symmetry and resolves individual adjacent Nup96 proteins in the majority of cases. **j**, Side views of all eight Nup96 pairs in both rings reveal the angled

orientation as well as, in some cases, adjacent individual Nup96 proteins. **k**, The Cryo-EM structure of the nuclear pore complex indicates that a given Nup96 protein will have neighbors spaced at 11 nm, 39 nm, 71 nm, 93 nm and 101 nm. **l**, Performing clustering and nearest neighbor analysis for DNA-PAINT data reveals a peak at approx. 40 nm, corresponding to the distance between two Nup96 pairs, but not below that. RESI, on the other hand, features a first peak at approx. 15 nm, corresponding to the distance between adjacent Nup96 while taking linkage error (label size) into account. **m**, Analysis of first to tenth nearest neighbor distances for RESI and DNA-PAINT recapitulates the distances from (**k**), but only RESI resolves the smallest distance. All scale bars: 20 nm.

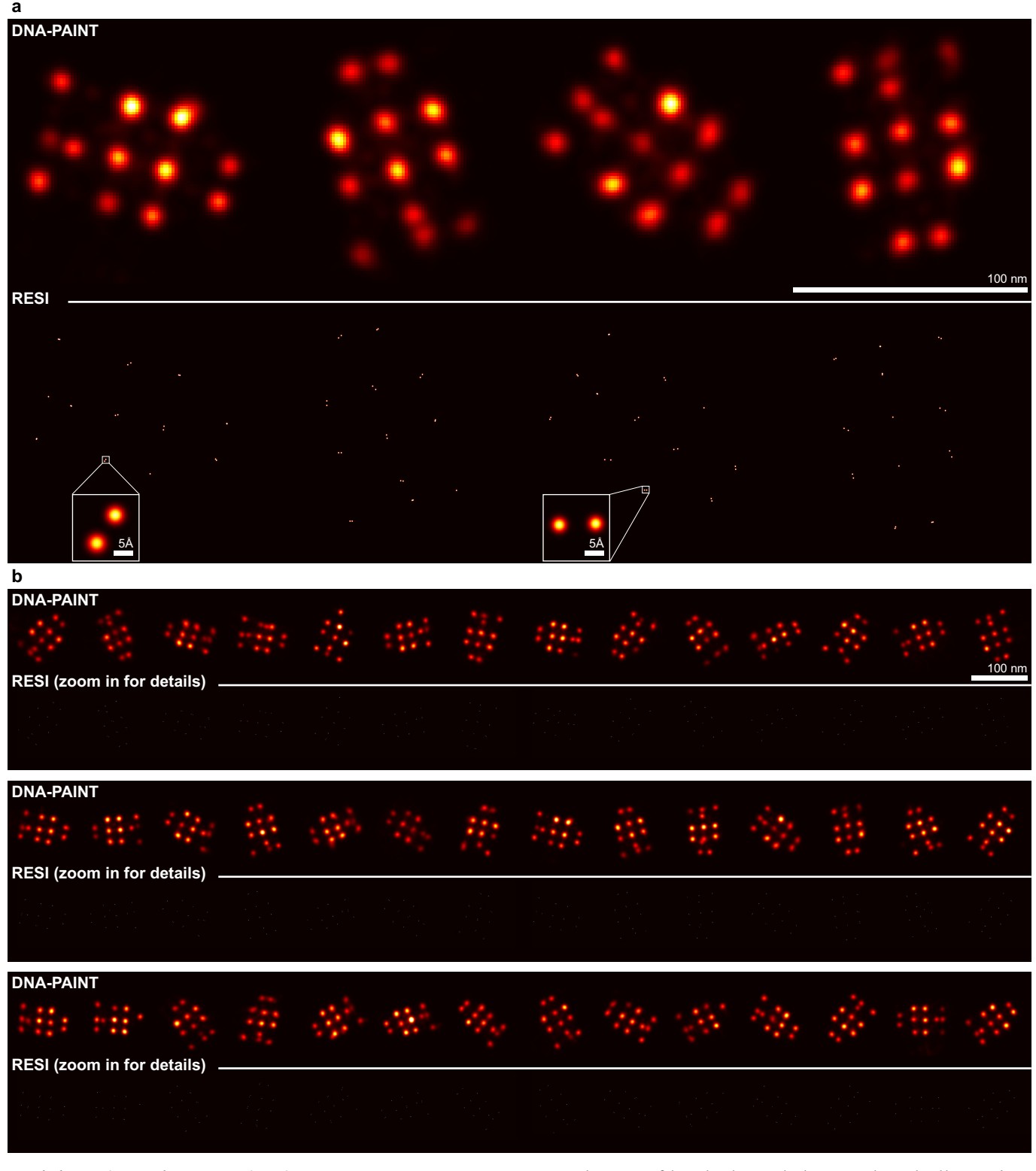

**Extended Data Fig. 8 | Sub-nm DNA origami.** Representative DNA origami from across the field of view of the measurement. **a**, Four DNA origami, shown at DNA-PAINT resolution (upper row) and RESI resolution (lower row). The inserts show pairs of directly adjacent docking strands resolved by RESI. **b**, 42 additional DNA origami, shown at DNA-PAINT resolution (upper rows) and RESI resolution (lower rows).

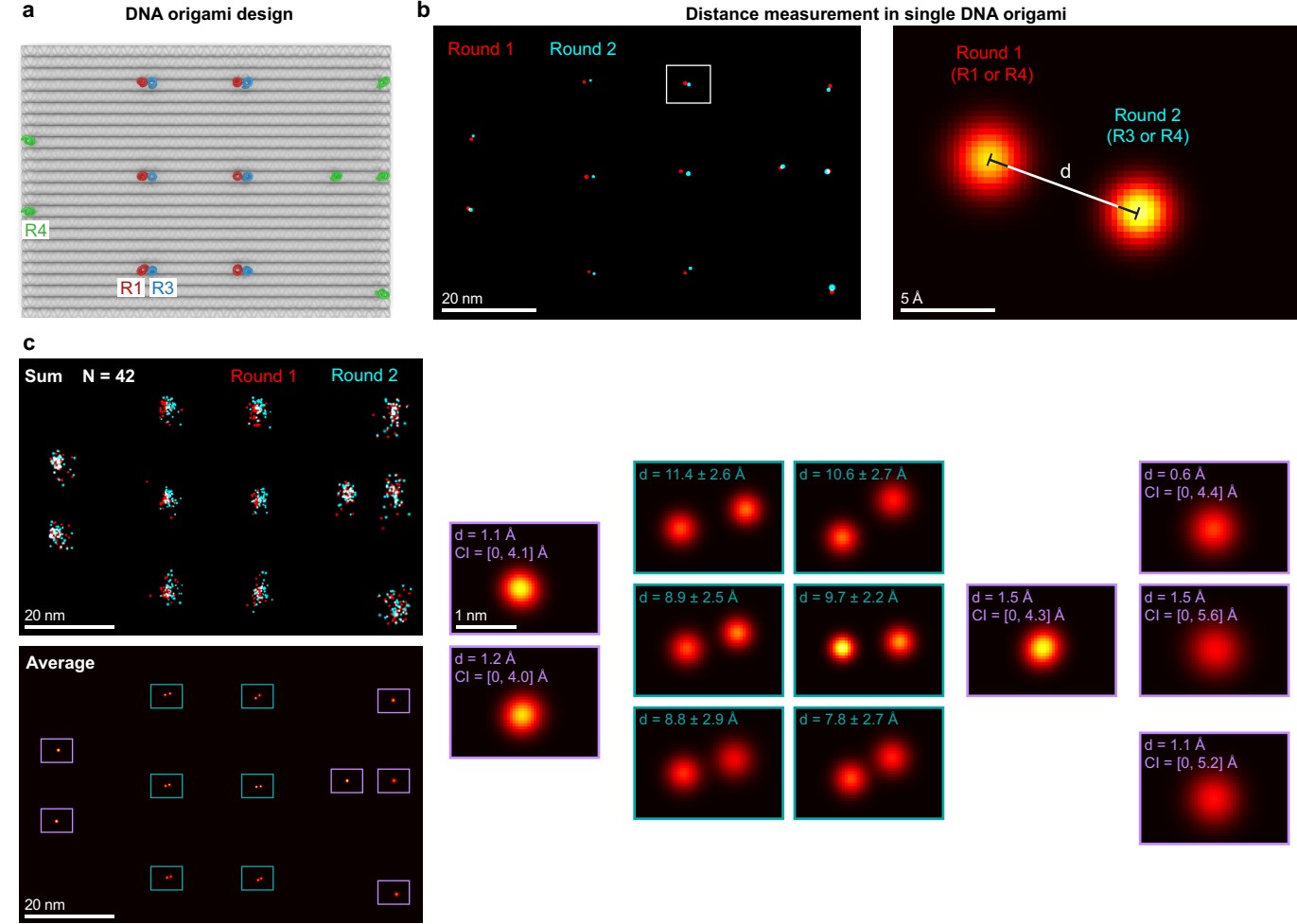

**Extended Data Fig. 9 | Sub-nm RESI measurements. a**, DNA origami featuring six alignment strands (green R4) and six pairs of orthogonal docking strands (red R1, blue R3) spaced one base pair apart. **b**, RESI representation with RESI-localizations from round 1 in red and round 2 in blue illustrates excellent alignment. The distances between RESI-localizations from round 1 and 2 are defined as illustrated. **c**, Overlaying 42 DNA origami and performing a particle average recovers the structure with an alignment uncertainty of 1.2 Å CI = [0, 4.6] Å, showing distances between the average positions of the sites at 9.5 ± 2.6 Å (mean over six distances in the average ± mean over the error-propagated uncertainties of the six distances). Same scale applies to all magnification panels. CI describes 68% confidence interval.

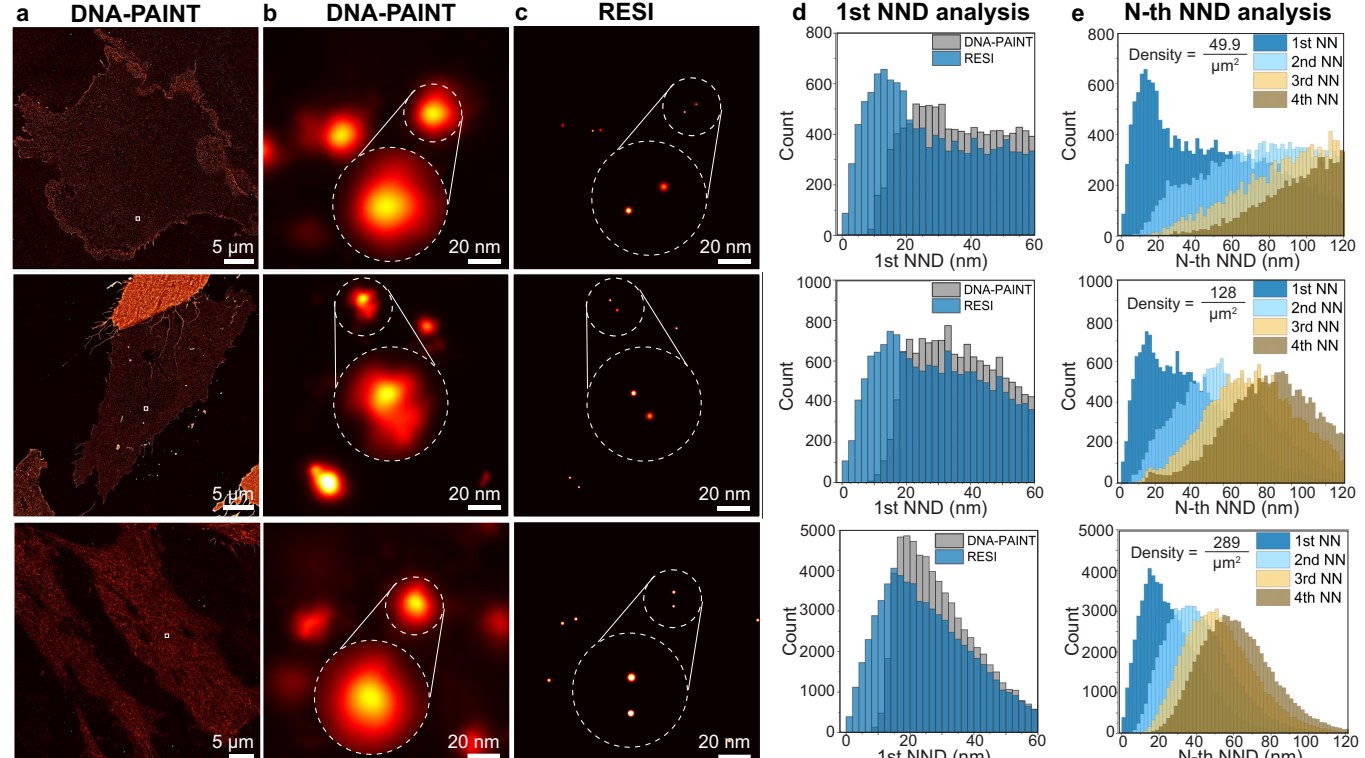

**Extended Data Fig. 10 | RESI resolves CD20 dimers in untreated CHO cells for different expression levels. a**, DNA-PAINT imaging of whole mEGFP-CD20-expressing CHO cells, labeled with anti-GFP-nanobodies, shows homogeneously distributed molecules for three independent experiments. **b**, Zoom-in regions of DNA-PAINT show cases in which dimers could not be resolved. **c**, RESI reveals sub-10-nm spaced receptor pairs, which are unresolvable in the DNA-PAINT cases. **d**, Whole-cell analysis of first nearest neighbor distances (1st NNDs) of CD20 receptors (histograms of the distances are displayed). Only RESI, but not DNA-PAINT, allows the routine detection of sub-10-nm distances between proteins. **e**, RESI-localization precision below 1 nm allows for routine detection of sub-10-nm distances, resulting in an accurate assessment of the first NND.

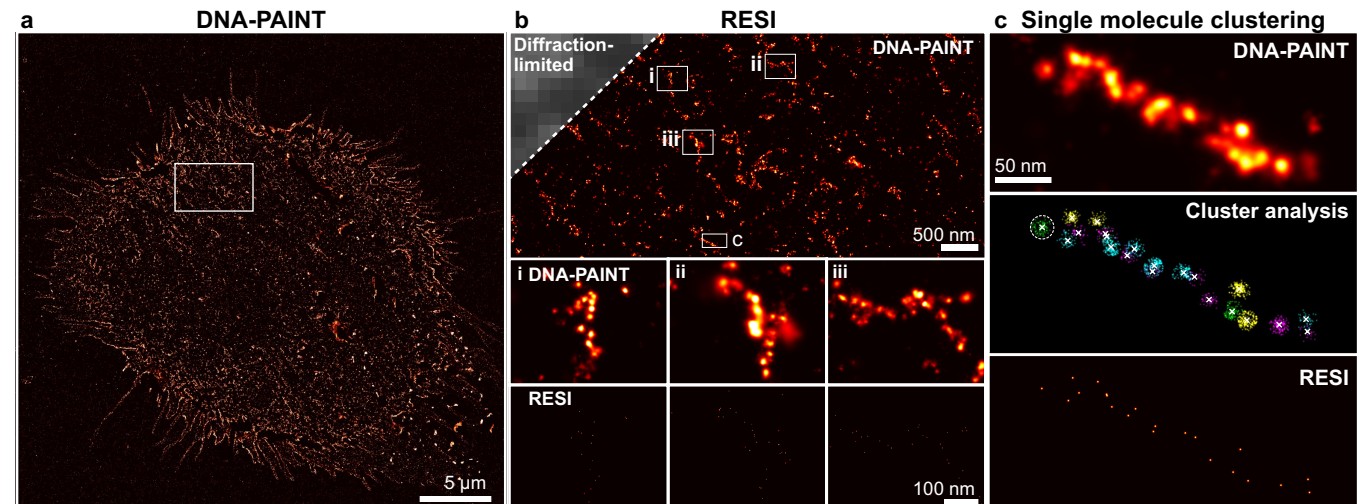

**Extended Data Fig. 11 | RESI resolves the substructure in RTX-induced chain-like arrangements of CD20 receptors with sub-nanometer precision. a**, DNA-PAINT overview image of mEGFP-CD20 expressing CHO cells treated with RTX. **b**, Labeling with DNA-conjugated anti-GFP-nanobodies and imaging with DNA-PAINT reveals higher-order organization after RTX-treatment. RESI (insets i–iii) achieves molecular resolution and thereby resolves the molecular arrangement of mEGFP-CD20. **c**, DNA-PAINT imaging shows clustered CD20 molecules. Performing RESI with sequences R1, R2, R3 and R4 in four separate imaging rounds (color-coded) allows for clustering of localizations originating from a single target. From the clustered localizations, RESI-localizations were calculated, enabling true single-protein resolution.

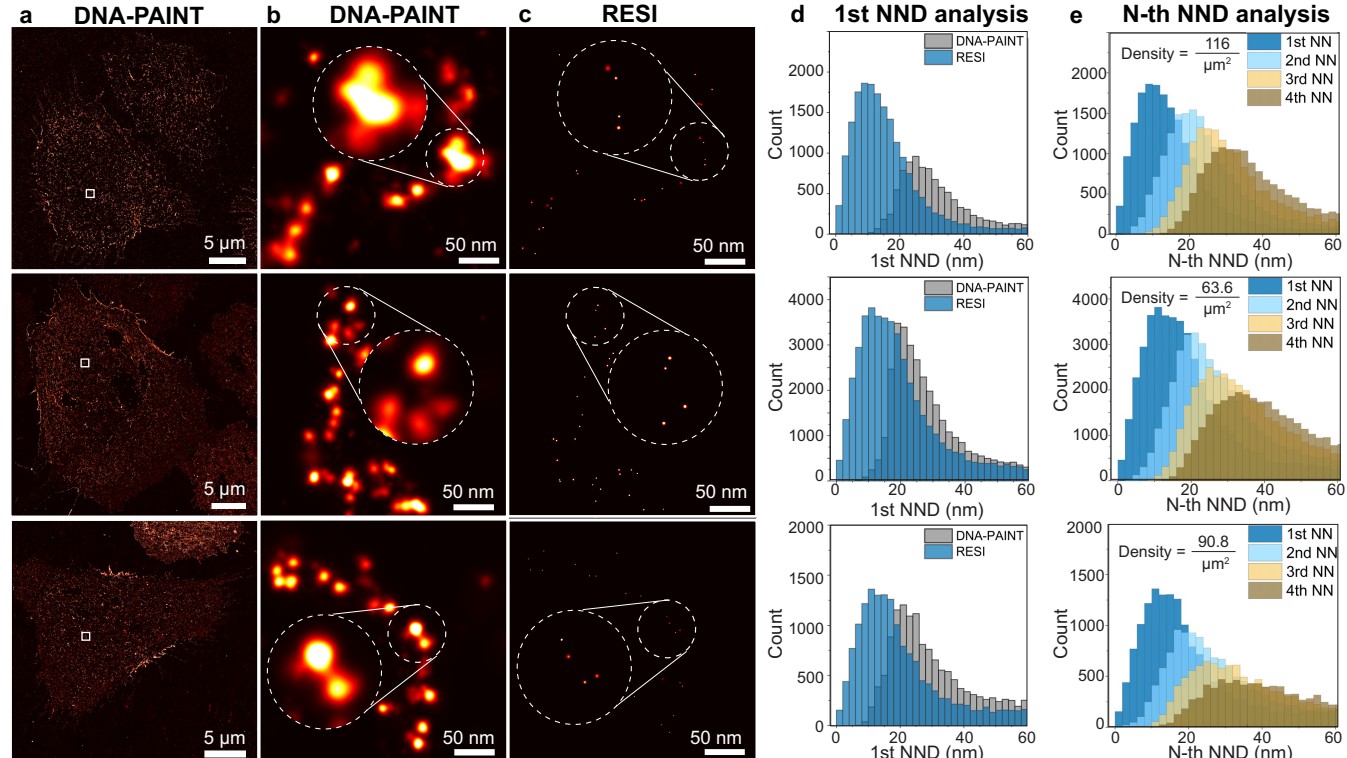

**Extended Data Fig. 12 | RESI reveals higher order arrangement of CD20 dimers in Rituximab-treated CHO cells. a**, DNA-PAINT imaging of whole mEGFP-CD20-expressing CHO cells, labeled with anti-GFP-nanobodies, shows clustered CD20- molecules in Rituximab-treated cells for three independent experiments. **b**, Zoom-in regions of DNA-PAINT show mEGFP- CD20 clustered into chain-like arrangements. **c**, RESI reveals sub-10-nm spaced receptor pairs within the clusters, unresolvable by DNA-PAINT. **d**, Whole-cell analysis of first nearest neighbor distances (1st NNDs) of CD20 receptors bound to Rituximab (histograms of the distances are displayed). Only RESI, but not DNA-PAINT, allows the routine detection of sub-10-nm distances between proteins. **e**, Routine detection of sub-10-nm distances by RESI recapitulates the first NND measured in the untreated case. Notably the NND peaks measured in the three repeats are consistent, independently of the protein density.

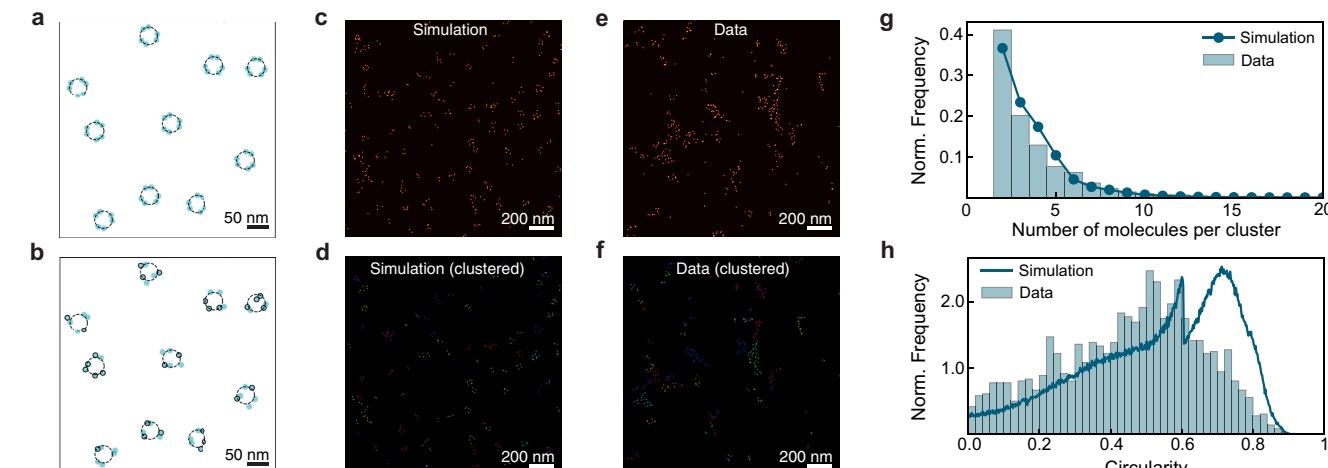

**Extended Data Fig. 13 | Comparison of Rituximab treated CD20 data to simulated CD20 hexamer arrangements. a**, Example of ground truth simulated CD20 hexamers (light blue circles, simulated as triangles of dimers with intra-dimer distances of 13.5 nm as measured experimentally) with random distribution and orientation on a 2D surface at the experimentally determined density. **b**, Label uncertainty and labeling efficiency (black circles indicate labeled molecules) are taken into account in the simulation for a realistic comparison. **c**, Simulated proteins in hexameric arrangements represented as gaussians. **d**, Hexamers after DBSCAN cluster analysis (colors indicate clusters). **e**, RESI image of CD20 data after RTX-treatment. **f**, RESI-localizations of CD20 data after DBSCAN cluster analysis (colors indicate clusters). **g**, Number of molecules per detected cluster for the experimental data and the simulated hexamers. **h**, Circularity metric of experimental data and the simulated hexamers after convex hull analysis of the clusters. We note that the sharp drop at 0.605 stems from the maximum circularity metric for clusters where the convex hull is defined by three molecules. Notably, the absence of a circularity peak at ~0.7 in the experimental data suggests that CD20 molecules are not arranged in isolated ring-like hexameric structures.

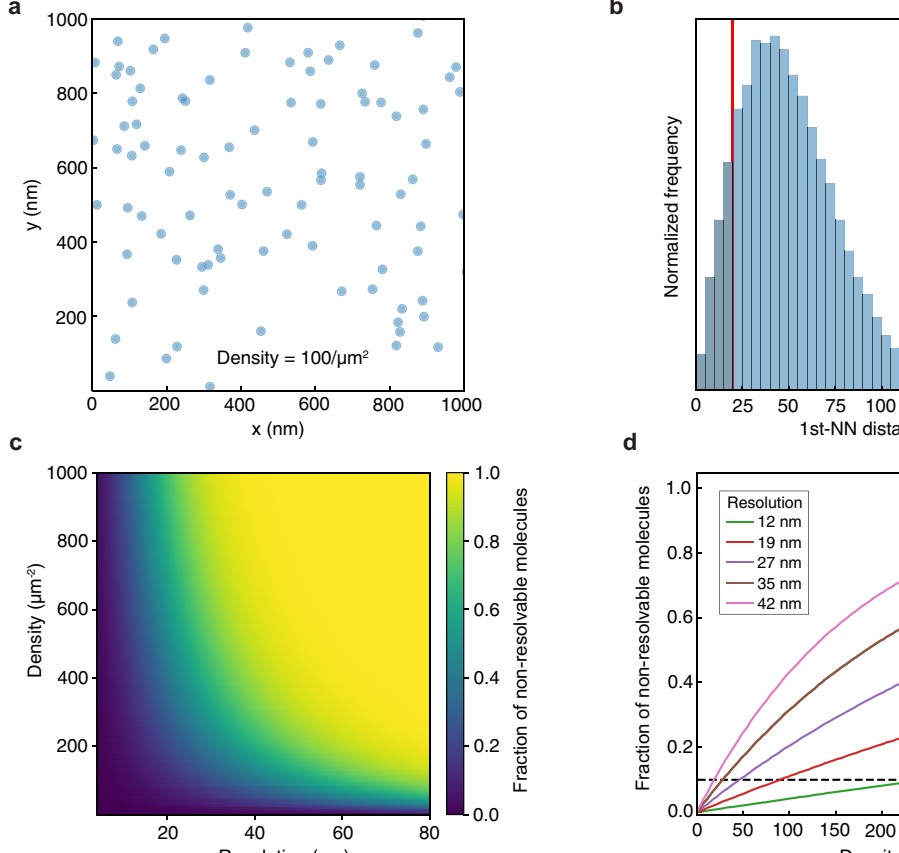

**Extended Data Fig. 14 | Stochastic labeling. a**, Exemplary simulation of proteins with a Complete Spatial Random (CSR) distribution of a given density. **b**, Histogram of Nearest Neighbor Distances (NNDs). The red line indicates the smallest distance (*d*) that can be resolved by DNA-PAINT for a given set of imaging parameters. The fraction of molecules with a NN below this distance threshold (blue, shaded) can be computed for a given density and a given DNA-PAINT resolution. **c**, 2D map of the fraction of non-resolvable molecules as a function of density and resolution. **d**, 1D cuts of c at different resolutions (color-coded) can be used as a user guide to estimate the number of multiplexing rounds needed to perform RESI efficiently given a certain target fraction of non-resolvable distances.

**Extended Data Table 1 | Imaging and RESI parameters**

| Data | Round | Exposure time (ms) | Number of frames | Power (at objective, mW) | Power density (W / cm$^2$) | Imagers | Buffer | Cluster Radius xy (nm) | Cluster Radius z (nm) | Cluster N$_{min}$ | K_mean |
|---|---|---|---|---|---|---|---|---|---|---|---|
| Extended Data Figure 2 | Round 1 | 100 | 40 000 | 52 | 260 | 700 pM 5' R1, 700 pM R4 | B+ (PCA, PCD, Trolox) | 4 | - | 50 | 423 |
| | Round 2 | | | | | 700 pM R3, 700 pM R4 | | | | | 338 |
| Extended Data Figure 4 | Round 1 | 100 | 40 000 | 56 | 280 | 500 pM R1, 250 pM R4 | B+ (PCA, PCD, Trolox) | 7 | 20 | 30 | 114 |
| | Round 2 | | | | | 350 pM R3, 250 pM R4 | | | | | 112 |
| Figure 2, Extended Data Figure 6, Extended Data Figure 7 | DNA PAINT | 150 | 40 000 | 18 | 90 | 15 pM R1, 15 pM R2, 15 pM R3, 15 pM R4 | C+ (PCA, PCD, Trolox) | 11 | 22 | 10 | - |
| | Round 1 | | 10 000 | | | 150 pM R1 | | 11 | 22 | 10 | 23 |
| | Round 2 | | | | | 100 pM R2 | | | | | 50 |
| | Round 3 | | | | | 100 pM R3 | | | | | 31 |
| | Round 4 | | | | | 100 pM R4 | | | | | 43 |
| Figure 3 & Extended Data Figure 9 | Round 1 | 100 | 30 000 | 56 | 280 | 500 pM 5' R1, 800 pM R4 | B+ (PCA, PCD, Trolox) | 4 | - | 50 | 250 |
| | Round 2 | | | | | 300 pM R3, 800 pM R4 | | | | | 258 |
| Figure 4 (Untreated) & Extended Data Figure 10 | DNA PAINT | 100 | 160000 | 30 | 150 | 100 pM R1, 100 pM R2, 50 pM R3, 100 pM R4 | C (PCA, PCD, Trolox) | 9 | - | 15 | - |
| | Round 1 | 100 | 40000 | | | 400 pM R1 | | 9 | - | 15 | 80 |
| | Round 2 | | | | | 400 pM R2 | | | | | 59 |
| | Round 3 | | | | | 200 pM R3 | | | | | 87 |
| | Round 4 | | | | | 400 pM R4 | | | | | 119 |
| Figure 4 (RTX treated), Extended Data Figure 11, Extended Data Figure 12 | Round 1 | 100 | 40000 | 30 | 150 | 300 pM R1 | | 9 | - | 15 | 58 |
| | Round 2 | | | | | 300 pM R2 | | | | | 46 |
| | Round 3 | | 35000 | | | 300 pM R3 | | | | | 94 |
| | Round 4 | | | | | 300 pM R4 | | | | | 121 |

Overview of DNA-PAINT image acquisition parameters alongside clustering and RESI parameters.