## [Peer Review File · Nature]

Manuscript Title: Ångström-resolution fluorescence microscopy

Reviewer Comments & Author Rebuttals

Reviewer Reports on the Initial Version:

Referee #1 (Remarks to the Author):

The authors resolve targets on DNA-origami and in cells with single-molecule fluorescence, closer than has been reported previously, with accuracy and resolution apparently limited by the labelling technique. They achieve this by labelling nearby targets with orthogonal labels in different rounds of imaging and finding the centres of the resulting localisation clusters in each round. DNA-PAINT in particular allows the spatial centres of the clusters to be estimated very precisely as result of the replenishment of the imaging strands. They resolve labelled DNA origami targets separated by less than 1 nm and resolve labelled Nup96 copies within the NPC subunit separated by ~ 12 (xy) by 5 (z) nm in an averaged SMLM NPC structure, which are new achievements and appear to be a clear improvement over the state-of-the-art in this area. They are also able to investigate CD20 dimers with labels separated by less than 10 nm. These results are of immediate interest to many people in the SMLM field, and others concerned with supramolecular protein complex structure are also likely to be interested.

Some limitations should be highlighted in general, and clarity could be improved in places. The first issue that I believe should be highlighted early on and more often is that the improved resolution is only obtained when targets closer than the DNA-PAINT cluster size are labelled orthogonally. When targets this close are labelled with the same type of docking strand (in the case of DNA-PAINT), RESI will not distinguish them. In the DNA origami, this distinguishability is programmed into the sample, but in cellular imaging, there are probabilities for nearby molecules either being randomly orthogonally labelled or not. The authors do state this and give derivations in supplementary, but in my opinion it would benefit readers for this to be more obvious from the start and in the discussion. As the authors point out, their use of four orthogonal docking strands does still allow for imaging the majority of targets in this way in their demonstrations. A consideration of the probabilities of being able to label and resolve the structure of trimers, tetramers etc. on particular length scales with labelling probabilities in mind might be interesting.

Secondly, considerations of resolution and structural accuracy could be improved. For instance, "label-size-limited" resolution is claimed in the Nup96 data, but then localization inaccuracy from linkage error must become a more important confounding factor in structural imaging and interpretation of structural data. Discussing this is of benefit to readers. In the CD20 case, "RESI at one nanometer resolution" is referred to, but one nanometer resolution (distance between resolved targets) is not demonstrated here and may not be possible in principle because of the size of the labelling nanobody. Referring to estimated localization precision would be preferable.

Novelty:

Previous work has used the fact that orthogonal labels (e.g. two colour channels in dSTORM) can allow unconfounded distributions, and distances between them, to be obtained for targets closer than the localization precision (e.g. Letierrier, C. et al. Cell Rep 13, 2781 (2015)). This has also included simulations of instances of the same protein with different labels at zero distance from one another (e.g. Levet, F. et al. Nat Commun 10, 2379 (2019)). However, the representation of a cluster of localizations by its weighted mean and weighted standard error on the mean (SEM) may be novel for spatial analysis in SMLM (although such a representation of a data distribution is common in general), and this allows SEM to be used as a new precision metric when the cluster is not confounded by nearby targets in the same channel.

Particularly in the case of DNA-PAINT, when many localizations can be obtained for the same single target, using the SEM results in a useful improvement in effective precision for single, isolated targets, which also allows higher-precision spatial analysis of nearby target distributions, when nearby targets can be orthogonally labelled. Also, a label exchange method like DNA-PAINT also allows this analysis method to benefit in principle from higher localization accuracy for nearby targets than an acquisition method where all targets are permanently labelled and stochastically switched, as the authors point out.

However, when nearby targets are randomly labelled in the same channel, the precision of the SEM estimate is not retained. For targets in the same imaging channel, the resolution remains limited by the photonic signal and noise as is usually considered, moving back from RESI resolution to conventional SMLM (e.g. dSTORM) consideration. For their test cases, the authors calculate how what fraction of proteins will be unresolvable using RESI, because of their proximity and random labelling. For unknown samples where the stoichiometry and proximity of targets is not yet known, this analysis may result by default in an artifactual collapsing of multiple targets onto one super-resolved target. It would be interesting to hear a discussion of this.

Data and methodology:

DNA-PAINT is ideal as an acquisition tool for data on which to use this processing method which takes advantage of the availability of many localizations per cluster per imaging channel and multiple channels.

In general the data is visualized well, but in Suppl. Fig. 2, 3, 6, and to a lesser extent 5, when the reader is instructed to zoom in for details, the figure still does not seem very informative after zooming in. In my experience, it was still hard to see RESI localizations and not easy to compare with the adjacent DNA-PAINT image.

I did not find information on how z-information was obtained for 3D localizations, which is important to include.

I did not see 3D information in Suppl. Fig. 3, which seems important to include in this figure.

The alignment error, although small, is visible and stands out in Extended Data Fig. 2e. This may be worth referring to or explaining.

Fig. 2d: It may helpful to explain what the background structure is, i.e. not part of the acquired data.

Statistics and interpretation:

Fig. 2i and/or main text (could use other parts of Fig. 2 also): It would be helpful to compare with a structure with linkage errors taken into account, somehow, given that the RESI precision that the authors use is smaller than this linkage error.

“This spatial arrangement [Nup96 pairs] is consistently resolved for all Nup96 protein pairs (Extended Data Fig. 3)”: While the resolution obtained appears impressive, Extended Data Fig. 3j does not obviously illustrate this statement in its current form, as the densities are often indistinct or have more than two maxima per protein pair.

Fig. 3 and text: Information is missing or unclear as to how the distance between the labelled targets was measured from the RESI clusters or their visualizations.

What are the distance uncertainties? s.d.?

7 Å is used as the comparison distance for the results. However, Fig. 3d shows an inter-base pair distance of 7 Å, measured along the orientation of one DNA strand. Defined like this, the average of the inter-base pairs distances in xy would be less than 7 Å. Therefore it is not currently clear whether this is the right quantitative comparison, and the authors should think this through and present the distance and quantitative comparisons again appropriately.

Given that the manuscript currently does use 7 Å for comparison:

8.5 ± 1.7 Å may be considered in agreement with 7 Å, if this is specified in the manuscript to mean within one s.d. The 8.5 ± 1.7 Å result is not compared with 7 Å in the main text, only the Fig. 3 legend.

The 9.5 ± 1.2 Å result should be directly compared with the 8.5 ± 1.7 Å result. The presence of this second experiment should also be explained. I would not necessarily say that 9.5 ± 1.2 Å is in good agreement with 7 Å, because 7 Å is > 2 x s.d. away from 9.5 Å. What the authors mean here may need revising or defining.

In Extended Data Fig. 4c, a distance from R1-R3 is measured at 1.2 nm, and in 4d, the average distances appear to be ~1 nm or greater. How does this fit with the stated 9.5 Å mean? e.g., if this is to do with alignment correction, more explanation and illustration of this is needed.

In Fig. 3, there is “a distance of 8.5 ± 1.7 Å”, with “a precision of ~1.2 Å for the single base-pair backbone distance”, and “an average localization precision of 1.3 Å for the experimental data”. What is meant here needs clarifying. 1.2.

The stated improvement over the state-of-the-art by “approx. a factor of six” seems overly precise,

without more detailed justification than referencing other work.

In Suppl. Fig. 7b, it would be good to justify the claim of “RESI-localization precision below 1 nm” in this experiment.

Monomer/dimer simulations:

- The authors used 50% labelling efficiency while the study they referenced found closer to 60% “effective labelling efficiency”, with true labelling efficiency being greater than this, taking into account undetected labelled targets. It is not clear why the current authors chose a different value.
- The authors carried out some simulations of monomer/dimer mixtures (Suppl. Note 4). However, the basis of their choice of the proportions reported as their best values is not clear. What is the figure of merit? How big was the range of proportions that gave qualitatively similar results, if a quantitative uncertainty could not be obtained? Can a range of proportions be plotted?
- What density of the targets was used for the dCSR simulation (Fig. 4f)? Was this the experimental density? That may help with the comparison, or an explanation of the choice of density.
- A comparison of DNA-PAINT and RESI for Rituximab-treated cells similar to Fig. 4e may help to complete the picture.

A reference or justification of the use of l_p^2 (as opposed to l_p) in the weighted means and SEMs would be beneficial (Suppl. Note 2).

A more detailed explanation of the use of the covariance matrix, including what the covariance(s) is (are) that it describes, would be helpful in Suppl. Note 1a.

Suppl. Note 3 (Case 2): The authors note that a script is provided to output graphs of resolution vs density, but according to the nature of the technique and the previous explanation in the Note, this should also include the fraction of proteins that will not be resolved.

Conclusions/Discussion:

The new scaling law has been defined and used for new, higher-precision spatial analysis from SMLM when repeated localizations can be correctly identified as associated with the same target site, and when nearby target sites are orthogonally labelled. However, the discussion does not currently mention that in a new sample, there may be more than one nearby target site labelled non-orthogonally, which would by default give misinformation for a fraction of the sample structure. The technique is still useful, and this is more or less of a difficulty depending on the structure, and there may be intelligent ways of taking it into account, but I believe it should be included in the discussion.

Angstrom spatial resolution does indeed appear to have been achieved by imaging adjacent DNA base pairs, even if the distance is over-estimated (see Statistics and interpretation).

It is not clear what is meant by an improvement in robustness and precision as a result of averaging out uncertainties such as those due to mechanical instability. This would need to be more explicitly compared with SMLM in general, including DNA-PAINT without RESI.

Even though RESI SMLM requires many localizations per target site, it is still a widefield technique, which does give it a speed advantage over MINFLUX and MINSTED as mentioned, which must currently sequentially attempt localization of only one molecule in the FOV (or several, when these techniques are parallelized). Techniques like SIMFLUX and SIMPLE may be worth mentioning, which give a smaller improvement in resolution over widefield SMLM than RESI, but at a higher speed.

“Quantifying receptor organization down to single protein level for any receptor density”: It is perhaps not clear that this applies for all densities, for all receptors. Receptors with higher stoichiometry, for instance, may need more consideration with regards to challenge of the probability of orthogonal labelling and dealing with data from non-orthogonally labelled complexes. It would be interesting to consider what the local density limit is for RESI to be effective.

“the method of choice for studying three-dimensional protein structure and conformational changes”: This probably needs some elaboration or refinement, since it is not clearly better than cryo-EM or other techniques, but could be complementary to them. The following statement is better, about bridging the gap.

References:

References to recent advances involving expansion microscopy may also be beneficial for comparison. Possibly also SIMFLUX, SIMPLE, etc. as mentioned. A reference to and comparison with RASTMIN, along with MINFLUX and MINSTED, would also benefit the general reader.

Clarity:

Main text line 1: This definition of localization precision is no longer universal for SMLM, as this now includes MINFLUX etc. and SIMFLUX etc.. Referring to “widefield Single-Molecule Localization Microscopy” instead would be correct.

Bottom page 4: Suppl. Fig. 4 is referenced, but this does not describe the situation for Nup96 labelling and imaging. Suppl. Fig. 4 is instead relevant to the CD20 experiment.

Referee #2 (Remarks to the Author):

The manuscript by Reinhardt et al. presents an exciting fluorescence microscopy method to determine distances between biomolecules at unprecedented resolution within the angstrom range. The method, called RESI, can be applied to detect single molecule arrangements within oligomeric proteins, as well as macromolecular complexes in a cellular context. This method will help to fill an important gap between in vitro single-particle cryo-EM imaging and in cellulo cryo-ET, as well as current super-resolution fluorescence microscopy methods. As such, I am confident that this work will be of great interest to the structural and cellular biology communities.

The text is concise and clear, and the figures nicely done to support the scientific claims. Although the results obtained applying RESI to DNA origamis and, in the cellular context to the nuclear pore

complex are impressive and conclusive, those regarding the human receptor CD20 appear somehow weaker:

1-The introductory paragraph in the CD20 section doesn't describe clearly previous single-particle cryo-EM work on CD20 in complexes with fragments from therapeutic antibodies. Rouge et al. have shown that CD20 forms homodimers in complex with two RTX monovalent Fab molecules. In such complex, each Fab molecule binds a large area on the CD20 extracellular surface that extends over the 2 protomers. Moreover, the two Fab molecules bound to dimeric CD20 form extensive homotypic interactions, casting some doubts on whether the CD20 homodimer is induced or facilitated by binding of RTX-Fabs. This is highly unlikely because the structural complementarity of the CD20 protomer-protomer interface is very good. Moreover, independent cryo-EM and biophysical studies by Kumar et al. showed that in complex with divalent F(ab')₂ fragments of Ofatumumab, another type I antibody, the individual Fab molecules bind to only one CD20 protomer, and lack homotypic interactions, yet CD20 displays an identical dimeric structure, demonstrating that the homodimer is indeed the native oligomeric arrangement of CD20, and it is not induced by RTX Fab binding. Finally, SEC-MALS analysis of purified CD20 in different detergent solutions showed that the CD20 elutes as dimers (Kumar et al. Supp Fig. 1). Therefore, the sentence in the current manuscript stating "Furthermore, both studies cannot address CD20 organization in the absence of RTX Fabs" is misleading, please edit this part.

2-It has further been shown using cryo-EM and divalent RTX F(ab)₂' that due to steric constraints the two monovalent Fab molecules that bind a CD20 dimer come from different F(ab)₂' molecules (Kumar et al.). As a natural consequence of this, type I IgG molecules concatenate CD20 dimers upon binding, as demonstrated in vitro. Therefore, the result presented in the manuscript suggesting that RTX concatenate CD20 dimers in cells is more confirmatory than revealing. What in my view is more novel, and it is a direct consequence of working on the more native context of the cell is fact that RTX forms linear rather than circular RTX-CD20 higher-order concatenated assemblies, as observed in vitro and likely due to free tumbling of the molecules in detergent solutions (Rouge et al., and Kumar et al.). But such discussion is missing in the text. Therefore, I would suggest to tone-down a bit the revealing aspect of the discovery, and highlight the fact that as opposed to detergent solutions, in cells the predominant assemblies are linear, as it appears in the images in Supp Fig 8 of the manuscript. Also, do the authors detect any indication of circular arrangements? And is the number of CD20 molecules in the linear arrangements rather constant or random?

3- The above points are mechanistically important, because type I mAbs (RTX or OFA), as opposed to type II, bound to CD20 are expected to form penta- and hexameric circular arrangements that potentially activate the complement system by binding to its C1q component, which is a hexa-headed protein. RESI seems a method ideally suited to explore this kind of questions, and it would be really interesting to see the RESI analysis of cells opsonized with type I (RTX) vs type II (OBZ) mAbs, and the effect of C1q on the two type of cells. I leave to the authors the decision on whether such experiments belong to the current manuscript, but in the absence of more data like those, I would strongly suggest to lower the "revealing" tone of the results regarding CD20 concatenation.

Referee #3 (Remarks to the Author):

The manuscript by Reinhardt et al. presents an optical super-resolution imaging modality where

targeted sites are located with precisions in the order of one to ten Angstroms. This is successfully demonstrated in samples of DNA-origami and antiGFP-nanobody-labelled nucleoporin proteins and B-cell receptors.

The crucial elements to achieve such performance are sample (1) sparsity and (2) repeated localizations. Sparsity is a long-harvested feature of DNA-PAINT: fluorophores are transiently at their targets, thus bypassing unwanted photophysical interactions between dyes belonging to neighboring targets. Yet, when targets are closer than the localization precision, it's not possible to distinguish them nor to combine several localizations to further improve the resulting precision by averaging. To solve this problem, Reinhardt et al. induced further sparsity by randomly labeling with barcodes and imaging sequentially.

One could argue for hours whether the concept presented here is novel or not. The authors rightfully acknowledge that their work is a "reapplication" of the sparse localization trick that broke the diffraction limit. This time it is used to improve limitations brought by photon emission rate/binding time/background. It must be stated that these limitations are not fundamental: a hypothetical brighter dye or a more efficiency localization scheme (the type of MINFLUX/MINSTED) would yield higher precisions and allow to combine localization events. However, those performance limitations are very real and the authors present a powerful and accessible approach to overcome them.

The authors have reexamined fundamental aspects of super resolution imaging and produced an explicit and direct procedure with outstanding performance. One that has been all along in front of our eyes, since the technology and the reagents were available for almost a decade.

For the reasons stated above I absolutely recommend the manuscript for publication, provided that the points below are addressed.

Major points

1. Beyond dimers

The authors present an analysis of the stochastic labelling in the case of dimers and of randomly distributed targets. As the authors claim the applicability of RESI to bridge light microscopy and structural biology, the concrete request for the authors is to include an analysis of the scalability of the method beyond dimers, as the required number of strands will increase tremendously.

A back of the envelope calculation yields $P(\text{diff. seq.}) = \frac{\#\text{diff.seq.}}{\#\text{all seq.}} = \frac{n^h}{(n! / (n-h)!)}$, with n the number of distinct strands and h the order of the multimer. Already for analyzing trimer with $n=4$ strands, only 37% of completely labeled trimers would have distinct strands. With a fifth strand this goes to 48%. Tetramers and pentamers are below 20% for 4 and 5 strands. The limitations in this regard must be explicitly pointed out. Likewise if there are ways to bypass them.

For example, the authors have used the fastest docking strands available with repeated multiple binding to increasing the k_{on} , this might put a limit to how many strands can be really used in the future, as these are minimalist strands with repetitions and not many orthogonal sets can be assembled.

2. Beyond one color

The authors present superb performance for single-color imaging, but there is absolutely no mention of imaging distinct species. It could be argued that this is beyond the scope of the paper, but again, as the authors present the method as a bridge with structural biology, it is a must to distinguish e.g. distinct subunits of a complex. It could also be argued that this might be trivial, as distinct subunits could be targeted with distinct strands directly. The concrete request is, again, to include an analysis of the scalability of RESI for multi color scenarios.

3. Timings

The authors explicitly mention the long acquisition times of around 100 min for full field of view images of RESI. This counts for figure 2 and 3. They must also explicitly mention the experiments on fig. 4 take around 4.4 hour for each conditions, adding up to ~8-9 hours.

The authors should include (e.g. in supplementary figure 4) an approximation of the acquisition time for the sake of experiment design. For example, combining the expected k_{on} of the strands, the density of the expected targets and the desired M repetitions that achieve a given resolution.

4. Illumination scheme

It is not reported in these experiments if the illumination is TIRF, HiLo or what kind. I presume it is TIRF. The authors should report the illumination scheme and whether it leads to limitations.

Minor points

1. I encourage the authors not to use ill-defined terms such as 'ultra-resolution'. This work is still super resolution microscopy, with more resolution than usual. The field suffers from the use of ill-defined terms and metrics.

2. The authors highlight their scaling with $1/\sqrt{M}$. I encourage them to use log-log scale in figures 1f and 3e to further highlight it, and also make it clear when/if the curves saturate.

3. The 5x and 7x repetitions of the docking strands seem to not play a role in the final resolution. Meaning that the different conformations might be averaged out. This should be commented.

4. Section order. I suggest to exchange figure (and sections) 2 and 3. This yield a smoother cadence for the manuscript, from orgami to 'protein rulers', to biological question.

5. Channel alignment. If DNA origami requires an element by element fiducial realignment for the different exchange round, how is this supposed to affect cell measurements? Is one expected to make local drift corrections based on some reference? Is this the ultimate limitation for measurements in cells?

6. Introduction. The authors mention "Collecting an arbitrarily large number of localizations yields an arbitrary increase in precision, ultimately limited solely by the size of the labeling probe." It should be somehow clarified, especially for the non-expert, that this will indeed yield an extremely precise localization, but with a bias/offset from the epitope of interest.

7. Discussion. The authors mention "even in environments where achieving high optical localization

precision is impossible". This is somewhat vague. To what scenarios do the authors refer? Exemplify. Include numbers and expectations.

8. Discussion. The authors mention "under physiological conditions" for the CD20 measurements. What do the authors mean? The cells are fixed.

Final comment

On quality of figures, clarity of the presentation and message, use of statistics, modelling of nearest neighbor distances and referencing the work is at the highest standards.

Author Rebuttals to Initial Comments:

----- Referees' comments -----
Author response in blue, italic

Referee #1 (Remarks to the Author):

The authors resolve targets on DNA-origami and in cells with single-molecule fluorescence, closer than has been reported previously, with accuracy and resolution apparently limited by the labelling technique. They achieve this by labelling nearby targets with orthogonal labels in different rounds of imaging and finding the centres of the resulting localisation clusters in each round. DNA-PAINT in particular allows the spatial centres of the clusters to be estimated very precisely as result of the replenishment of the imaging strands. They resolve labelled DNA origami targets separated by less than 1 nm and resolve labelled Nup96 copies within the NPC subunit separated by ~12 (xy) by 5 (z) nm in an averaged SMLM NPC structure, which are new achievements and appear to be a clear improvement over the state-of-the-art in this area. They are also able to investigate CD20 dimers with labels separated by less than 10 nm. These results are of immediate interest to many people in the SMLM field, and others concerned with supramolecular protein complex structure are also likely to be interested.

We thank the reviewer for the thorough and very supportive review of our work. We are grateful to the points raised, which we address in our response below.

Some limitations should be highlighted in general, and clarity could be improved in places. The first issue that I believe should be highlighted early on and more often is that the improved resolution is only obtained when targets closer than the DNA-PAINT cluster size are labelled orthogonally. When targets this close are labelled with the same type of docking strand (in the case of DNA-PAINT), RESI will not distinguish them. In the DNA origami, this distinguishability is programmed into the sample, but in cellular imaging, there are probabilities for nearby molecules either being randomly orthogonally labelled or not. The authors do state this and give derivations in supplementary, but in my opinion it would benefit readers for this to be more obvious from the start and in the discussion. As the authors point out, their use of four orthogonal docking strands does still allow for imaging the majority of targets in this way in their demonstrations. A consideration of the probabilities of being able to label and resolve the structure of trimers, tetramers etc. on particular length scales with labelling probabilities in mind might be interesting.

We agree with the reviewer that the degree of orthogonal labeling is an important parameter that has to be considered before starting RESI experiments and during RESI data analysis.

*To improve clarity, we have now included a simple model that can be used to estimate the detection efficiency of DNA-PAINT-unresolvable multimers (dimers to hexamers) depending on the number of orthogonal labels used (see **methods section, formerly supplementary note 3**). Even if all distances between individual molecules within a multimer are not DNA-PAINT resolvable (e.g. all below approx. 10 nm), all multimers up to trimers can be reliably detected with 4-plex RESI, with 75% dimer detection efficiency for dimers and 38% detection efficiency for trimers. DNA-PAINT unresolvable trimers are most likely present if a second NND peak appears in a RESI 4-plex measurement. In this case we would recommend performing a 6- to 8-plex RESI experiment to assess the possibility of tetramers. Please note, that it is unlikely that higher order multimers above tetramers contain only molecules that are unresolvable in DNA-PAINT, which means that even if one or two individual molecules in a pentamer/hexamer were labeled with the same sequence, they can still be detected with DNA-PAINT. This is highlighted in the CD20 case for which the dimer distance is at the limit of DNA-PAINT resolution. Thus, we can already resolve a fraction of the dimers without RESI so labeling two proximal CD20 molecules with the same sequence does not necessarily mean that the two molecules are unresolvable in all cases. In addition to the above mentioned considerations, clustered data of individual RESI imaging rounds can be subjected to a quality check by performing qPAINT analysis. It is expected that two distinct populations for the number of binding events per RESI cluster as the cluster is composed of either one or two molecules (for the proportion of molecules that are*

labeled non-orthogonally). This information could also be further used to estimate the most likely positions of the two non-resolvable molecules.

In order to enable RESI imaging for denser targets (i.e. necessity for more orthogonal sequences) we note that, although the speed optimized sequences are the preferred option, all previously used “classic” DNA-PAINT sequences would work for the imaging rounds (see doi:10.1039/C6SC05420J for >50 orthogonal sequences). A combination of this full set of sequences with automated routines of fluidic exchanges will hence allow to resolve higher-order oligomers at the expense of relatively long acquisition times (which could in turn be alleviated by employing spectral multiplexing, enabling simultaneous three-color acquisition). Furthermore, speed-optimized multiplexing could be extended to 12 rounds using a combination of right- and left-handed [doi:10.1038/s41587-020-00753-y] DNA molecules.

Secondly, considerations of resolution and structural accuracy could be improved. For instance, “label-size-limited” resolution is claimed in the Nup96 data, but then localization inaccuracy from linkage error must become a more important confounding factor in structural imaging and interpretation of structural data. Discussing this is of benefit to readers.

We agree with the reviewer that the impact of the label size on structural measurements is of high relevance to the reader: The finite label size and thus potentially biased orientation leads to linkage error-induced inaccuracies in distance measurements. We have now added a sentence to point the reader to this fact.

In the CD20 case, “RESI at one nanometer resolution” is referred to, but one nanometer resolution (distance between resolved targets) is not demonstrated here and may not be possible in principle because of the size of the labelling nanobody. Referring to estimated localization precision would be preferable.

We thank the reviewer and changed the phrasing accordingly.

Novelty:

Previous work has used the fact that orthogonal labels (e.g. two colour channels in dSTORM) can allow unconfounded distributions, and distances between them, to be obtained for targets closer than the localization precision (e.g. Leterrier, C. et al. Cell Rep 13, 2781 (2015)). This has also included simulations of instances of the same protein with different labels at zero distance from one another (e.g. Levet, F. et al. Nat Commun 10, 2379 (2019)). However, the representation of a cluster of localizations by its weighted mean and weighted standard error on the mean (SEM) may be novel for spatial analysis in SMLM (although such a representation of a data distribution is common in general), and this allows SEM to be used as a new precision metric when the cluster is not confounded by nearby targets in the same channel.

Particularly in the case of DNA-PAINT, when many localizations can be obtained for the same single target, using the SEM results in a useful improvement in effective precision for single, isolated targets, which also allows higher-precision spatial analysis of nearby target distributions, when nearby targets can be orthogonally labelled. Also, a label exchange method like DNA-PAINT also allows this analysis method to benefit in principle from higher localization accuracy for nearby targets than an acquisition method where all targets are permanently labelled and stochastically switched, as the authors point out.

We are grateful to the reviewer for this comment and agree with the points discussed.

However, when nearby targets are randomly labelled in the same channel, the precision of the SEM estimate is not retained. For targets in the same imaging channel, the resolution remains limited by the photonic signal and noise as is usually considered, moving back from RESI resolution to conventional SMLM (e.g. dSTORM) consideration. For their test cases, the authors calculate how what fraction of proteins will be unresolvable using RESI, because of their proximity and random labelling. For unknown samples where the stoichiometry and proximity of targets is not yet known, this

analysis may result by default in an artifactual collapsing of multiple targets onto one super-resolved target. It would be interesting to hear a discussion of this.

We agree with the reviewer and point to our discussion of this issue above.

Data and methodology:

DNA-PAINT is ideal as an acquisition tool for data on which to use this processing method which takes advantage of the availability of many localizations per cluster per imaging channel and multiple channels.

In general the data is visualized well, but in Suppl. Fig. 2, 3, 6, and to a lesser extent 5, when the reader is instructed to zoom in for details, the figure still does not seem very informative after zooming in. In my experience, it was still hard to see RESI localizations and not easy to compare with the adjacent DNA-PAINT image.

We agree with the reviewer and thank her/him for highlighting this. We altered the figures such that the RESI localizations are now more visible and placed them adjacent to the corresponding DNA-PAINT images for better clarity.

I did not find information on how z-information was obtained for 3D localizations, which is important to include.

We apologize for this oversight. 3D DNA-PAINT imaging was performed using an astigmatism lens in the detection path. We now added this information to the "Microscope setup" paragraph in the methods section: "3D imaging was performed using an astigmatism lens (Nikon Instruments, N-STORM) in the detection path. Raw microscopy data was acquired using μ Manager (Version 2.0.1)."

I did not see 3D information in Suppl. Fig. 3, which seems important to include in this figure.

We thank the reviewer for this suggestion and have now modified the figure accordingly to highlight this fact.

The alignment error, although small, is visible and stands out in Extended Data Fig. 2e. This may be worth referring to or explaining.

We agree with the reviewer. Generally, the alignment error ultimately limits the accuracy with which RESI can recover absolute distances. We note that this alignment error is most likely due to the stability/flexibility of the underlying DNA origami structure. For the rectangular origami, the average alignment error is only 0.6 nm and for the case of the DNA origami disk structure, the average alignment error of R4 alignment sites is 1.3 nm: in both cases noticeable but still significantly smaller than the distance of the target strands. We note that the alignment error is significantly reduced down to 0.1 nm in the average structure, pointing towards the structural flexibility and heterogeneity of DNA origamis at this length scale.

Fig. 2d: It may be helpful to explain what the background structure is, i.e. not part of the acquired data.

We thank the reviewer for this suggestion. We have now added a comment in the figure description that this represents the same CryoEM density as in Figure 2b.

Statistics and interpretation:

Fig. 2i and/or main text (could use other parts of Fig. 2 also): It would be helpful to compare with a structure with

linkage errors taken into account, somehow, given that the RESI precision that the authors use is smaller than this linkage error.

We thank the reviewer for bringing up this point. As we do not precisely know the orientation/flexibility of the probe molecules (GFP and nanobody), we can only estimate a range for the precise linkage error, which is in good agreement with the distances we measure given the size of GFP+nanobody.

“This spatial arrangement [Nup96 pairs] is consistently resolved for all Nup96 protein pairs (Extended Data Fig. 3)”: While the resolution obtained appears impressive, Extended Data Fig. 3j does not obviously illustrate this statement in its current form, as the densities are often indistinct or have more than two maxima per protein pair.

We agree with the reviewer and have now modified the statement in the main text, stating that we resolve this for most Nup96 protein pairs. We also highlight the fact that the “outliers” are most likely due to structural heterogeneity combined with limitations in label size and efficiency, which would be further reduced by improvements in labeling technology.

Fig. 3 and text: Information is missing or unclear as to how the distance between the labelled targets was measured from the RESI clusters or their visualizations.

We thank the reviewer for raising this issue: RESI distance measurements are performed by taking the euclidean distance of the RESI localizations, which in turn are the center points of the detected localization clusters.

What are the distance uncertainties? s.d.?

*We are grateful to the reviewer for pointing us to this issue and agree that this was not clear in the main text. The coordinate and the precision for each RESI localization was calculated as the weighted mean and weighted standard error of the mean of their respective DNA-PAINT group of localizations (see **Methods, formerly supplementary note 2**). To determine the uncertainty of the euclidean distance, we performed error propagation.*

7 Å is used as the comparison distance for the results. However, Fig. 3d shows an inter-base pair distance of 7 Å, measured along the orientation of one DNA strand. Defined like this, the average of the inter-base pairs distances in xy would be less than 7 Å. Therefore it is not currently clear whether this is the right quantitative comparison, and the authors should think this through and present the distance and quantitative comparisons again appropriately.

We apologize for the possible confusion and thank the reviewer for bringing up this issue. We note that in fact we measure the backbone distance of two bases along one strand in the helix, and not the distance between two base pairs along the helical axis of the double strands. Staple extensions in this specific DNA origami structure are designed so that this backbone distance is in-plane with the measurement surface, thus we would expect to measure approx. 7 Angstrom distance between both points. We have now added the 0.34 nm distance between two base pairs along the helical axis to further clarify this.

Given that the manuscript currently does use 7 Å for comparison:

8.5 ± 1.7 Å may be considered in agreement with 7 Å, if this is specified in the manuscript to mean within one s.d. The 8.5 ± 1.7 Å result is not compared with 7 Å in the main text, only the Fig. 3 legend.

The 9.5 ± 1.2 Å result should be directly compared with the 8.5 ± 1.7 Å result. The presence of this second experiment should also be explained. I would not necessarily say that 9.5 ± 1.2 Å is in good agreement with 7 Å, because 7 Å is > 2 x s.d. away from 9.5 Å. What the authors mean here may need revising or defining.

We thank the reviewer for raising this concern and in fact pointing us to a mistake in calculating the distance error of the measurement for the mean distances.

- 8.5 Å is the result for the distance of one pair of two adjacent orthogonal strands in a single exemplary DNA origami (the one displayed in Figure 3d). The uncertainty 1.7 Å is calculated by computing the error propagation of the individual RESI-localization precisions of 1.2 Å.
- 9.5 Å is the average distance over $N = 42$ origami, each containing up to 6 pairs of adjacent orthogonal strands. The 2.6 Å uncertainty is calculated by computing the average of the error propagated uncertainties per average point pair. We note that this value was incorrect (it read 1.2 Å before) in the previous version and do apologize for this oversight.
- The value 9.5 ± 2.6 Å is actually in good agreement with 7 Å defined as $7 \text{ Å} \in (6.9, 12.1) \text{ Å}$ using a ± 1 std criterion.

In Extended Data Fig. 4c, a distance from R1-R3 is measured at 1.2 nm, and in 4d, the average distances appear to be ~ 1 nm or greater. How does this fit with the stated 9.5 Å mean? e.g., if this is to do with alignment correction, more explanation and illustration of this is needed.

We are grateful to the reviewer for pointing us to this potentially confusing issue.

Indeed, the primary cause for the discrepancy is likely induced by alignment inaccuracies. The alignment per origami is conducted using R4 alignment sites only. The average, however, is aligned using all structural information available from one imaging round, thus not biasing the individual alignment. Thus, the process of averaging minimizes the R4 alignment error from 6.0 Å to approx. 1.2 Å, and, by minimizing the remaining potential orientation bias at this stage, also retrieves a more accurate estimation of the R1-R3 distances. Due to structural flexibility and heterogeneity of the DNA origami, we believe that this average most faithfully represents the true underlying structure.

By removing panels c-f in **Extended Data Figure 5 (formerly 4)** and panels h-p in **Extended Data Figure 3 (formerly 2)**, which displayed per-origami measurements, we believe we contribute to a clearer and less ambiguous presentation of our experimental results.

In Fig. 3, there is “a distance of 8.5 ± 1.7 Å”, with “a precision of ~ 1.2 Å for the single base-pair backbone distance”, and “an average localization precision of 1.3 Å for the experimental data”. What is meant here needs clarifying.

We have modified the figure description accordingly.

The stated improvement over the state-of-the-art by “approx. a factor of six” seems overly precise, without more detailed justification than referencing other work.

The reports of smallest distances resolved by DNA-PAINT [Dai et al, Nat Nanotechnology 2016], MINFLUX [Balzarotti et al, Science (2017)], and ROSE [Gu et al, Nature Methods 2019] are ~ 5 nm. The “approx. factor of six” statement is based on the single bp DNA origami measurement. However, we have decided to delete that part of the sentence for clarity reasons focusing on our own results and to let the reader compare with other state-of-the-art methods in super-resolution fluorescence microscopy.

In Suppl. Fig. 7b, it would be good to justify the claim of “RESI-localization precision below 1 nm” in this experiment.

We thank the reviewer for pointing us to this issue. The RESI localization precision calculated based on the average number of localizations is (just) below 1 nm. The claim is based on this calculation.

Monomer/dimer simulations:

- The authors used 50% labelling efficiency while the study they referenced found closer to 60% “effective labelling efficiency”, with true labelling efficiency being greater than this, taking into account undetected labelled targets. It is not clear why the current authors chose a different value.

We agree with the reviewer that this leads to potential confusion. We in fact used in-house quantified labeling efficiencies of DNA-labeled anti-GFP-nanobodies. We thus do not expect to match the 60% reported apparent labeling efficiency of dye-labeled antibodies in the referenced study. We have added this accordingly.

- The authors carried out some simulations of monomer/dimer mixtures (Suppl. Note 4). However, the basis of their choice of the proportions reported as their best values is not clear. What is the figure of merit? How big was the range of proportions that gave qualitatively similar results, if a quantitative uncertainty could not be obtained? Can a range of proportions be plotted?

*We appreciate the feedback and have now improved our parameter estimation. We have developed and implemented an iterative non-linear least squares algorithm with the simulations as its basis. A corresponding paragraph in the **Methods** section of the manuscript has been added. This leads to a slightly better estimation of the parameters, yielding 13.5 nm (previously 12 nm) as the average dimer distance, 5.5 nm (previously 5 nm) as the label uncertainty and 47% (previously 45%) as the fraction of dimers.*

- What density of the targets was used for the dCSR simulation (Fig. 4f)? Was this the experimental density? That may help with the comparison, or an explanation of the choice of density.

Yes, it was the experimentally determined density, ~ 50 molecules/ μm^2 .

- A comparison of DNA-PAINT and RESI for Rituximab-treated cells similar to Fig. 4e may help to complete the picture.

*We thank the reviewer for this suggestion and have now added this to **Extended Data Figures 6 and 8** (formerly **Supplementary Figs. 7 and 8**).*

A reference or justification of the use of l_p^2 (as opposed to l_p) in the weighted means and SEMs would be beneficial (Suppl. Note 2).

Under the hypothesis that the localizations are independent and normally distributed with the same mean, the weighted mean based on inverse variances as weights is the maximum likelihood estimator of the mean of the whole set of localizations. Therefore the variance of the weighted mean is minimal (the estimator is optimal) when the inverse variance of individual measurements ($1/l_p^2$) are chosen as weights.

A more detailed explanation of the use of the covariance matrix, including what the covariance(s) is (are) that it describes, would be helpful in Suppl. Note 1a.

The covariance matrix is a widely used calculation to estimate the variance of data sets with dimensions $N > 2$. In this case the covariance of the X and Y coordinates of each RESI-localization is computed. We note that the formula had a mistake, it should read $\text{cov}(x, \mathbf{y})$. A commonly used [e.g. Masullo et al, Nano Letters 2020] metric to evaluate the precision of a localization method using a single scalar value is to compute the average of the diagonal elements of the $\text{cov}(x, y)$ matrix. That is equivalent to compute $\text{VAR} = \text{tr}(\text{cov}(x, y))$, and hence $\sigma = \sqrt{\text{VAR}} = \sqrt{\text{tr}(\text{cov}(x, y))}$.

Suppl. Note 3 (Case 2): The authors note that a script is provided to output graphs of resolution vs density, but according to the nature of the technique and the previous explanation in the Note, this should also include the fraction of proteins that will not be resolved.

We thank the reviewer for pointing out this mistake in the Supplementary Note 3. We have now changed the figure references to the correct ones.

To further increase the clarity, we changed the sentence “A 2D map of density vs resolution vs F can be computed (Supplementary Figure 2c).” to “We plot here F, which is the fraction of non-resolvable molecules, as a function of both density and resolution (Extended Data Figure 10c).”

Conclusions/Discussion:

The new scaling law has been defined and used for new, higher-precision spatial analysis from SMLM when repeated localizations can be correctly identified as associated with the same target site, and when nearby target sites are orthogonally labelled. However, the discussion does not currently mention that in a new sample, there may be more than one nearby target site labelled non-orthogonally, which would by default give misinformation for a fraction of the sample structure. The technique is still useful, and this is more or less of a difficulty depending on the structure, and there may be intelligent ways of taking it into account, but I believe it should be included in the discussion.

We have now added this point to the discussion.

Angstrom spatial resolution does indeed appear to have been achieved by imaging adjacent DNA base pairs, even if the distance is over-estimated (see Statistics and interpretation).

We have now clarified this point by improving the statistical estimation of the distance error.

It is not clear what is meant by an improvement in robustness and precision as a result of averaging out uncertainties such as those due to mechanical instability. This would need to be more explicitly compared with SMLM in general, including DNA-PAINT without RESI.

*Independent sources of uncertainty can affect the localization process even at theoretically unlimited optical precision. Mathematically independent sources of noise can be expressed as: $\sigma_{tot}^{**2} = \sigma_{opt}^{**2} + \sigma_{extra}^{**2}$. The most relevant example is the mechanical instability at frequencies that are comparable with the acquisition frame rate. MINFLUX, MINSTED and RASTMIN will be ultimately limited in precision by the quality of the mechanical instability (also called “drift”) correction of the experimental setup. This has been discussed qualitatively by Balzarotti et al (Science 2017, MINFLUX), Weber et al (Nature Photonics 2021, MINSTED) and has also been modeled quantitatively by Masullo et al (Light: Science and Applications 2022, RASTMIN - Fig. 2d and Suppl Fig. 7). Depending on the precision of the active stabilization system, the residual localization uncertainty due to mechanical vibrations lies in the 0.5 - 2.0 nm range, posing a hard limit to the total localization precision even in the case of extremely high optical localization precision.*

RESI relaxes this factor substantially by averaging out any remaining significant source of mechanical noise. This allows to reliably obtain 0.1 nm total localization precision even without an active feedback-loop based stability system using conventional drift correction using cross correlation and fiducial markers.

Even though RESI SMLM requires many localizations per target site, it is still a widefield technique, which does give it a speed advantage over MINFLUX and MINSTED as mentioned, which must currently sequentially attempt localization of only one molecule in the FOV (or several, when these techniques are parallelized). Techniques like SIMFLUX and SIMPLE may be worth mentioning, which give a smaller improvement in resolution over widefield SMLM than RESI, but at a higher speed.

We have now modified the text to ensure a balanced comparison of RESI to other SMLM methods and cite work appropriately.

“Quantifying receptor organization down to single protein level for any receptor density”: It is perhaps not clear that this applies for all densities, for all receptors. Receptors with higher stoichiometry, for instance, may need more consideration with regards to challenge of the probability of orthogonal labelling and dealing with data from non-orthogonally labelled complexes. It would be interesting to consider what the local density limit is for RESI to be effective.

We agree with the reviewer on this and think that this point is covered by the improved discussion regarding the orthogonal labeling of targets and achievable target separability in RESI. We have furthermore modified the sentence in the discussion.

“the method of choice for studying three-dimensional protein structure and conformational changes”: This probably needs some elaboration or refinement, since it is not clearly better than cryo-EM or other techniques, but could be complementary to them. The following statement is better, about bridging the gap.

We agree with the reviewer and have modified the statement in the discussion section.

References:

References to recent advances involving expansion microscopy may also be beneficial for comparison. Possibly also SIMFLUX, SIMPLE, etc. as mentioned. A reference to and comparison with RASTMIN, along with MINFLUX and MINSTED, would also benefit the general reader.

We have now added references to these techniques.

Clarity:

Main text line 1: This definition of localization precision is no longer universal for SMLM, as this now includes MINFLUX etc. and SIMFLUX etc.. Referring to “widefield Single-Molecule Localization Microscopy” instead would be correct.

We thank the reviewer for this comment. We agree and have modified the text accordingly.

Bottom page 4: Suppl. Fig. 4 is referenced, but this does not describe the situation for Nup96 labelling and imaging. Suppl. Fig. 4 is instead relevant to the CD20 experiment.

We have deleted the reference to Supplementary Figure 4 in this case.

Referee #2 (Remarks to the Author):

The manuscript by Reinhardt et al. presents an exciting fluorescence microscopy method to determine distances between biomolecules at unprecedented resolution within the angstrom range. The method, called RESI, can be applied to detect single molecule arrangements within oligomeric proteins, as well as macromolecular complexes in a cellular context. This method will help to fill an important gap between in vitro single-particle cryo-EM imaging and in cellulo cryo-ET, as well as current super-resolution fluorescence microscopy methods. As such, I am confident that this work will be of great interest to the structural and cellular biology communities.

The text is concise and clear, and the figures nicely done to support the scientific claims. Although the results obtained applying RESI to DNA origamis and, in the cellular context to the nuclear pore complex are impressive and conclusive, those regarding the human receptor CD20 appear somehow weaker:

We are grateful to this reviewer for the very supportive review of our work and are happy to clarify the points raised below.

1 - The introductory paragraph in the CD20 section doesn't describe clearly previous single-particle cryo-EM work on CD20 in complexes with fragments from therapeutic antibodies. Rouge et al. have shown that CD20 forms homodimers in complex with two RTX monovalent Fab molecules. In such complex, each Fab molecule binds a large area on the CD20 extracellular surface that extends over the 2 protomers. Moreover, the two Fab molecules bound to dimeric CD20 form extensive homotypic interactions, casting some doubts on whether the CD20 homodimer is induced or facilitated by binding of RTX-Fabs. This is highly unlikely because the structural complementarity of the CD20 protomer-protomer interface is very good. Moreover, independent cryo-EM and biophysical studies by Kumar et al. showed that in complex with divalent F(ab')₂ fragments of Ofatumumab, another type I antibody, the individual Fab molecules bind to only one CD20 protomer, and lack homotypic interactions, yet CD20 displays an identical dimeric structure, demonstrating that the homodimer is indeed the native oligomeric arrangement of CD20, and it is not induced by RTX Fab binding. Finally, SEC-MALS analysis of purified CD20 in different detergent solutions showed that the CD20 elutes as dimers (Kumar et al. Supp Fig. 1).

Therefore, the sentence in the current manuscript stating "Furthermore, both studies cannot address CD20 organization in the absence of RTX Fabs" is misleading, please edit this part.

We thank the reviewer for pointing us to this issue. We also agree that there is convincing evidence for CD20 dimers in the absence of RTX (and any other) antibodies and state this in the text. We have modified our statement such that it now reads:

"However, current studies cannot assess CD20 organization in the context of intact cells bound to RTX full antibodies to verify this proposed model. Moreover, even though in vitro studies showed that CD20 dimers can form without antibody-binding, currently the quantitative assessment of CD20 dimerization in untreated cells is missing."

2 - It has further been shown using cryo-EM and divalent RTX F(ab)₂' that due to steric constraints the two monovalent Fab molecules that bind a CD20 dimer come from different F(ab)₂' molecules (Kumar et al.). As a natural consequence of this, type I IgG molecules concatenate CD20 dimers upon binding, as demonstrated in vitro. Therefore, the result presented in the manuscript suggesting that RTX concatenate CD20 dimers in cells is more confirmatory than revealing. What in my view is more novel, and it is a direct consequence of working on the more native context of the cell is fact that RTX forms linear rather than circular RTX-CD20 higher-order concatenated assemblies, as observed in vitro and likely due to free tumbling of the molecules in detergent solutions (Rouge et al., and Kumar et al.). But such discussion is missing in the text. Therefore, I would suggest to tone-down a bit the revealing aspect of the discovery, and highlight the fact that as opposed to detergent solutions, in cells the predominant assemblies are linear, as it appears in the images in Supp Fig 8 of the manuscript. Also, do the authors detect any indication of circular arrangements? And is the number of CD20 molecules in the linear arrangements rather constant or random?

We agree with the reviewer and thank her/him for pointing us towards the investigation of linear vs. circular CD20 arrangements in intact cells. To this end, we have added a simulation of hexameric ("ring-like") arrangements of CD20 molecules including probe uncertainty for the fluorescence microscopy study. We quantify the simulated and experimental data by cluster analysis and subsequent circularity assessment. We come to the conclusion that it seems unlikely that isolated circular arrangements of CD20 molecules exist in intact cell membranes. Furthermore, the number of molecules per cluster is not constant. However, we hypothesize that CD20 arrangements with densities and distances compatible with C1q docking are likely to occur within the observed cluster patterns (See also comment to point 3 below). We thank the reviewer again for suggesting such an analysis. We also have adapted our statements in the text to highlight the more confirmatory nature of our results.

3 - The above points are mechanistically important, because type I mAbs (RTX or OFA), as opposed to type II, bound to CD20 are expected to form penta- and hexameric circular arrangements that potently activate the complement system by binding to its C1q component, which is a hexa-headed protein. RESI seems a method ideally suited to explore this kind of questions, and it would be really interesting to see the RESI analysis of cells opsonized with type I (RTX) vs type II (OBZ) mAbs, and the effect of C1q on the two type of cells. I leave to the authors the decision on whether such experiments belong to the current manuscript, but in the absence of more data like those, I would strongly suggest to lower the “revealing” tone of the results regarding CD20 concatenation.

We are grateful to the reviewer for this suggestion and very much agree that this should be a logical next step in the assessment of the mode of action of type I vs. type II mAbs. However, as the focus of this work is on the technology development of RESI and proof-of-concept applications rather than deeper biological investigations, we aim to study these in a follow-up.

Referee #3 (Remarks to the Author):

The manuscript by Reinhardt et al. presents an optical super-resolution imaging modality where targeted sites are located with precisions in the order of one to ten Angstroms. This is successfully demonstrated in samples of DNA-origami and antiGFP-nanobody-labelled nucleoporin proteins and B-cell receptors.

The crucial elements to achieve such performance are sample (1) sparsity and (2) repeated localizations. Sparsity is a long-harvested feature of DNA-PAINT: fluorophores are transiently at their targets, thus bypassing unwanted photophysical interactions between dyes belonging to neighboring targets. Yet, when targets are closer than the localization precision, it's not possible to distinguish them nor to combine several localizations to further improve the resulting precision by averaging. To solve this problem, Reinhardt et al. induced further sparsity by randomly labeling with barcodes and imaging sequentially.

One could argue for hours whether the concept presented here is novel or not. The authors rightfully acknowledge that their work is a "reapplication" of the sparse localization trick that broke the diffraction limit. This time it is used to improve limitations brought by photon emission rate/ binding time/background. It must be stated that these limitations are not fundamental: a hypothetical brighter dye or a more efficiency localization scheme (the type of MINFLUX/MINSTED) would yield higher precisions and allow to combine localization events. However, those performance limitations are very real and the authors present a powerful and accessible approach to overcome them.

The authors have reexamined fundamental aspects of super resolution imaging and produced an explicit and direct procedure with outstanding performance. One that has been all along in front of our eyes, since the technology and the reagents were available for almost a decade.

For the reasons stated above I absolutely recommend the manuscript for publication, provided that the points below are addressed.

We are very grateful to this reviewer for the supportive and encouraging comments on our manuscript and are happy to address the issues raised.

Major points

1. Beyond dimers

The authors present an analysis of the stochastic labelling in the case of dimers and of randomly distributed targets. As the authors claim the applicability of RESI to bridge light microscopy and structural biology, the concrete request for the

authors is to include an analysis of the scalability of the method beyond dimers, as the required number of strands will increase tremendously.

A back of the envelope calculation yields $P(\text{diff. seq.}) = \frac{\#\text{diff.seq.}}{\#\text{all seq.}} = \frac{n^h}{(n! / (n-h)!)}$, with n the number of distinct strands and h the order of the multimer. Already for analyzing trimer with $n=4$ strands, only 37% of completely labeled trimers would have distinct strands. With a fifth strand this goes to 48%. Tetramers and pentamers are below 20% for 4 and 5 strands. The limitations in this regard must be explicitly pointed out. Likewise if there are ways to bypass them. For example, the authors have used the fastest docking strands available with repeated multiple binding to increasing the k_{on} , this might put a limit to how many strands can be really used in the future, as these are minimalist strands with repetitions and not many orthogonal sets can be assembled.

As this issue has been raised by Reviewer #1 as well, we reiterate our comment below.

We agree with the reviewer that the degree of orthogonal labeling is an important parameter that has to be considered before starting RESI experiments and during RESI data analysis.

*To improve clarity, we have now included a simple model that can be used to estimate the detection efficiency of DNA-PAINT-unresolvable dimers to hexamers depending on the number of orthogonal labels used (see **methods section, formerly supplementary note 3**). Even if all distances between individual molecules within a multimer are not DNA-PAINT resolvable (e.g. all below approx. 10 nm), all multimers up to trimers can be reliably detected with 4-plex RESI, with 75% dimer detection efficiency for dimers and 38% detection efficiency for trimers. DNA-PAINT unresolvable trimers are most likely present if a second NND peak appears in a RESI 4-plex measurement. In this case we would recommend performing a 6- up to 8-plex RESI experiment to assess the possibility of tetramers. Please note, that it is unlikely that higher order multimers above tetramers contain only molecules that are unresolvable in DNA-PAINT, which means that even if one or two individual molecules in a pentamer/hexamer were labeled with the same sequence, they can still be detected with DNA-PAINT. This is highlighted in the CD20 case for which the dimer distance is at the limit of DNA-PAINT resolution, Thus, we can already resolve a fraction of the dimers without RESI so labeling two proximal CD20 molecules with the same sequence does not necessarily mean that the two molecules are unresolvable in all cases.*

In addition to the above mentioned considerations, clustered data of individual RESI imaging rounds can be subjected to a quality check by performing qPAINT analysis. It is expected that two distinct populations for the number of binding events per RESI cluster as the cluster is composed of either one or two molecules (for the proportion of molecules that are labeled non-orthogonally). This information could also be further used to estimate the most likely positions of the two non-resolvable molecules.

In order to enable RESI imaging for denser targets (i.e. necessity for more orthogonal sequences) we note that, although the speed optimized sequences are the preferred option, all previously used "classic" DNA-PAINT sequences would work for the imaging rounds (see doi:10.1039/C6SC05420J for >50 orthogonal sequences). A combination of this full set of sequences with automated routines of fluidic exchanges will hence allow to resolve higher-order oligomers at the expense of relatively long acquisition times (which could in turn be alleviated by employing spectral multiplexing, enabling simultaneous three-color acquisition). Furthermore, speed-optimized multiplexing could be extended to 12 rounds using a combination of right- and left-handed [doi:10.1038/s41587-020-00753-y] DNA molecules.

2. Beyond one color

The authors present superb performance for single-color imaging, but there is absolutely no mention of imaging distinct species. It could be argued that this is beyond the scope of the paper, but again, as the authors present the method as a bridge with structural biology, it is a must to distinguish e.g. distinct subunits of a complex. It could also be argued that this might be trivial, as distinct subunits could be targeted with distinct strands directly. The concrete request is, again, to include an analysis of the scalability of RESI for multi color scenarios.

We thank the reviewer for raising this important point. Extension of RESI to "multicolor" imaging of distinct molecular species could be achieved in a straightforward "brute force" approach by assigning specific sets of orthogonal sequences to a specific species. With respect to speeding up this process, existing multiplexing approaches such as spectral demixing etc. could be combined with RESI.

As discussed above, there is the option to increase the number of orthogonal sequence species by resorting to non-speed-optimized DNA-PAINT sequences.

3. Timings

The authors explicitly mention the long acquisition times of around 100 min for full field of view images of RESI. This counts for figure 2 and 3. They must also explicitly mention the experiments on fig. 4 take around 4.4 hour for each conditions, adding up to ~8-9 hours.

We agree with the reviewer that this should be pointed out more explicitly. We now have added this information to the main text.

The authors should include (e.g. in supplementary figure 4) an approximation of the acquisition time for the sake of experiment design. For example, combining the expected k_{on} of the strands, the density of the expected targets and the desired M repetitions that achieve a given resolution.

We agree with the reviewer that this information is quite useful. We now have added a section to the methods part.

4. Illumination scheme

It is not reported in these experiments if the illumination is TIRF, HiLo or what kind. I presume it is TIRF. The authors should report the illumination scheme and whether it leads to limitations.

We apologize for this oversight and have now added this information to the methods section. In brief, TIR illumination has been used for the two- and three-dimensional DNA origami data as well as the CD20 acquisition. HILO has been employed for the acquisition of the NPC data.

Minor points

1. I encourage the authors not to use ill-defined terms such as 'ultra-resolution'. This work is still super resolution microscopy, with more resolution than usual. The field suffers from the use of ill-defined terms and metrics.

We agree with the reviewer that ultra-resolution is an ill-defined term. We now changed this accordingly.

2. The authors highlight their scaling with $1/\sqrt{M}$. I encourage them to use log-log scale in figures 1f and 3e to further highlight it, and also make it clear when/if the curves saturate.

We thank the reviewer for this comment. However, we would like to keep the current linear plot of the scaling, as we find this more instructive and intuitive to understand for a broader audience. We experimentally observe no saturation.

3. The 5x and 7x repetitions of the docking strands seem to not play a role in the final resolution. Meaning that the different conformations might be averaged out. This should be commented.

We agree with the reviewer that this seems initially surprising, but have added a statement in the appropriate section of the discussion.

4. Section order. I suggest to exchange figure (and sections) 2 and 3. This yield a smoother cadence for the manuscript, from orgami to 'protein rulers', to biological question.

We appreciate the comment regarding the section order but respectfully disagree with the reviewer on this point. We would rather like to keep the current order based on the following logic: Section 2 is concerned with testing of RESI with well-established in vitro and cellular systems. Section 3 shows for the first time sub 1-nm resolution in optical microscopy, one of the most intriguing results of our work.

5. Channel alignment. If DNA origami requires an element by element fiducial realignment for the different exchange round, how is this supposed to affect cell measurements? Is one expected to make local drift corrections based on some reference? Is this the ultimate limitation for measurements in cells?

This points to a very interesting question. We note that due to the nature of the surface attachment chemistry (BSA/Biotin/Streptavidin/Biotin/Origami), the DNA origami structures tend to undergo translational and rotational movement in the lower single-digit nanometer scale on the surface. However, we argue that this limitation is actually less of an issue for cellular experiments, where samples are covalently crosslinked by fixation with many more surface attachment points as compared to DNA origami structures.

6. Introduction. The authors mention "Collecting an arbitrarily large number of localizations yields an arbitrary increase in precision, ultimately limited solely by the size of the labeling probe." It should be somehow clarified, especially for the non-expert, that this will indeed yield an extremely precise localization, but with a bias/offset from the epitope of interest.

We agree with the reviewer and have now modified the text also with regards to points raised by reviewer #1. For completeness, we here state our response again: The impact of the label size on structural measurements is of high relevance to the reader: The finite label size and thus potentially biased orientation leads to linkage error-induced inaccuracies in distance measurements. We have now added a sentence to point the reader to this fact.

7. Discussion. The authors mention "even in environments where achieving high optical localization precision is impossible". This is somewhat vague. To what scenarios do the authors refer? Exemplify. Include numbers and expectations.

We agree with the reviewer that this is somewhat of a vague statement. As we have included more discussion about the requirements for efficient stochastic RESI labeling, we now decided to remove this comment in the discussion section.

8. Discussion. The authors mention "under physiological conditions" for the CD20 measurements. What do the authors mean? The cells are fixed.

We thank the reviewer for pointing us to this potentially confusing point. We have now clarified this by mentioning that measurements have been performed in cells rather than in vitro.

Final comment

On quality of figures, clarity of the presentation and message, use of statistics, modelling of nearest neighbor distances and referencing the work is at the highest standards.

We are very grateful and honored by the appreciation of our work by the reviewer and thank her/him again for the very supportive assessment of our work.

Reviewer Reports on the First Revision:

Referee #1 (Remarks to the Author):

The manuscript has been enhanced by the authors' work on it. The following details remain to consider in this revision.

Following the authors' responses:

In their response, the authors explained that there are many available DNA-PAINT sequences available for multiplexing. This would be helpful to include in the manuscript.

It is still very difficult to see RESI details in Suppl. Figs. 1b and 4b. Perhaps a smaller number of origamis could be shown at higher magnification (as in 1a and 4a), with a link to be able to access the rest of the data elsewhere? Perhaps similarly in Suppl. Figs. 2 and 3, although the colours seem to help in that case.

I could not really see RESI data in Extended Data Fig. 7b top panel, even when zooming in quite a way. Maybe this display of DNA-PAINT/RESI is not helpful at this scale, and it would be best to only rely on the higher magnification views in 7b middle and bottom panels and in 7c.

Suppl. Fig. 1 has unexplained 'a' and 'b' panels.

Suppl. Figs. 2 and 3. These contain 'a' and 'b' panels. The legend should explain what these are. I assume the colour scale applies to both 'a' and 'b', but it would also be helpful to spell this out.

Fig. 3: The authors responded that the 7 Å backbone distance is specifically in the plane of the measurement surface. This information is helpful and is also needed in the text of the manuscript. Perhaps specifically describing this as 2D imaging may also help.

Line 162-163: It is the RESI localizations, not the docking strands, that have 1.2 Å localization precision.

It would be good to bring the mention in the main text of the 7 Å (7.0 Å?) design distance closer to this result as well.

It must now be clearly specified that the uncertainty in the 9.5 ± 2.6 nm distribution of distance measurements is the standard deviation of the central distance estimates. Otherwise this will be confused with the SEM used elsewhere, which could be misleading.

I agree that 7 Å is within 1 s.d. of the mean result, but it is right on the edge, so I would recommend clarifying this by using a similar phrase to 'within 1 s.d.', rather than just 'good'.

The authors state that the measurement is not biased upwards by the electrostatic repulsion because 9.5 Å is within 2.6 (1 s.d.) of 7 Å. It would be more convincing to give the result of a one-

tailed statistical test for this upwards bias.

It would be helpful to use 7.0 Å (if that is the case), rather than both 7 Å and ~7 Å (main text and legend). Similarly, to use only 1.2 Å precision, if that is the case, rather than both 1.2 Å and ~1.2 Å.

The authors now use only average representations of the alignment error in Extended Data Figs. 5 and 3. However, this notation, e.g. 1.1 Cl (0, 4.14) Å, is confusing. It needs to be well-explained, possibly each time, or possibly changed and a new notation explained.

The authors explained in their response that RESI localization precision was calculated at just under 1 nm from the numbers of localizations. The reader would benefit from a justification, and the numbers in the calculation made would be the best, perhaps in a supplementary table containing similar calculations for all of the experiments. I think this would be useful material.

The authors explained that they used in-house quantification of labelling efficiency of CD20 at 50%, but this detail is missing from the text. Is there a method they can cite for this?

The authors explained that they used experimentally determined density at ~50 molecules/μm² in the simulation of Fig. 4f. This number is missing in the Methods.

Similar explanation of the densities for the CSR distributions is needed for 4g and 4h, if they are generated in the same way (or other explanation of they are not).

Discussion: The authors explained that non-optical Gaussian uncertainties in the measurement are averaged out by RESI, including mechanical instability. Are there any other examples of such extra uncertainties that are dealt with, or it is only mechanical instability?

The authors also state that this is unique compared with other SMLM techniques, but is it also true of DNA-(or other-)PAINT without RESI?

The authors have now included references to other, faster but less precise SMLM methods in the introduction, but this comparison was not discussed. Including such a discussion may be beneficial to the general reader.

Other points:

The authors say they have resolved base pairs at 9.5 Å separation in the introduction (lines 25-26), but I would suggest they could more accurately say the pairs of base pairs they resolved were separated by 7.0 Å (if this is correct).

There is a missing localizations spot in Extended Data Fig. 2c, compared with 2d. This is confusing.

Extended Data Fig. 3h high-magnification views: Needs scale bars for the left- and right-hand parts of each part of the average result.

Line 142: It could be good to first mention the 7.0 Å (if that is correct) design distance around here.

Lines 220-221: Should this say Extended Data Fig. 6a, instead of 6a-c, also 7a instead of 7a-c and 8a instead of 8a-c?

Line 241-242 and Extended Data Fig. 9: It seems a bit strong to say that isolated hexamers are proved to be completely absent, but at least they are not predominant according to the results.

Line 260: 'two adjacent imagers are never present simultaneously' is only fully true for orthogonally labelled molecules of interest.

Methods: Imaging parameters and duration: Lines 538-539. This was hard to follow. Can more steps be inserted in the mathematical argument?

[see also attached file with referee 1's comments on issues raised by referee 3 and the authors' responses to those]

Referee #2 (Remarks to the Author):

The authors made an honest effort to improve the manuscript, and only I have a minor comment regarding CD20: In line 192 "... CD20 in the presence of full antibody ref36" the reference there should be 37 (Kumar et al), and not 36, as in reference 36 the CD20-mAb rings were hypothesised as physiological complexes. Please, also in that paragraph it should be somewhere mentioned that the cryo-EM analyses were made in detergent solutions, otherwise it will be hard for a reader who is not familiar with the cryo-EM papers to understand the problem regarding detergent vs cell contexts.

Referee #1 (Remarks to the Authors, regarding their response to issues raised by referee 3 in initial round of review):

Referee #1 response in red

2. Beyond one color

The authors present superb performance for single-color imaging, but there is absolutely no mention of imaging distinct species. It could be argued that this is beyond the scope of the paper, but again, as the authors present the method as a bridge with structural biology, it is a must to distinguish e.g. distinct subunits of a complex. It could also be argued that this might be trivial, as distinct subunits could be targeted with distinct strands directly. The concrete request is, again, to include an analysis of the scalability of RESI for multi color scenarios.

We thank the reviewer for raising this important point. Extension of RESI to "multicolor" imaging of distinct molecular species could be achieved in a straightforward "brute force" approach by assigning specific sets of orthogonal sequences to a specific species. With respect to speeding up this process, existing multiplexing approaches such as spectral demixing etc. could be combined with RESI.

As discussed above, there is the option to increase the number of orthogonal sequence species by resorting to non-speedoptimized DNA-PAINT sequences.

> I did not see this in the revised manuscript. It would be a useful note to add in

3. Timings

The authors explicitly mention the long acquisition times of around 100 min for full field of view images of RESI. This counts for figure 2 and 3. They must also explicitly mention the experiments on fig. 4 take around 4.4 hour for each conditions, adding up to ~8-9 hours.

We agree with the reviewer that this should be pointed out more explicitly. We now have added this information to the main text.

> Addressed

The authors should include (e.g. in supplementary figure 4) an approximation of the acquisition time for the sake of experiment design. For example, combining the expected k_{on} of the strands, the density of the expected targets and the desired M repetitions that achieve a given resolution.

We agree with the reviewer that this information is quite useful. We now have added a section to the methods part.

> Addressed ('Imaging parameters and duration')

4. Illumination scheme

It is not reported in these experiments if the illumination is TIRF, HiLo or what kind. I presume it is TIRF. The authors should report the illumination scheme and whether it leads to limitations.

We apologize for this oversight and have now added this information to the methods section. In brief, TIR illumination has been used for the two- and three-dimensional DNA origami data as well as the CD20 acquisition. HILO has been employed for the acquisition of the NPC data.

> Addressed ('Microscope setup'), apart from possible limitations (as far I see), but I would suggest that such limitations are common to DNA-PAINT in general, and may not be needed in this paper.

Minor points

1. I encourage the authors not to use ill-defined terms such as 'ultra-resolution'. This work is still super resolution microscopy, with more resolution than usual. The field suffers from the use of ill-defined terms and metrics.

We agree with the reviewer that ultra-resolution is an ill-defined term. We now changed this accordingly.

> Addressed

2. The authors highlight their scaling with $1/\sqrt{M}$. I encourage them to use log-log scale in figures 1f and 3e to further highlight it, and also make it clear when/if the curves saturate.

We thank the reviewer for this comment. However, we would like to keep the current linear plot of the scaling, as we find this more instructive and intuitive to understand for a broader audience. We experimentally observe no saturation.

> I agree that the current scale is more intuitive to understand.

> The markings on the axes are in unusual places though, particularly $\sigma_{RESI} = 1, 5, 8, 12, \dots$ in Fig. 3e. Should '5' read '4'? Apart from that I can see why they have been chosen this way.

> However, this has highlighted that Fig. 1f does not go down to $\sigma_{RESI} = 0$ (stopping at about $\sigma_{RESI} = 0.5$). The reader would benefit from the scale going down to zero, so that an illusion is avoided where σ_{RESI} appears to approach zero more closely than the actual estimates.

3. The 5x and 7x repetitions of the docking strands seem to not play a role in the final resolution. Meaning that the different conformations might be averaged out. This should be commented.

We agree with the reviewer that this seems initially surprising, but have added a statement in the appropriate section of the discussion.

> I did not find this statement in the discussion.

4. Section order. I suggest to exchange figure (and sections) 2 and 3. This yield a smoother cadence for the manuscript, from origami to 'protein rulers', to biological question.

We appreciate the comment regarding the section order but respectfully disagree with the reviewer on this point. We would rather like to keep the current order based on the following logic:

Section 2 is concerned with testing of RESI with well-established in vitro and cellular systems. Section 3 shows for the first time sub 1-nm resolution in optical microscopy, one of the most intriguing results of our work.

> I am happy with the order not being changed.

5. Channel alignment. If DNA origami requires an element by element fiducial realignment for the different exchange round, how is this supposed to affect cell measurements? Is one expected to make local drift corrections based on some reference? Is this the ultimate limitation for measurements in cells?

This points to a very interesting question. We note that due to the nature of the surface attachment chemistry (BSA/Biotin/Streptavidin/Biotin/Origami), the DNA origami structures tend to undergo translational and rotational movement in the lower single-digit nanometer scale on the surface. However, we argue that this limitation is actually less of an issue for cellular experiments, where samples are covalently crosslinked by fixation with many more surface attachment points as compared to DNA origami structures.

> I do not feel this must be added to the paper.

6. Introduction. The authors mention "Collecting an arbitrarily large number of localizations yields an arbitrary increase in precision, ultimately limited solely by the size of the labeling probe." It should be somehow clarified, especially for the non-expert, that this will indeed yield an extremely precise localization, but with a bias/offset from the epitope of interest.

We agree with the reviewer and have now modified the text also with regards to points raised by reviewer #1. For completeness, we here state our response again: The impact of the label size on structural measurements is of high relevance to the reader: The finite label size and thus potentially biased orientation leads to linkage error-induced inaccuracies in distance measurements. We have now added a sentence to point the reader to this fact.

> This is added at the end of 'RESI resolves single nuclear pore complex proteins'. The discussion also alludes to this ('RESI performance and accuracy...'), but could benefit from a more direct reminder of this limitation as well.

7. Discussion. The authors mention "even in environments where achieving high optical localization precision is impossible". This is somewhat vague. To what scenarios do the authors refer? Exemplify. Include numbers and expectations.

We agree with the reviewer that this is somewhat of a vague statement. As we have included more discussion about the requirements for efficient stochastic RESI labeling, we now decided to remove this comment in the discussion section.

> Ok

8. Discussion. The authors mention "under physiological conditions" for the CD20 measurements. What do the authors mean? The cells are fixed.

We thank the reviewer for pointing us to this potentially confusing point. We have now clarified this by mentioning that measurements have been performed in cells rather than in vitro.

> Ok

Final comment:

On quality of figures, clarity of the presentation and message, use of statistics, modelling of nearest neighbor distances and referencing the work is at the highest standards.

We are very grateful and honored by the appreciation of our work by the reviewer and thank her/him again for the very supportive assessment of our work.

Author Rebuttals to First Revision:

Author response in blue, italic

-----Comments on revised manuscript----

Referee #1:

The manuscript has been enhanced by the authors' work on it. The following details remain to consider in this revision.

We thank the reviewer for the supportive review of our revised manuscript.

Following the authors' responses:

In their response, the authors explained that there are many available DNA-PAINT sequences available for multiplexing. This would be helpful to include in the manuscript.

We have now added a reference that points the reader to currently available sequences for multiplexing (approx. 52 sequences).

It is still very difficult to see RESI details in Suppl. Figs. 1b and 4b. Perhaps a smaller number of origamis could be shown at higher magnification (as in 1a and 4a), with a link to be able to access the rest of the data elsewhere? Perhaps similarly in Suppl. Figs. 2 and 3, although the colours seem to help in that case.

I could not really see RESI data in Extended Data Fig. 7b top panel, even when zooming in quite a way. Maybe this display of DNA-PAINT/RESI is not helpful at this scale, and it would be best to only rely on the higher magnification views in 7b middle and bottom panels and in 7c.

We thank the reviewer for pointing our attention to these issues. The problem likely arose during PDF file compression. We will make sure that the final production version consists of vectorized PDF images with highest quality.

Suppl. Fig. 1 has unexplained 'a' and 'b' panels.

We apologize for the oversight and have now modified the figure description accordingly.

Suppl. Figs. 2 and 3. These contain 'a' and 'b' panels. The legend should explain what these are. I assume the colour scale applies to both 'a' and 'b', but it would also be helpful to spell this out.

We apologize for the oversight and have now modified the figure description accordingly.

Fig. 3: The authors responded that the 7 Å backbone distance is specifically in the plane of the measurement surface. This information is helpful and is also needed in the text of the manuscript. Perhaps specifically describing this as 2D imaging may also help.

We agree with the reviewer and have modified the text accordingly.

Line 162-163: It is the RESI localizations, not the docking strands, that have 1.2 Å localization precision.

We agree with the reviewer, the localization precision does indeed refer to the RESI localizations. To clarify this sentence, we removed the reference to the localization precision of RESI localizations here. The information is presented more clearly in the caption of Figure 3.

It would be good to bring the mention in the main text of the 7 Å (7.0 Å?) design distance closer to this result as well.

As suggested by the reviewer, we now state the design distance at the beginning of the section.

It must now be clearly specified that the uncertainty in the 9.5 ± 2.6 nm distribution of distance measurements is the standard deviation of the central distance estimates. Otherwise this will be confused with the SEM used elsewhere, which could be misleading.

We agree, and have now specified this in the caption of Extended Data Figure 5.

I agree that 7 Å is within 1 s.d. of the mean result, but it is right on the edge, so I would recommend clarifying this by using a similar phrase to 'within 1 s.d.', rather than just 'good'.

We have clarified this point in the legend of Figure 3 and main text.

The authors state that the measurement is not biased upwards by the electrostatic repulsion because 9.5 Å is within 2.6 (1 s.d.) of 7 Å. It would be more convincing to give the result of a one-tailed statistical test for this upwards bias.

We have performed the one-tailed statistical test, yielding a p-value of 0.0028. We have removed the statement about no bias based on electrostatic repulsion.

It would be helpful to use 7.0 Å (if that is the case), rather than both 7 Å and ~7 Å (main text and legend). Similarly, to use only 1.2 Å precision, if that is the case, rather than both 1.2 Å and ~1.2 Å.

We now use $\sim 7 \text{ \AA}$ at all positions referring to the crystallography measurement. We also clarified the precision by stating “an average precision of 1.2 \AA ”.

The authors now use only average representations of the alignment error in Extended Data Figs. 5 and 3. However, this notation, e.g. 1.1 CI (0, 4.14) A, is confusing. It needs to be well-explained, possibly each time, or possibly changed and a new notation explained.

We thank the reviewer for raising this issue. We have modified and streamlined the notation so it adheres to the common notation of confidence intervals (which is now also mentioned in the figure description).

The authors explained in their response that RESI localization precision was calculated at just under 1 nm from the numbers of localizations. The reader would benefit from a justification, and the numbers in the calculation made would be the best, perhaps in a supplementary table containing similar calculations for all of the experiments. I think this would be useful material.

We have now added a supplementary table with the average number of localizations K per binding site for each experiment.

The authors explained that they used in-house quantification of labelling efficiency of CD20 at 50%, but this detail is missing from the text. Is there is a method they can cite for this?

We are currently preparing a manuscript for this novel quantification method. We have added a description of the approach in the methods section with a comment pointing towards the work in preparation. The added section reads as follows:

“For quantification of the labeling efficiency of the DNA-conjugated GFP nanobody, we used a transiently transfected CHO cell line expressing a GFP- and Alfa-tag at the c-terminus of a monomeric membrane protein (e.g. CD86). We then labeled GFP- and Alfa-tag using their cognate nanobodies conjugated to two orthogonal docking sequences and performed two rounds of Exchange-PAINT. We then obtained the best-fitting parameters for a sample composed by pairs of GFP/Alfa-tag, and isolated Alfa-tags, similarly to how the CD20 dimers/monomers analysis is performed. The ratio of these two populations is then used as an estimation of the labeling efficiency. The full details of the quantification approach will be available in a manuscript currently in preparation.”

The authors explained that they used experimentally determined density at ~ 50 molecules/ μm^2 in the simulation of Fig. 4f. This number is missing in the Methods.

We apologize for the oversight and have now added the relevant information.

Similar explanation of the densities for the CSR distributions is needed for 4g and 4h, if they are generated in the same way (or other explanation of they are not).

We now state in the main text that the CSR distributions are performed at the experimentally measured density.

Discussion: The authors explained that non-optical Gaussian uncertainties in the measurement are averaged out by RESI, including mechanical instability. Are there any other examples of such extra uncertainties that are dealt with, or it is only mechanical instability?

For clarity, we rephrased as “averaging out the uncertainty effects of mechanical instability (sigma_MEC), provided the latter is normally distributed”.

The authors also state that this is unique compared with other SMLM techniques, but is it also true of DNA-(or other-)PAINT without RESI?

When an average is applied to groups of localizations that are sparse and can be clearly distinguished, this holds true. RESI becomes essential when the labeling is too dense for classical SMLM, including DNA-PAINT, to resolve the individual targets.

The authors have now included references to other, faster but less precise SMLM methods in the introduction, but this comparison was not discussed. Including such a discussion may be beneficial to the general reader.

We have decided to include relevant super-resolution references but not discuss current approaches in greater detail, but rather focus on the advances that RESI contributes. We think that giving a comprehensive overview would be better suited for a review and lead to a less focused storyline of the current manuscript.

Other points:

The authors say they have resolved base pairs at 9.5 Å separation in the introduction (lines 25-26), but I would suggest they could more accurately say the pairs of base pairs they resolved were separated by 7.0 Å (if this is correct).

We rephrased this accordingly: “Excitingly, using RESI, we experimentally resolve the backbone distance of single bases, reported at ~7 Å from crystallographic measurements, at an average separation of 9.5 +/- 2.6 Ångstrom in DNA origami.”

There is a missing localizations spot in Extended Data Fig. 2c, compared with 2d. This is confusing.

One possible explanation of this effect could be transient inaccessibility of the R4 strand. However, this does not pose an issue as 5 out of 6 alignment positions are available.

Extended Data Fig. 3h high-magnification views: Needs scale bars for the left- and right-hand parts of each part of the average result.

We thank the reviewer for highlighting this. The scale bars were moved to the upper left magnified panel to make them more intuitive to find and a statement was added to the figure legend that the same scale applies to all magnifications.

Line 142: It could be good to first mention the 7.0 Å (if that is correct) design distance around here.

We thank the reviewer and agree that this makes the manuscript clearer to follow.

Lines 220-221: Should this say Extended Data Fig. 6a, instead of 6a-c, also 7a instead of 7a-c and 8a instead of 8a-c?

Thank you for pointing us to this issue. We have changed this, mentioning Extended Data Figures 10a (previously 6a), 11a (previously 7a), and 12a (previously 8a) only.

Line 241-242 and Extended Data Fig. 9: It seems a bit strong to say that isolated hexamers are proved to be completely absent, but at least they are not predominant according to the results.

We agree and have modified our statements accordingly.

Line 260: 'two adjacent imagers are never present simultaneously' is only fully true for orthogonally labelled molecules of interest.

Yes, this is correct.

Methods: Imaging parameters and duration: Lines 538-539. This was hard to follow. Can more steps be inserted in the mathematical argument?

We have clarified this in more detail now.

Referee #2:

The authors made an honest effort to improve the manuscript, and only I have a minor comment regarding CD20: In line 192 ".... CD20 in the presence of full antibody ref36" the reference there should be 37 (Kumar et al), and not 36, as in reference 36 the CD20-mAb rings were hypothesised as physiological complexes. Please, also in that paragraph it should be somewhere mentioned that the cryo-EM analyses were made in detergent solutions, otherwise it will be hard for a reader who is not familiar with the cryo-EM papers to understand the problem regarding detergent vs cell contexts.

We thank the reviewer for the supportive review of our revised manuscript. We have implemented the suggested changes.

Referee #1 (Remarks to the Authors, regarding their response to issues raised by referee 3 in initial round of review)

2. Beyond one color

The authors present superb performance for single-color imaging, but there is absolutely no mention of imaging distinct species. It could be argued that this is beyond the scope of the paper, but again, as the authors present the method as a bridge with structural biology, it is a must to distinguish e.g. distinct subunits of a complex. It could also be argued that this might be trivial, as distinct subunits could be targeted with distinct strands directly. The concrete request is, again, to include an analysis of the scalability of RESI for multi color scenarios.

We thank the reviewer for raising this important point. Extension of RESI to "multicolor" imaging of distinct molecular species could be achieved in a straightforward "brute force" approach by assigning specific sets of orthogonal sequences to a specific species. With respect to speeding up this process, existing multiplexing approaches such as spectral demixing etc. could be combined with RESI. As discussed above, there is the option to increase the number of orthogonal sequence species by resorting to non-speed optimized DNA-PAINT sequences.

> I did not see this in the revised manuscript. It would be a useful note to add in.

We decided not to include a discussion about this issue in the main text to keep the text as concise as possible, also considering that the extension to multicolor/multitargets is straightforward.

3. Timings

The authors explicitly mention the long acquisition times of around 100 min for full field of view images of RESI. This counts for figure 2 and 3. They must also explicitly mention the experiments on fig. 4 take around 4.4 hour for each conditions, adding up to ~8-9 hours.

We agree with the reviewer that this should be pointed out more explicitly. We now have added this information to the main text.

> Addressed

The authors should include (e.g. in supplementary figure 4) an approximation of the acquisition time for the sake of experiment design. For example, combining the expected k_{on} of the strands, the density of the expected targets and the desired M repetitions that achieve a given resolution.

We agree with the reviewer that this information is quite useful. We now have added a section to the methods part.

> Addressed ('Imaging parameters and duration')

4. Illumination scheme

It is not reported in these experiments if the illumination is TIRF, HiLo or what kind. I presume it is TIRF. The authors should report the illumination scheme and whether it leads to limitations.

We apologize for this oversight and have now added this information to the methods section. In brief, TIR illumination has been used for the two- and three-dimensional DNA origami data as well as the CD20 acquisition. HILO has been employed for the acquisition of the NPC data.

> Addressed ('Microscope setup'), apart from possible limitations (as far I see), but I would suggest that such limitations are common to DNA-PAINT in general, and may not be needed in this paper.

Minor points

1. I encourage the authors not to use ill-defined terms such as 'ultra-resolution'. This work is still super resolution microscopy, with more resolution than usual. The field suffers from the use of ill-defined terms and metrics.

We agree with the reviewer that ultra-resolution is an ill-defined term. We now changed this accordingly.

> Addressed

2. The authors highlight their scaling with $1/\sqrt{M}$. I encourage them to use log-log scale in figures 1f and 3e to further highlight it, and also make it clear when/if the curves saturate.

We thank the reviewer for this comment. However, we would like to keep the current linear plot of the scaling, as we find this more instructive and intuitive to understand for a broader audience. We experimentally observe no saturation.

- > I agree that the current scale is more intuitive to understand.
- > The markings on the axes are in unusual places though, particularly $\sigma_{\text{RESI}} = 1, 5, 8, 12, \dots$ in Fig. 3e. Should '5' read '4'? Apart from that I can see why they have been chosen this way.
- > **However, this has highlighted that Fig. 1f does not go down to $\sigma_{\text{RESI}} = 0$ (stopping at about $\sigma_{\text{RESI}} = 0.5$). The reader would benefit from the scale going down to zero, so that an illusion is avoided where σ_{RESI} appears to approach zero more closely than the actual estimates.**

We thank the reviewer for this suggestion and have adjusted the axis accordingly. We are grateful for pointing us to a mistake in the axis labeling, we have now changed it to $\sigma_{\text{RESI}} = 0, 5, 10, 15, 20$. We have also adjusted the scale of Fig. 1f to start from 0 rather than 0.5.

3. The 5x and 7x repetitions of the docking strands seem to not play a role in the final resolution. Meaning that the different conformations might be averaged out. This should be commented.

We agree with the reviewer that this seems initially surprising, but have added a statement in the appropriate section of the discussion.

- > I did not find this statement in the discussion.

We have addressed this in the following sentence in the discussion:

"This is achieved due to three specific advantages of DNA-PAINT leading to unbiased target sampling: the rotational flexibility of the target-bound docking strand (even in the case of longer repetitive sequence motifs) and the freely rotating dipole of the dye attached to the imager sequence, as well as the fact that two adjacent imagers are never present simultaneously."

4. Section order. I suggest to exchange figure (and sections) 2 and 3. This yield a smoother cadence for the manuscript, from origami to 'protein rulers', to biological question.

We appreciate the comment regarding the section order but respectfully disagree with the reviewer on this point. We would rather like to keep the current order based on the following logic: Section 2 is concerned with testing of RESI with well-established in vitro and cellular systems. Section 3 shows for the first time sub 1-nm resolution in optical microscopy, one of the most intriguing results of our work.

- > I am happy with the order not being changed.

5. Channel alignment. If DNA origami requires an element by element fiducial realignment for the different exchange round, how is this supposed to affect cell measurements? Is one expected to make local drift corrections based on some reference? Is this the ultimate limitation for measurements in cells?

This points to a very interesting question. We note that due to the nature of the surface attachment

chemistry (BSA/Biotin/Streptavidin/Biotin/Origami), the DNA origami structures tend to undergo translational and rotational movement in the lower single-digit nanometer scale on the surface. However, we argue that this limitation is actually less of an issue for cellular experiments, where samples are covalently crosslinked by fixation with many more surface attachment points as compared to DNA origami structures.

> I do not feel this must be added to the paper.

6. Introduction. The authors mention "Collecting an arbitrarily large number of localizations yields an arbitrary increase in precision, ultimately limited solely by the size of the labeling probe." It should be somehow clarified, especially for the non-expert, that this will indeed yield an extremely precise localization, but with a bias/offset from the epitope of interest.

We agree with the reviewer and have now modified the text also with regards to points raised by reviewer #1. For completeness, we here state our response again: The impact of the label size on structural measurements is of high relevance to the reader: The finite label size and thus potentially biased orientation leads to linkage error-induced inaccuracies in distance measurements. We have now added a sentence to point the reader to this fact.

> This is added at the end of 'RESI resolves single nuclear pore complex proteins'. The discussion also alludes to this ('RESI performance and accuracy...'), but could benefit from a more direct reminder of this limitation as well.

7. Discussion. The authors mention "even in environments where achieving high optical localization precision is impossible". This is somewhat vague. To what scenarios do the authors refer? Exemplify. Include numbers and expectations.

We agree with the reviewer that this is somewhat of a vague statement. As we have included more discussion about the requirements for efficient stochastic RESI labeling, we now decided to remove this comment in the discussion section.

> Ok

8. Discussion. The authors mention "under physiological conditions" for the CD20 measurements. What do the authors mean? The cells are fixed.

We thank the reviewer for pointing us to this potentially confusing point. We have now clarified this by mentioning that measurements have been performed in cells rather than in vitro.

> Ok

Final comment

On quality of figures, clarity of the presentation and message, use of statistics, modelling of nearest neighbor distances and referencing the work is at the highest standards.

We are very grateful and honored by the appreciation of our work by the reviewer and thank her/him again for the very supportive assessment of our work.